PREPARED FOR SUBMISSION TO JHEP

# Quantum vorticity: a not so effective field theory

**Gabriel Cuomo,**[a,b] **Fanny Eustachon,**[c] **Eren Firat,**[d] **Brian Henning,**[e,f] **Riccardo Rattazzi**[d]

[a]*Center for Cosmology and Particle Physics, Department of Physics, New York University, New York, NY 10003, USA*

[b]*Department of Physics, Princeton University, Princeton, NJ 08544, USA*

[c]*CPHT, CNRS, École polytechnique, Institut Polytechnique de Paris, 91120 Palaiseau, France*

[d]*Theoretical Particle Physics Laboratory (LPTP), Institute of Physics, EPFL, Lausanne, Switzerland*

[e]*Department of Physics, University of California, Santa Barbara, CA 93106, USA*

[f]*Kavli Institute for Theoretical Physics, University of California, Santa Barbara, CA 93106, USA*

*E-mail:* gc3265@nyu.edu, fanny.eustachon@polytechnique.edu, eren.firat@epfl.ch, bqhenning@gmail.com, riccardo.rattazzi@epfl.ch

ABSTRACT: We provide a comprehensive picture for the formulation of the perfect fluid in the modern effective field theory formalism at both the classical and quantum level. Due to the necessity of decomposing the hydrodynamical variables $(\rho, p, u^\mu)$ into other internal degrees of freedom, the procedure is inherently not unique. We discuss and compare the different inequivalent formulations. These theories possess a peculiarity: the presence of an infinite dimensional symmetry implying a vanishing dispersion relation $\omega = 0$ for the transverse modes. This sets the stage for UV-IR mixing in the quantum theory, which we study in the different formulations focussing on the incompressible limit. We observe that the dispersion relation gets modified by quantum effects to become $\omega \propto \mathbf{k}^2$, where the fundamental excitations can be viewed as vortex-anti-vortex pairs. The spectrum exhibits infinitely many types of degenerate quanta. The unusual sensitivity to UV quantum fluctuations renders the implementation of the defining infinite symmetry somewhat subtle. However we present a lattice regularization that preserves a deformed version of such symmetry.

## 1  Introduction and summary

Our modern understanding is that the laws of physics are fundamentally quantum mechanical and that classical physics can only emerge, under some circumstances, as an effective approximation. Nevertheless in the everyday practice of theoretical physics and also in its teaching, we often first present a classical system and then consider its so-called quantization. While this way of thinking may, in principle, be philosophically incorrect, it does work well in practice for basically all mechanical systems. The harmonic oscillator, the particle in a coulombic potential and the electromagnetic field all represent classical systems with a consistent, and interesting, quantized version.

The present paper considers instead a system, the perfect fluid, for which things do not appear to work as smoothly as in all those other cases when going to its quantized version. The difficulty is intrinsically associated with the *absolute* softness of the vorticity modes, whose linearized classical dispersion relation around the stationary configuration is just $\omega = 0$ for arbitrary $\mathbf{k}$. This implies, classically, the failure of the usual association of short wavelengths and high frequencies, and in fact underlies the phenomenon of turbulence where motions on long scales evolve into motions at ever shorter scales. In field theoretic terms, such degenerate dispersion relation indicates that the UV does not fully decouple from the IR. At the same time, experience teaches that the softer a classical mode the larger its quantum mechanical fluctuations. All these facts make for a system where UV quantum fluctuations can play a different and bigger role than in ordinary effective quantum field theory. Finding out how

this difference is realized, and also how dangerously so, seems to us an intriguing question to explore per se. But the knowledge of situations at the edge of the ordinary EFT arena may perhaps, in the long run and optimistically, acquire exportable value. For instance, EFT is not only a solid platform where addressing questions like cosmological constant and Higgs mass hierarchy, but also a golden cage limiting our imagination to address them. In this last respect any different instance of the interplay between UV and IR may have value. Finally, it is not excluded that the question we are considering may have direct phenomenological consequence for finite density systems: does the universality class of the quantum mechanical zero temperature fluid exist?

This is not the first paper trying to address the question we just outlined and it follows a few studies performed over the last decade [1–4]. As a matter of fact also Landau, in his original paper on superfluid Helium [5], was faced with the same question, though we think, his answer was not correct. Luckily also his starting hypothesis was not quite correct, in such a compensating way that the resulting dynamics correctly described the superfluid universality class. We will comment in more detail on previous work in the text, but the present article is a direct follow up to ref. [4], whose results it clarifies and extends in various ways. One main result is that like there exist two not fully equivalent descriptions of the classical perfect fluid, there are two not fully equivalent quantum theories. These have the same energy levels, but in a different Hilbert space, i.e. with inequivalent multiplicities. Another result is that the degenerate dispersion relation of the vorticity modes is indeed lifted by quantum effects to $\omega \propto \mathbf{k}^2$. However the infinite symmetry strikes back by predicting infinitely many types of such quanta. A detailed summary of our results is contained in the next two subsections.

Euler's perfect fluid is arguably the oldest example of a classical field theory; yet, by the breakdown of the decoupling between UV and IR, its quantum formulation is significantly more subtle than that of many later-developed examples, such as Maxwell theory. However, this phenomenon is not unique to Euler's theory. Recent research has extensively explored the continuum limit of various lattice models of fractons—excitations with restricted mobility (see, e.g., [6–9]). Also the universal continuum QFT descriptions of these models depart from the standard EFT paradigm, and short distance modes do not fully decouple at low energy. Similar to the case of fluids, this behavior is often linked to the presence of soft modes in the naive classical field theory formulation. Other similarities include the existence of infinitely dimensional *exotic symmetries*, an extensive degeneracy of low-energy states, and the lack of (linearly realized) Lorentz symmetry. In this context, our work presents perhaps the most surprising example of a local quantum system exhibiting exotic UV/IR connections.[1]

Finally, we must comment on the relation of our work with the vastly more developed domain of ordinary imperfect fluids, which are ubiquitous in nature. Technically hydrodynamics universally describes the transport of energy, momentum and possibly other conserved charges in systems at finite temperature. In the context of thermal QFT the study has mostly

---

[1]It was pointed out in [10] that the motion of vortices presents several features common to other fracton excitations, including the conservation of the dipole and the quadrupole moment. In this light, perhaps, the similarities between the quantum mechanics of vorticity and that of other fracton systems is not so surprising.

focussed on linear response theory, where remarkable results have been obtained, for instance, in the prime principle computation of transport coefficients [11–15] in weakly coupled QFTs. The study of strongly coupled hot QFT through the AdS/CFT [16, 17] has then sparked the development of an effective field theory approach to hydrodynamics [18–21]. That is based on a Lagrangian construction but in the Schwinger-Keldysh (SK) double time formalism, suitable for describing quantum as well as statistical fluctuations. Although the resulting Lagrangian shares properties with one of the two perfect fluid descriptions we consider in this paper, in particular the symmetries, its interpretation and use are quite different. The effective SK Lagrangian for fluids is just the generator of long range thermal correlators, whose singularities do not have the intepretation of quanta excited from a ground state. In other words the system described by that Lagrangian "should not be quantized". That is the reason why the results of our study do not apply to ordinary fluids, at least not strictly. Nonetheless, as we mention in the final outlook, our study also invites further questions on the Lagrangian description of thermal hydrodynamics.

## 1.1   Summary of part I

In Part I of this work, we provide a detailed discussion of the classical perfect fluid and of its (inequivalent) Lagrangian formulations. While this is mostly a review, the presentation is original and includes some novel derivations (such as the derivation of the Clebsch formulation starting from the comoving coordinates' description in sec. 4.1). These results will provide the starting point of our analysis of the quantum perfect fluid(s) in Part II of this work.

The perfect fluid stress tensor is characterized by the energy density $\rho$, the pressure density $p$, and the four-velocity $u^\mu$. The pressure is related to the energy density by an equation of state $p = p(\rho)$. The dynamics is then specified by Euler's equations, which are nothing but the conservation of the stress energy tensor. It turns out that such a simple system of equations admits several nontrivial properties. First, linearized fluctuations consists of a sound mode, with dispersion relation $\omega(\mathbf{k}) = c_s|\mathbf{k}|$, and a transverse mode with trivial dispersion, $\omega(\mathbf{k}) = 0$. Additionally, the absence of dissipation results into several conservations laws, including the conservation of a nontrivial current, normally identified with the entropy current, and the convective conservation of vorticity (Helmholtz theorem). We review these and other nontrivial consequences of Euler's equation in sec. 2.

The description of the perfect fluid based on the classical equations of motion is phenomenologically satisfactory from some point of views. However, for our purposes it is important to work with a Lagrangian field theory formulation, so as to be able to apply path-integral methods in our study of the quantum perfect fluid. Additionally, as emphasized in [22, 23], having a Lagrangian formulation for perfect fluids is also useful at the classical level. This is because from a modern perspective, hydrodynamics is an effective field theory, and therefore Euler's equations are expected to receive corrections at higher order in derivatives. A Lagrangian description simplifies and systematizes the task of finding consistent modifications

of Euler's equations.[2]

We are therefore led to discuss the construction of Lagrangians for the perfect fluid. It turns out that cannot be carried out purely in terms of the energy density and of the velocity [25]. This is obvious in odd spacetime dimensions, since Euler's equation require an odd number of initial conditions while Lagrangian systems describe canonical pairs, but it remains true also in even dimensions.[3] One is thus forced to introduce additional *internal* fields, whose physical interpretations we discuss in the main text.

There are two inequivalent ways to introduce such additional variables, which we review, respectively, in sec. 3 and sec. 4. These two formulations lead to equivalent equations of motion for the hydrodynamic variables $\rho$ and $u_\mu$, also at higher order in derivatives, but they have different phase spaces and dynamics as a whole. In these formulations, the properties of the perfect fluid are encoded in peculiar infinite dimensional symmetry groups that act on the internal fields. These symmetries ensure that the equations for the fluid's density and velocity do not depend explicitly on these variables, and allow recovering the infinitely many stationary solutions of Euler's equations (as we discuss in detail in sec. 4.4). We will also show that these symmetries imply that a generalization of Helmholtz theorem holds to all order in derivatives.

We begin our discussions of fluid Lagrangians in sec. 3 by reviewing the "comoving" formulation following ref. [22]. In this formulation the internal variables $\varphi^I(t, \mathbf{x})$ are intuitively related with the infinitesimal *fluid elements*, whose trajectory specifies the fluid's motion. The defining physical property of the fluid—that its energy remains constant under deformations that do not involve compression and dilation—is encoded in the invariance of the action under volume-preserving diffeormophisms that act on the fluid elements $\varphi^I(t, \mathbf{x})$. This infinite dimensional symmetry group is spontaneously broken by a specific fluid configuration, and the internal fields are thus interpreted as the corresponding Goldstones [26]. The nonlinear realization of the internal symmetry is the origin of the trivial dispersion relation $\omega(\mathbf{k}) = 0$ of the transverse mode. We also show that, reminiscently of some recently studied fractonic systems [8, 9, 27], the volume-preserving diff. symmetry is responsible for the existence of a tensor symmetry in the language of ref. [27], which is equivalent to the convective conservation of vorticity.

We then move on to discuss the "Clebsch" formulation of the fluid in sec. 4. Focusing on the physical cases of two and three spatial dimensions, we derive the Clebsch action starting from the comoving formulation of sec. 3 by requiring that it describes the same hydrodynamic flows. Also the Clebsch fields are invariant under an infinite dimensional symmetry group, whose algebra coincides with that of area-preserving diffeomorphisms. However, unlike in the

---

[2]See [24] for a review of the standard approach to (dissipative) fluids based on "constituent relations". We also remark that it is known that not all possible non-dissipative modifications of the constituent relation (2.1) can be obtained from a Lagrangian description [20]. We take the perspective that a perfect fluid is *defined* as a system whose equations of motion can be obtained by a Lagrangian formulation and reduce to Euler's equations (2.4) at leading order in derivatives.

[3]In three spatial dimensions the problem is related with the conservation of helicity (2.18) [25].

comoving formulation, such group is linearly realized by the fluid at rest. The most important technical result of sec. 4 is the action for the fluid in the incompressible limit, which we derive in sec. 4.3 using the well-known duality between a shift invariant scalar field and a $d-1$-form gauge field. The simplicity of the Clebsch Lagrangian in the incompressible limit will be one of the main advantages of this formulation when studying the quantum theory. We also discuss the physical interpretation of the Clebsch variables and the relation between the area-preserving diff. symmetry and the stationary solutions of Euler's equations.

We should comment that the situation for the perfect fluid, where there exist several inequivalent Lagrangian descriptions that yield Euler's equations, is completely analogous to that of a rotating rigid body at fixed center of mass position. In that case as well the equations of motions are written purely in terms of the three angular velocities, with no reference to the angular coordinates of the body, which however appear in the Lagrangian. In fact, it is also possible to write different actions that yield the same Hamiltonian and equations for the angular velocities, but whose Lagrangian coordinates are not invariant under rotations (but they are under different symmetry groups) and cannot be thought therefore as the angular coordinates of a rigid body. We review these basic facts about the rigid body in appendix A.[4]

## 1.2   Summary of part II

In Part II we discuss the quantum perfect fluid, or more precisely the different quantum perfect fluids. Indeed, an important implication of the discussion of Part I of this work is that the theory of the quantum perfect fluid is itself ambiguous, and depends on which classical Lagrangian formulation we take as a starting point.[5] The difference between the two choices is subtle. Indeed, both the comoving and the Clebsch systems have formally identical Hamiltonians when expressed in terms of the fluid variables. This implies that both systems must have the same energy spectrum at the quantum level. However, the degeneracies of the states—and more generally the structure of the Hilbert space—need not to match. This is indeed what we will find.

One might wonder if there is a preferred physical choice between these two options. We shall be agnostic about this question, and report our results for both the comoving and the Clebsch fluid.

We also remark that, since the quantization of the longitudinal modes doesn't present particular difficulties, in Part II of this paper we focus on the transverse modes by taking the incompressible limit, in which the sound mode is integrated out.

---

[4]Notice that, as for the rigid body, Euler's eqs. (2.3) admit a natural Hamiltonian formulation without reference to additional internal variables, but with Poisson brackets that do not provide an invertible symplectic structure [25, 28–32]. We will not review the Hamiltonian formulation of the perfect fluid in full generality, but we will derive and analyze the Hamiltonian for the incompressible fluid in Part II of this work.

[5]In this work we only deal with perfect, non-dissipative, fluids, and we only consider standard Lagrangian formulations. For dissipative fluids the EFT is formulated on a Schwinger-Keldysh contour, see [18, 19], and thus admits a less straightforward interpretation in terms of a Hilbert space and includes extra couplings compared to the perfect fluid. In particular, as well known, the existence of viscosity gives rise to the diffusion pole $\omega(\mathbf{k}) \simeq -iD\mathbf{k}^2$ in the propagator of the pathological mode.

As reviewed above, the challenges in the quantization of the perfect fluid arise due to the existence of a mode with flat dispersion relation $\omega_{\mathbf{k}} = 0$ for any momentum $\mathbf{k}$. In [4] these difficulties were overcome for the two-dimensional incompressible fluid in the comoving formulation. The analysis of the quantum fluid in the comoving formulation is however rather non-straightforward. At the conceptual level, the complications originate in the fact that the classical spontaneous breaking of the volume preserving diff. symmetry is inconsistent at the quantum level. Indeed, the absence of gradient energy for the transverse modes makes a perturbative treatment based on weakly coupled Fock states inadequate. Correspondingly, quantum effects lead to symmetry restoration analogously to nonlinear sigma models in two spacetime dimensions [1]. To study the comoving fluid, the authors of [4] resorted to a discretized model of $N \times N$ matrices, that approaches the continuum theory for $N \to \infty$. It was found that the ground-state is a singlet of the symmetry group and is unique. The main prediction of [4] is that such quantum theory includes gapless excitations, dubbed vortons. These states form an infinite dimensional representation of the area-preserving symmetry group, which is linearly realized at the quantum level, and admit a quadratic dispersion relation $\omega_{\mathbf{k}} \propto \mathbf{k}^2$. The existence of such states, however, is far from obvious from the classical formulation reviewed in sec. 3.

It turns out that the quantization of the fluid in the Clebsch formulation, whose classical aspects are reviewed in sec. 4, is a much more straightforward task. The technical simplifications originate principally from the linear realization of the area-preserving symmetry of the Clebsch fields on the classical ground state. Quantum mechanically that leads to a tractable Fock space structure. All the essential results can then be derived directly in the continuum theory. This is the topic of sec. 5 of this work.

Our results confirm the findings of [4]: the existence of vorton states with quadratic dispersion relation $\omega_{\mathbf{k}} \propto \mathbf{k}^2$. This was a foreordained conclusion: as formerly explained, the comoving and Clebsch systems have formally identical Hamiltonians when expressed in terms of the fluid variables. We also find that the vorton states are infinitely degenerate. Such degeneracy arises however in a different way than in the comoving description: the Hilbert space admits a familiar Fock structure, but for every integer $n > 1$ there exist a $n$-particle bound state, formally made of vortons with completely overlapping wave-function, which is degenerate and essentially identical to the single vorton state. We also argue that, unlike the comoving system, the ground-state breaks spontaneously the symmetry group of the fluid and is thus infinitely degenerate at finite volume. These predictions are robust consequences of the symmetries of the model.

The mechanism underlying the existence of these infinitely degenerate bound-states relies on the fact that the infinite symmetry is valid at arbitrary short-distance, in other words it is exact and not just emergent in the IR. It is therefore similar to the UV/IR mixing phenomena that occur in certain exotic theories that recently attracted some attention in relation to fracton physics [8, 9]. As in those cases, it is desirable to have a lattice discretization of the incompressible fluid, that can be regarded both as a regulated version of the continuum model and as a specific UV completion.

Motivated by these considerations, in sec. 6 we discuss in detail how the predictions of sec. 5 are borne out using a version of the discretized model of [4] in terms of the Clebsch fields. The discretized model is derived by requiring that it preserves a modified version of the symmetries of the Clebsch theory. Unlike most lattice theories, this requirement forces us to consider somewhat non-local interactions. i.e. not involving just nearest neighbours, and reducing to a local system only at very low momenta. Therefore this system is perhaps best understood as a matrix quantum-mechanics rather than a lattice model. The discretized model will also allow us to connect the Clebsch and the comoving formulations of the quantum fluid and to compare the two.

In sec. 7 we discuss two generalizations of our results. First, we discuss a model with formally the same Hamiltonian as the two-dimensional incompressible fluid, but in which the vortons are fermions. We will find that also in this case there exist infinitely many bound states degenerate with the single-vorton particle. We will then argue that the quantum fluid in three spatial dimensions behaves analogously to the two-dimensional case. In particular, it features an infinitely degenerate gapless state with quadratic dispersion relation both in the comoving and in the Clebsch formulation. In sec. 7.3 we compare our findings with previous works.

To summarize, the quantum perfect fluid, both in the comoving and the Clebsch formulation, displays highly unusual features, most notably an infinitely degenerate spectrum. While theoretically interesting, it seems unlikely that a system with such peculiarities may be realized experimentally. One might wonder if it is at least possible to construct a system that features states that resemble the vorton particles in all other aspects but the degeneracy, i.e. states with the same low energy dispersion dispersion relation and interactions as the vortons.

We report on partial progress on this question in sec. 8, where we show that a particle anti-particle bound state in 2+1 QED, an analogue of positronium, reproduces some of the features of the vorton theory. However, the analogy works only up to a point. Indeed, the 2+1 dimensional positronium theory features some contact interactions that are forbidden in the vorton model by the area-preserving diff. symmetry. This (expected) mismatch further illustrates how special the quantum perfect fluid is compared to an ordinary QFT.

# Part I
# Classical perfect fluid

## 2   Review of fluid mechanics

### 2.1   Relativistic fluids

Euler's equations describe an idealized fluid where viscosity and heat conduction are neglected. The fluid's motion is completely specified by conservation of energy and momentum. In detail,

one parametrizes the stress tensor as [33]

$$T^{\mu\nu} = (\rho + p)u^\mu u^\nu + pg^{\mu\nu}. \tag{2.1}$$

where $\rho$ is the relativistic energy density and $p$ the pressure of the fluid, while $u^\mu$ is the relativistic velocity vector, such that $u^\mu u_\mu = -1$. The pressure and the energy density are related by the equation of state

$$p = p(\rho). \tag{2.2}$$

The relativistic Euler's equation are nothing but the conservation of the stress energy tensor:

$$\partial_\mu T^{\mu\nu} = 0. \tag{2.3}$$

Combining (2.1) and (2.3), we get more explicitly:

$$u^\nu \partial_\mu(\rho u^\mu) + \rho u^\mu \partial_\mu u^\nu + p\partial_\mu(u^\mu u^\nu) + (g^{\nu\mu} + u^\nu u^\mu)\partial_\mu p = 0 \tag{2.4}$$

In $d$-spatial dimensions, the equations (2.4) provide a set of $d+1$ first order differential equations for $d+1$ independent variables ($\rho$ and $u^i$), and thus completely specify the motion of the fluid.[6]

It is straightforward to solve (2.3) for small fluctuations around the static flow, expanding as $\rho = \rho_0 + \delta\rho(t, \mathbf{x})$ and $u^\mu \simeq \delta_0^\mu + \delta_i^\mu \delta u^i(t, \mathbf{x})$. One finds that the spectrum of fluctuations consist of a *sound* longitudinal mode with dispersion relation $\omega(\mathbf{k}) = c_s|\mathbf{k}|$, where $c_s^2 = p'(\rho_0)$, and (for $d > 1$) a transverse mode with $\omega(\mathbf{k}) = 0$. As we will explain in the second part of this paper, the degenerate nature of the dispersion relation of the transverse mode is what makes the quantum mechanics of the perfect fluid interesting and non-trivial.

Rather than considering the energy density $\rho$ as an independent field, it is customary to work in terms of an auxiliary variable $T$, whose physical interpretation we discuss below. The equation of state (2.2) is then replaced by two equations $\rho = \rho(T)$ and $p = p(T)$. To specify such equations in terms of the equation of state, we also introduce another function $s(T)$ and demand that

$$\rho + p = sT, \quad dp = sdT \quad \implies \quad d\rho = Tds. \tag{2.5}$$

For instance, given $p = p(T)$, these relations imply $s = p'(T)$ and $\rho = p'(T)T - p(T)$. Equivalently, given $\rho = \rho(s)$, we find $T = \rho'(s)$ and $p = s\rho'(s) - \rho(s)$. Using eqs. (2.5) and the EOM (2.3), one can easily prove the existence of a conserved current defined as

$$S^\mu = su^\mu \quad \implies \quad \partial_\mu S^\mu = -\frac{1}{T}u_\nu \partial_\mu T^{\mu\nu} = 0. \tag{2.6}$$

---

[6]The fluid we are describing is the simplest featureless fluid. One often considers in addition to the system (2.4) an extra variable $n$, associated with a conserved current $J^\mu = nu^\mu$ (see e.g. [23, 33]), which is distinguished from the entropy current that we discuss below. The equation of state then provides one relation between $n$, $\rho$ and $p$, while the conservation of the energy-momentum tensor and $J^\mu$ provide $d+2$ equations for the $d+2$ dynamical variables ($\rho$, $u^i$ and $n$). In this work we do not consider such fluids, and restrict ourselves to the minimal setup consisting solely of $d+1$ independent hydrodynamical quantities.

It is customary to interpret $T$ as the temperature of the fluid, and $s$ as the entropy density. Eqs. (2.5) are then the usual thermodynamic relations, and $S^\mu$ is the entropy current. The conservation of the latter indeed ensures the absence of dissipation in a perfect fluid.[7]

The existence of a conserved current however also allows for an alternative interpretation of eqs. (2.5) and (2.6). Namely, we can suppose we are dealing with a system at zero temperature possessing a conserved $U(1)$ charge. Then, consistently with eqs. (2.5), we can identify $T$ with the chemical potential and $s$ with the $U(1)$ charge density. This viewpoint is less common since it does not generalize to dissipative fluids, for which $\partial_\mu S^\mu \neq 0$, but is nonetheless discussed sometimes, see e.g. [22, 34, 35] and references therein.

For our purposes, it is important to remark that, as long as we are concerned with a perfect fluid, Euler's equations imply the existence of a conserved current. For definiteness we shall refer to $S^\mu$ as entropy current, but we shall remain agnostic about its physical interpretation.

Using $s$ and $T$ we can recast Euler's equations in a different form. To this aim, we define the antisymmetric vorticity tensor in terms of the quantities above as

$$\omega_{\mu\nu} = \partial_\mu \left( \frac{\rho + p}{s} u_\nu \right) - \partial_\nu \left( \frac{\rho + p}{s} u_\mu \right) = \partial_\mu (T u_\nu) - \partial_\nu (T u_\mu) . \tag{2.8}$$

It is possible to show that[8]

$$\omega_{\mu\alpha} u^\alpha = 0 . \tag{2.10}$$

Eqs. (2.6) and (2.10) provide $d + 1$ independent equations and are therefore equivalent to Euler's equations (2.3).

In perfect fluids, the vorticity is conserved along the fluid's flow. This property corresponds to the vanishing of the Lie derivative of $\omega_{\mu\nu}$ with respect to $u^\mu$:

$$(\mathcal{L}_u \omega)_{\mu\nu} \equiv u^\alpha \partial_\alpha \omega_{\mu\nu} + \omega_{\mu\alpha} \partial_\nu u^\alpha + \omega_{\alpha\nu} \partial_\mu u^\alpha = 0 . \tag{2.11}$$

To prove this relation, one uses that $\omega$ is an exact form to rewrite the Lie derivative as $(\mathcal{L}_u \omega)_{\mu\nu} = \partial_\nu (\omega_{\mu\alpha} u^\alpha) - \partial_\mu (\omega_{\nu\alpha} u^\alpha)$, which vanishes by eq. (2.10).[9] Note that, by the same

---

[7]Note that, in terms of $s$, $T$ and $u^\mu$, the Euler equation consists of eq. (2.6) plus the equation $u^\alpha \partial_\alpha (T u_\mu) = -\frac{1}{s} \partial_\mu p = -\partial_\mu T$. The latter is equivalent, in form language, to the following geometric equation:

$$\mathcal{L}_u (Tu) = -\frac{1}{s} dp = -dT . \tag{2.7}$$

Here $u_\mu$, and hence $T u_\mu$, are one-forms. Remarkably, eq. (2.7) holds also for fluids with conserved charges.

[8]This can be shown as follows

$$\begin{aligned} s u^\mu \omega_{\mu\alpha} &= s u^\mu \partial_\mu (T u_\alpha) - s u^\mu \partial_\alpha (T u_\mu) \\ &= s u^\mu u_\alpha \partial_\mu T + s T u^\mu \partial_\mu (u_\alpha) - s u^2 \partial_\alpha (T) - s T \partial_\alpha (u^2/2) \\ &= u^\mu u_\alpha \partial_\mu p + (\rho + p) u^\mu \partial_\mu (u_\alpha) + \partial_\alpha (p) \\ &= \text{Euler's equation } = 0 , \end{aligned} \tag{2.9}$$

where we used $u^2 = -1$, $\rho + p = sT$ and $dp = sdT$.

[9]As $d\mathcal{L}_v(\cdots) = \mathcal{L}_v d(\cdots))$, eq. (2.11) also immediately follows from the geometric eq. (2.7).

argument, eq. (2.10) also implies that the Lie derivative of the vorticity along any vector parallel to the fluid velocity vanishes: $\mathcal{L}_v \omega = 0$ for any $v_\mu \propto u_\mu$.

To understand the physical meaning of eq. (2.11), consider a 1-parameter family of closed curves $\mathcal{C}(\tau)$ that is generated from a initial closed curve $\mathcal{C}(0)$ nowhere tangent to $u$ by displacing each point of $\mathcal{C}(0)$ by some distance along the field lines of a vector $v \propto u$.[10] Then, considering any surface $S(\tau)$ enclosed by the curve $\mathcal{C}(\tau)$, i.e. satisfying $\partial S = \mathcal{C}$, and using Stokes' theorem, we have

$$\frac{d}{d\tau} \oint_{\mathcal{C}(\tau)} T u_\mu dx^\mu = \frac{d}{d\tau} \int_{S(\tau)} \omega_{\mu\nu} dx^\mu \wedge dx^\nu = 0 \,, \tag{2.12}$$

where in the last equality we used that $\mathcal{L}_v \omega = 0$ for any vector $v \propto u$. Choosing $v = u$, eq. (2.12) states that the circulation of the vector $T u^\mu$ around a closed curve is conserved along the fluid's flow, and is therefore the relativistic version of Kelvin's vorticity theorem. Eq. (2.11) is sometimes referred to as Helmholtz theorem or Kelvin-Helmholtz theorem.

Let us finally make some comments specifically about fluids in two and three spatial dimensions. In two spatial dimensions, there is surprisingly an infinite set of conserved quantities besides those that follow from the energy-momentum tensor and the entropy current. To see this, note that in two spatial dimensions eq. (2.10) may be solved as

$$\omega_{\mu\nu} = \varepsilon_{\mu\nu\rho} u^\rho \Omega s \,, \tag{2.13}$$

where $\Omega$ is a scalar defined by this relation and we introduced a factor of $s$ for future convenience. Then, since $\omega$ is an exact form, we obtain

$$\partial_\mu (u^\mu \Omega s) = 0 \quad \overset{\partial_\mu S^\mu = 0}{\Longrightarrow} \quad u^\mu \partial_\mu \Omega = 0 \,. \tag{2.14}$$

It follows that we have infinitely many conserved quantities obtained integrating over an arbitrary spacelike surface $\Sigma$ as

$$C^{(k)} = \int_\Sigma d^2 \Sigma_\mu S^\mu \Omega^k \,, \tag{2.15}$$

for arbitrary values of $k$. The quantities (2.15) are invariant under arbitrary deformations of the spacelike surface $\Sigma$. For $k = 0$, eq. (2.15) reduces to the conservation of entropy, while for $k = 1$ it gives the conservation of the total vorticity. The $C^{(k)}$'s are sometimes referred to as the Casimirs of the fluid (see e.g. [4, 25] for further details).

In three spatial dimensions there is only one conserved Casimir, besides the obvious conservation laws associated with the spacetime symmetries and the entropy current. This can be obtained by noticing that from (2.10) it follows that

$$\varepsilon^{\mu\nu\rho\sigma} \omega_{\mu\nu} \omega_{\rho\sigma} \propto \omega_{\mu\nu} u^\nu \varepsilon^{\mu\lambda\rho\sigma} u_\lambda \omega_{\rho\sigma} = 0 \,, \tag{2.16}$$

---

[10]Explicitly, parametrizing the curve $\mathcal{C}(\tau)$ via an affine parameter $\sigma \in (0, 1)$ and coordinates $x^\mu(\tau, \sigma)$ such that $x^\mu(\tau, 0) = x^\mu(\tau, 1)$, this means that the tangent vector $t^\mu = \frac{\partial x^\mu}{\partial \sigma}$ satisfies $\frac{\partial t^\mu}{\partial \tau} = t^\nu \partial_\nu v^\mu$.

which, because of the definition (2.8), is equivalent to the conservation equation

$$\partial_\mu \left( \varepsilon^{\mu\nu\rho\sigma} T u_\nu \omega_{\rho\sigma} \right) = 0 \,. \tag{2.17}$$

We therefore obtain that the fluid helicity, defined as

$$C = \int_\Sigma d^3\Sigma_\mu \varepsilon^{\mu\nu\rho\sigma} T u_\nu \omega_{\rho\sigma} \,, \tag{2.18}$$

is conserved, i.e. it is invariant under deformations of $\Sigma$.[11]

## 2.2 Non-relativistic and incompressible fluids

A non-relativistic fluid is one where $p/\rho \ll 1$ over a significant range of $s$ [36], or equivalently over a significant range of $T$. As, by eq. (2.5), $p = \rho'(s)s - \rho(s)$, a non-relativistic fluid is then defined by the condition $s\rho'/\rho \simeq 1$, which, by conveniently restoring powers of the speed of light $c >> 1$, is solved by writing

$$\rho(s) = mc^2 s + \epsilon(s) + O\left(\frac{1}{c^2}\right) \,. \tag{2.21}$$

Here $m$ is an arbitrary constant and $\epsilon(s) = O(c^0)$ can be interpreted as the non-relativistic energy density. Similarly we have $p = \rho'(s)s - \rho(s) = \epsilon'(s)s - \epsilon(s) = O(c^0)$ while for the speed of sound we have

$$\frac{c_s^2}{c^2} = \frac{s\rho''(s)}{\rho'(s)} \simeq \frac{s\epsilon''}{mc^2} \ll 1 \,. \tag{2.22}$$

Note that if we interpret $s$ as the conserved particle number and $m$ as the mass of the elementary constituents, eq. (2.21) simply states that the energy density of the fluid consist of the sum of the rest mass contribution $\rho_m \equiv ms$ and the non-relativistic energy.

Using the equation of state (2.21) and defining the spatial velocity in the usual way,

$$u^\mu = \frac{1}{\sqrt{1 - \mathbf{v}^2/c^2}} \left( 1, \frac{\mathbf{v}}{c} \right) \,, \tag{2.23}$$

---

[11]This pattern persists in higher dimensions. For instance, in four spatial dimensions we can construct conserved quantities similarly to eq. (2.15) from a scalar $\Omega$ defined as

$$\varepsilon^{\lambda\mu\nu\rho\sigma} \omega_{\mu\nu}\omega_{\rho\sigma} = s u^\lambda \Omega \quad \Longrightarrow \quad u^\mu \partial_\mu \Omega = 0 \,. \tag{2.19}$$

In five spatial dimensions we instead have a unique conserved Casimir that follows from the conserved helicity current, which is now defined as

$$j_{hel}^\delta = \varepsilon^{\delta\lambda\mu\nu\rho\sigma} T u_\lambda \omega_{\mu\nu}\omega_{\rho\sigma} \quad \Longrightarrow \quad \partial_\mu j_{hel}^\mu = 0 \,. \tag{2.20}$$

Generalizing, we have an infinite number of conserved Casimirs in even dimensions, and only one in odd dimensions.

we obtain the non-relativistic Euler's equations by taking the limit $c \to \infty$ of (2.3). These consist of the mass continuity equation[12]

$$\frac{\partial s}{\partial t} + \boldsymbol{\nabla} \cdot (s\mathbf{v}) = 0 \,, \tag{2.25}$$

and of the momentum conservation equation

$$\frac{D\mathbf{v}}{Dt} \equiv \frac{\partial \mathbf{v}}{\partial t} + (\mathbf{v} \cdot \boldsymbol{\nabla}) \mathbf{v} = -\frac{\boldsymbol{\nabla} p}{s} \,, \tag{2.26}$$

where $\frac{D}{Dt}$ is the so-called material derivative. From this equation we derive Kelvin's theorem:

$$\Gamma(t) = \int_{\mathcal{C}(t)} \mathbf{v} \cdot d\mathbf{x} \quad \Longrightarrow \quad \frac{D\Gamma}{Dt} = 0 \,, \tag{2.27}$$

which is the non-relativistic version of eq. (2.12).

Finally, it will often be convenient for us to work in the incompressible limit. Physically, this amounts to considering flows where the fluid velocity $|\mathbf{v}|$ is much smaller than the speed of sound so that the propagation of sound waves is effectively instantaneous. Therefore, as we will make explicit in sec. 4.3, the sound mode can be integrated out [1] and the dynamics of the fluid is completely specified by the vorticity.

At the technical level, the incompressible limit is obtained by setting $s = \text{const.}$ in the nonrelativistic Euler's equations (2.25) and (2.26), which therefore reduce to

$$\boldsymbol{\nabla} \cdot \mathbf{v} = 0 \quad \text{and} \quad \frac{D\mathbf{v}}{Dt} + \frac{\boldsymbol{\nabla} p}{s} = \frac{\partial \mathbf{v}}{\partial t} + (\mathbf{v} \cdot \boldsymbol{\nabla}) \mathbf{v} + \frac{\boldsymbol{\nabla} p}{s} = 0 \,, \tag{2.28}$$

where the transversality condition implies that the pressure can be written in terms of the velocity as

$$\frac{p}{s} = -\frac{1}{\boldsymbol{\nabla}^2} \partial_i \partial_j (v_i v_j) \,. \tag{2.29}$$

In two spatial dimensions, the constraint on the velocity can be solved as

$$v_i = \frac{\varepsilon_{ij} \partial_j}{-\boldsymbol{\nabla}^2} \Omega + \bar{v}_i \,, \qquad \Omega = \boldsymbol{\nabla} \wedge \mathbf{v} \,, \tag{2.30}$$

where $\Omega$ is the non-relativistic vorticity and $\bar{v}_i = \text{const.}$[13] By acting with $\boldsymbol{\nabla}\wedge$ on eq. (2.28) the equation of motion are also shown to coincide with the convective conservation of vorticity

$$\begin{aligned} \frac{D\Omega}{Dt} &\equiv \frac{\partial \Omega}{\partial t} + \mathbf{v} \cdot \boldsymbol{\nabla}\Omega \\ &= \frac{\partial \Omega}{\partial t} + \bar{\mathbf{v}} \cdot \boldsymbol{\nabla}\Omega + \boldsymbol{\nabla}\Omega \wedge \frac{1}{-\boldsymbol{\nabla}^2}\boldsymbol{\nabla}\Omega = 0 \,, \end{aligned} \tag{2.31}$$

---

[12]Note that the subleading $O(c^0)$ order in the large $c$ expansion of eq. (2.3) for the $\nu = 0$ component of the stress tensor apparently gives an additional equation,

$$\partial_t \left( \rho_m \frac{\mathbf{v}^2}{2} + \epsilon \right) + \boldsymbol{\nabla} \cdot \left[ \mathbf{v} \left( \frac{1}{2}\rho_m \mathbf{v}^2 + \epsilon \right) \right] = -\boldsymbol{\nabla} \cdot (\mathbf{v}p) \,. \tag{2.24}$$

However, using eqs. (2.25) and (2.26), this is automatically solved by $p(s) = s\epsilon'(s) - \epsilon(s)$.

[13]In more general spaces $\bar{v}$ need only be a closed 1-form. Incompressibility further implies $\nabla \cdot \bar{v} = 0$, and hence $\bar{v}$ is harmonic. On topologically trivial spaces like $\mathbb{R}^2$ it is therefore a constant.

(where in the second line we used eq. (2.30)). In general spatial dimensions, Euler's equations state that the time derivative of the vorticity equals minus its Lie derivative with respect to the velocity. This condition can be written purely in terms of the vorticity tensor using

$$v^i = \frac{1}{\mathbf{\nabla}^2} \partial_j \Omega_{ji}, \qquad \Omega_{ij} = \partial_i v_j - \partial_j v_i, \tag{2.32}$$

along with the Bianchi identity constraint $\partial_i \Omega_{jk} \varepsilon^{ijk\cdots} = 0$.

We conclude this review by providing the expressions of the conserved Casimirs mentioned in the previous subsection in the incompressible limit. In two spatial dimensions the Casimirs (2.15) reduce to

$$C^{(k)} = \int d^2\mathbf{x} \left[\Omega(t, \mathbf{x})\right]^k, \tag{2.33}$$

while in three spatial dimensions the helicity becomes

$$C = \int d^3\mathbf{x} \, \mathbf{v} \cdot (\mathbf{\nabla} \wedge \mathbf{v}). \tag{2.34}$$

# 3 Fluid in the comoving formulation

## 3.1 The action

In this sec. we review the comoving coordinate formulation of the perfect fluid, following [22].

Let us consider a space-filling medium, such that at each space point there resides just one (and only one) *fluid element*. To describe the fluid we parametrize its fluid elements in space by their initial position (at, say, $t = 0$), that we denote with $d$ coordinates $\varphi^I$ with $I = 1, 2, \ldots, d$. The $\varphi^I$ are the comoving coordinates. In what is normally called the Lagrangian perspective, one considers the trajectory $\mathbf{x}(t, \varphi^I)$ of each fluid element as a function of time. Under our assumptions one can also invert this map and treat the comoving coordinates as fields in the physical space: $\varphi^I = \varphi^I(t, \mathbf{x})$. This offers instead the Eulerian perspective, where the flow is described in terms of variables defined at each point $(t, \mathbf{x})$ in space-time [14] As an example, the choice $\varphi^I(t, \mathbf{x}) = \alpha x^I$ describes a fluid at rest.

The basic idea is then to write a Lorentz invariant action for the comoving coordinates $\varphi^I$, now treated as dynamical fields. To specify whether the action we write describes a fluid, a solid, or some other medium, we need to understand the corresponding symmetries. As explained in [22], in this formulation, a fluid is defined as a system that is invariant under the *reshuffling* of the internal elements. In the continuous limit this symmetry consists of volume-preserving reparametrizations $SDiff(\mathcal{M}_\varphi)$ of the comoving coordinates:

$$\varphi^I \to f^I(\varphi), \quad \det\left(\frac{\partial f^I}{\partial \varphi^J}\right) = 1. \tag{3.1}$$

---

[14]Notice however, just to make the nomenclature cumbersome, that the *Eulerian* perspective will allow us to write down an ordinary *Lagrangian* field theory for the fluid.

These transformations are just deformations of the fluid that do not involve compression or dilatation.

It is then straightforward to write the most general Lagrangian invariant under the symmetry (3.1). We introduce the following invariant building block:

$$v^\mu = \frac{1}{d!}\varepsilon_{IJ\ldots}\varepsilon^{\mu\nu\rho\cdots}\partial_\nu\varphi^I\partial_\rho\varphi^J\ldots . \tag{3.2}$$

An elegant way to prove the invariance of $v^\mu$ is to write it as a volume form in the comoving space, as $\star v_\mu dx^\mu = \frac{1}{d!}\varepsilon_{IJ\ldots}d\varphi^I d\varphi^J \ldots$. The most general relativistic Lagrangian invariant under the symmetry at leading order in derivatives is given by

$$\mathcal{L}_1 = F(v^\mu v_\mu) . \tag{3.3}$$

In app. B.1 we derive the action (3.3) (in the non-relativistic limit) as the limit $N \to \infty$ of a system of $N$ interacting point-particles.

To identify the fluid variables, we note first that

$$v^\mu \partial_\mu \varphi^I = 0 \tag{3.4}$$

meaning that the comoving coordinates do not change along the vector field $v^\mu$. This gives

$$u^\mu = \frac{1}{\sqrt{-v^2}} v^\mu \tag{3.5}$$

the natural interpretation of relativistic velocity of the fluid. The stress tensor then takes the form (2.1) with the pressure and the density given by

$$p = F(v^2) - 2F'(v^2)v^2, \quad \rho = -F \tag{3.6}$$

where the prime denotes differentiation with respect to the argument. From this we also see that we can make the identifications $s = \sqrt{-v^2}$ and $T = 2F'\sqrt{-v^2}$. Therefore, the current $v^\mu$, which is topologically conserved

$$\partial_\mu v^\mu = 0 , \tag{3.7}$$

is identified with the entropy current $S^\mu$.

Let us now prove that that the equations of motion that follow from the action (3.3) coincide with Euler's equations. Defining, according to the above identification of $T$ and $u^\mu$, the vorticity as

$$\omega_{\mu\nu} = \partial_\mu \xi_\nu - \partial_\nu \xi_\mu , \qquad \xi_\mu = 2F'(v^2)v_\mu = \frac{\partial \mathcal{L}_1}{\partial v^\mu} . \tag{3.8}$$

the equations of motion are compactly expressed in form notation as:

$$\omega \wedge d\varphi^{I_1} \wedge d\varphi^{I_2} \wedge \ldots \wedge d\varphi^{I_{d-1}} \varepsilon_{JI_1\ldots I_{d-1}} = 0 \quad J = 1, \ldots, d . \tag{3.9}$$

which implies

$$u^\alpha \omega_{\alpha\mu} = 0 . \tag{3.10}$$

Eqs. (3.7) and (3.10) obviously coincide with eqs. (2.6) and (2.10), thus proving that the dynamics of density and velocity is equivalent to that specified by Euler's equations.

Some comments are in order. First, as remarked above, the static flow corresponds to the solution $\varphi^I(t, \mathbf{x}) = \alpha x^I$, where $\alpha = s^{\frac{1}{d}}$ is an arbitrary constant. This solution spontaneously breaks both the internal $SDiff(\mathcal{M}_\varphi)$ (which is indeed non linearly realized by construction) and the Poincaré group to a subgroup consisting of effective time and spatial translations, and rotations, generated by *diagonal* combinations of the internal and spacetime symmetry generators [26]. The close analogy between this system and the ordinary rigid body is illustrated in Appendix A. An important difference between the two systems is that the vacuum solution for the fluid $\varphi^I(t, \mathbf{x}) = \alpha x^I$, is not uniquely fixed by the pattern of symmetry breaking: here we have a family of solutions labelled by $\alpha$, a parameter measuring the entropy density or, equivalently, compression.

Now, expanding around the rest configuration as $\varphi^I(t, \mathbf{x}) = \alpha x^I + \pi^I$, we obtain the quadratic action

$$\mathcal{L}^{(2)} = 2\alpha^{2(d-1)} F'(-\alpha^2) \left[ \frac{1}{2} \dot{\vec{\pi}} - \frac{c_s^2}{2} (\boldsymbol{\nabla} \cdot \vec{\pi})^2 \right] , \tag{3.11}$$

which clearly shows that the transverse modes have no gradient energy. As emphasized in [1], the absence of a gradient term is directly connected to the nonlinear realization of the $SDiff(\mathcal{M}_\varphi)$ symmetry. That is because under the linearized $SDiff(\mathcal{M}_\varphi)$ action, $\vec{\pi}$ transforms according to $\vec{\pi}(t, \mathbf{x}) \to \vec{\pi}(t, \mathbf{x}) + \vec{f}_\perp(\mathbf{x})$ with $\boldsymbol{\nabla} \cdot \vec{f}_\perp(\mathbf{x}) = 0$: this forbids a gradient term for the transverse mode $\vec{\pi}_\perp$.

It is also straightforward to consider higher derivative modifications of the action (3.3). Invariance under the $SDiff(\mathcal{M}_\varphi)$ transformations implies that the action in general is a function of $v^\mu$ and derivatives $\partial_\nu$. Therefore, we find that the equations of motion read as in (3.9), but with the definition of $\xi_\mu$ in eq. (3.8) generalized to

$$\xi_\mu = \frac{\delta S}{\delta v^\mu} . \tag{3.12}$$

This modification implies that at higher order in derivatives eq. (2.10) takes the form

$$S^\mu \omega_{\mu\nu} = 0 . \tag{3.13}$$

The difference with respect to eq. (3.10) arises because the entropy current $S^\mu$ and $\xi_\mu$ are, in general, no longer parallel once higher derivative corrections are taken into account. Indeed, in this formulation $S^\mu$ is topologically conserved and thus its functional form in terms of the fields $\varphi^I$ is not modified by higher derivatives. Interestingly, eq. (3.13) implies that Helmholtz theorem (2.11) is modified to

$$(\mathcal{L}_{\bar{u}} \omega)_{\mu\nu} = 0 , \qquad \bar{u}^\mu = S^\mu / \sqrt{-S^2} . \tag{3.14}$$

In the non-relativistic limit, this expression implies that the circulation is conserved along the displaced flow defined by $\bar{u}^\mu = S^\mu / \sqrt{-S^2}$. Notice that when including higher derivatives, the energy momentum tensor also depends on derivatives of $\bar{u}^\mu$.

Finally, we briefly comment on the incompressible limit of the action (3.3). To this aim, it is simplest to use the Lagrangian perspective, invert the map $\varphi^I = \varphi^I(t, \mathbf{x})$ and treat $\mathbf{x}(t, \varphi^I)$ as the dynamical variables. The action now reads [22]

$$S = \int dt d^d\varphi \left| \frac{\partial \mathbf{x}}{\partial \varphi} \right| F\left( \left| \frac{\partial \mathbf{x}}{\partial \varphi} \right|^{-2} (1 - \dot{\mathbf{x}}^2) \right). \tag{3.15}$$

In [1] it was argued that nonrelativistic incompressible fluids are described by configurations $\varphi^I(\mathbf{x}, t)$ that at fixed time are area-preserving diffs of $\mathbf{x}$. In this regime the sound mode may be consistently integrated out to give the action

$$S_{inc.} = \int dt d^d\varphi \, \frac{\rho_m}{2} \dot{\mathbf{x}}^2, \tag{3.16}$$

where $\rho_m = ms$ is the constant mass density, and the fields $\mathbf{x} = \mathbf{x}(t, \varphi^I)$ are constrained to satisfy

$$\left| \frac{\partial \mathbf{x}}{\partial \varphi} \right| = 1. \tag{3.17}$$

It is this constraint that makes the dynamics nontrivial despite the simplicity of the Lagrangian. We refer the reader to [4] for a detailed analysis of the action (3.16).

## 3.2 Tensor symmetry and Kelvin's circulation theorem

As reviewed in sec. 2, the perfect fluid enjoys a number of nontrivial conservation laws, in particular Kelvin's circulation theorem (2.12). One of the advantages of the comoving Lagrangian formulation is that such conservation laws admit a natural explanation in terms of the $SDiff(\mathcal{M}_\varphi)$ invariance (3.1) [22]. In this section we briefly review this connection in the modern language of tensor global symmetries introduced in [27]. We also provide the expressions of the associated conserved currents in two dimensions for future purposes. This section is not relevant for the analysis of the quantum fluid that will follow in part II, and may be skipped at a first reading.

Let us first review the concept of tensor global symmmetry. Consider in $d$ spatial dimensions a theory invariant under translations and spatial rotations, but without boost invariance. Assume there exists an antisymmetric $k$-tensor current $\{J^{0i_1...i_{k-1}} = J^{0[i_1...i_{k-1}]}, J^{i_1...i_k} = J^{[i_1...i_k]}\}$ (where $i_1, \ldots, i_k$ denote spatial indices in the range $1, \ldots, d$) satisfying

$$\partial_0 J^{0i_1...i_{k-1}} + \partial_j J^{ji_1...i_{k-1}} = 0, \tag{3.18}$$

but, in general and crucially, such that

$$\partial_j J^{0ji_1...i_{k-2}} \neq 0. \tag{3.19}$$

This last condition distinguishes the tensor current $J$ from a relativistic $k-1$-form symmetry current [37]. It follows that for every codimension-$k-1$ closed surface $\mathcal{C}$ there exist a conserved

charge given by[15]

$$Q_{\mathcal{C}} = \int_{\mathcal{C}} J^{0i_1\ldots i_{k-1}} n_{i_1}^{(1)} \ldots n_{i_{k-1}}^{(k-1)} , \tag{3.20}$$

where the $n^{(j)}$'s form a basis of normal vectors on $\mathcal{C}$. Note that in the relativistic case the condition $\partial_j J^{0ji_1\ldots i_{k-2}} = 0$ implies that $Q_{\mathcal{C}}$ is independent of the surface $\mathcal{C}$. When $k = d$ the conserved charges (3.20) are simply contour integrals of the form

$$Q_{\mathcal{C}} = \oint_{\mathcal{C}} dx^i J_{V,i}^0 , \qquad J_{V,j}^0 = \frac{1}{(d-1)!} J^{0i_1\ldots i_{d-1}} \varepsilon_{ji_1\ldots i_{d-1}} . \tag{3.21}$$

We now show that Kelvin's circulation theorem can be understood as a consequence of a tensor symmetry current of the fluid in the comoving formulation. To this aim, note that the curve $\mathcal{C}(\tau)$ in eq. (2.12) (displaced along the field lines of $v^\mu = u^\mu$) corresponds to a time independent curve in $\varphi$ space. It is therefore natural to consider the formulation (3.15). The $SDiff(\mathcal{M}_\varphi)$ symmetry acts as $\mathbf{x}(t,\varphi) \to \mathbf{x}'(t,\varphi) = \mathbf{x}(t,\bar\varphi(\varphi))$, where $|\partial\bar\varphi/\partial\varphi| = 1$. Considering an infinitesimal transformation with spacetime-dependent coefficient

$$\bar\varphi^I = \varphi^I + \alpha(t,\varphi) f^I(\varphi) , \qquad \partial_I f^I = 0 , \tag{3.22}$$

and picking the therms proportional to $\partial\alpha$ in the variation of the action, we obtain the associated Noether's currents

$$j^0 = \frac{\delta L}{\delta \dot x^k} \partial_I x^k f^I(\varphi) , \qquad j^I = \frac{\delta L}{\delta \left|\frac{\partial \mathbf{x}}{\partial \varphi}\right|} f^I(\varphi) , \tag{3.23}$$

where $L$ is the Lagrangian. Choosing $f^I = \varepsilon^{IJ\cdots}$, we see that the current (3.23) reduces to a conserved antisymmetric tensor as in the discussion above

$$J^{0J\cdots} = \frac{\delta L}{\delta \dot x^k} \partial_I x^k \varepsilon^{IJ\cdots} , \qquad J^{IJ\cdots} = \frac{\delta L}{\delta \left|\frac{\partial \mathbf{x}}{\partial \varphi}\right|} \varepsilon^{IJ\cdots} \qquad \Longrightarrow \qquad J_{V,I}^0 = \frac{\delta L}{\delta \dot x^k} \partial_I x^k . \tag{3.24}$$

It is then simple to verify that the associated conserved charges (3.21) coincide with the integrals (2.12).[16]

---

[15]The condition (3.20) is equivalent to the conservation at any point of $G^{i_1\ldots i_{k-2}} = \partial_j J^{0ji_1\ldots i_{k-2}}$.

[16]We have

$$\begin{aligned} Q_{\mathcal{C}} &= \oint_{\mathcal{C}} d\varphi^I J_{V,I}^0 = \oint_{\mathcal{C}(t)} dx^j \frac{\partial \varphi^I}{\partial x^j} \frac{\delta L}{\delta \dot x^k} \frac{\partial x^k}{\partial \varphi^I} = \\ &= -\oint_{\mathcal{C}(t)} dx^j 2F' \left|\frac{\partial \mathbf{x}}{\partial \varphi}\right|^{-1} \dot x^j = \oint_{\mathcal{C}(t)} dx^i 2F' \left|\frac{\partial \mathbf{x}}{\partial \varphi}\right|^{-1} \partial_I x^i \dot\varphi^I \\ &= \frac{1}{(d-1)!} \oint_{\mathcal{C}(t)} dx^i 2F' \varepsilon_{IJ_1\ldots J_{d-1}} \varepsilon^{ij_1\ldots j_{d-1}} \partial_{j_1} \varphi^{J_1} \ldots \partial_{j_{d-1}} \varphi^{J_{d-1}} \dot\varphi^I \\ &= \oint_{\mathcal{C}(t)} dx^i 2F' v_i = \oint_{\mathcal{C}(t)} dx^i T u_i , \end{aligned} \tag{3.25}$$

where we used the explicit form of the current, $\dot x^j + \partial_I x^i \dot\varphi^I = 0$, and expanded the determinant $\left|\frac{\partial \mathbf{x}}{\partial \varphi}\right|^{-1} = \left|\frac{\partial \varphi}{\partial \mathbf{x}}\right|$. The conservation of eq. (3.25) coincides with (2.12) for $v^\mu = u^\mu$.

We stress that the conservation of the Noether currents (3.23) with local Lie parameter $f^I(\varphi)$ not only implies the conservation of the tensor current (3.24), but is completely equivalent to it.[17] Therefore the contour integrals (2.12) provide a complete basis for the conserved charges associated with the $SDiff(\mathcal{M}_\varphi)$ symmetry of the comoving fields.

Finally we also comment that it may be shown that the conserved integrals (2.15) for the two dimensional fluid coincide with the conserved Casimirs of the $SDiff(\mathcal{M}_\varphi)$ symmetry - see e.g. [4].

## 4  Fluid in the Clebsch formulation

### 4.1  From comoving coordinates to Clebsch fields

In [22] the authors showed that all solutions of the EOMs deriving from the action (3.3) with zero vorticity can be mapped to the solutions of a superfluid action. Below we provide a generalization of that argument to solutions with non-zero vorticity, and use it to derive the Clebsch formulation of the fluid. We will focus on two and three spatial dimensions, for which the action is formulated in terms of three real fields.[18]

To this aim, we rewrite the action (3.3) as

$$\mathcal{L} = P(\xi^2) - \xi_\mu v^\mu \,, \tag{4.1}$$

where $\xi^\mu$ is here an independent vector field and $v_\mu$ is given in eq. (3.2). Integrating out $\xi_\mu$, we recover the action (3.3) and recognize $P$ as the Legendre transform of $F$; i.e. $P$ is the thermodynamic pressure. Therefore, $\xi_\mu = T u_\mu$ on a solution, and its value completely specifies the fluid flow.

In what follows we first write the general form of $\xi^\mu$ that is determined by the equations of motion of $\varphi$ from eq. (4.1). It is here that the Clebsch variables make their first appearance. Secondly we show that the complete system of equations of motion can be derived using a Lagrangian that is purely written in terms of the Clebsch variables. While the end-result is the same, the argument leading to it is slightly different in two and three spatial dimensions. We therefore discuss these two cases separately, starting from the former.

In two spatial dimensions, the equations of motion for $\varphi$ that follow from eq. (4.1) are

$$\varepsilon^{\mu\nu\rho}\partial_\mu\xi_\nu\partial_\rho\varphi^I = 0\,, \quad I = 1,2\,. \tag{4.2}$$

A theorem by Pfaff [38] states that the most-general three-vector $\xi^\mu$ can be parametrized (locally) in terms of three fields $\bar\chi$, $\bar\Phi$ and $\bar\Phi^\dagger$ in the so-called Clebsch form:

$$\xi^\mu = \partial_\mu\bar\chi + \frac{i}{2}(\bar\Phi^\dagger\partial_\mu\bar\Phi - \bar\Phi\partial_\mu\bar\Phi^\dagger)\,. \tag{4.3}$$

---

[17]To verify that the conservation of the Noether current follows from that of the tensor, we write the Noether currents via $j^0 = \frac{1}{(d-1)!}J^{0IJ\cdots}\varepsilon_{\bar I J\ldots}f^{\bar I}(\varphi)$ and $j^{\bar I} = \frac{1}{d!}J^{IJ\cdots}\varepsilon_{IJ\ldots}f^{\bar I}(\varphi)$, and then use $\partial_I f^I = 0$.

[18]In $d = 2p$ and $2p+1$ dimensions, one generally needs $2p+1$ fields.

Using this parametrization, eqs. (4.2) reads

$$\varepsilon^{\mu\nu\rho}\partial_\mu\bar\Phi\partial_\nu\bar\Phi^\dagger\partial_\rho\varphi^I = 0\,, \quad I = 1,2\,, \tag{4.4}$$

whose general solution implies[19]

$$\bar\Phi^\dagger\partial_\mu\bar\Phi = \partial_\mu\psi + g_I(\varphi)\partial_\mu\varphi^I\,, \tag{4.6}$$

where $g_I(\varphi)$ is a generic function of $\varphi^1$ and $\varphi^2$, $I = 1,2$ and $\partial_\mu\psi$ is an arbitrary gradient term. Redefining $\chi = \bar\chi + \psi$, we thus obtain that the $\xi_\mu$ that solves eq. (4.2) may always be written as

$$\xi_\mu = \partial_\mu\chi + \frac{i}{2}(\Phi^\dagger\partial_\mu\Phi - \Phi\partial_\mu\Phi^\dagger)\,, \quad \text{where } \Phi = \Phi(\varphi)\,. \tag{4.7}$$

We now use that the equations of motion for $\xi_\mu$ imply that

$$2P'\xi^\mu = v^\mu\,, \tag{4.8}$$

from which it follows

$$\partial_\mu(2P'\xi^\mu) = 0\,, \qquad \xi^\mu\partial_\mu\varphi^I = 0\,. \tag{4.9}$$

For all physical flows, the $\varphi^I$'s define a non-degenerate volume form $v$ and $\xi$ is a non-vanishing vector. As $\Phi$ is just a (complex) function of the $\varphi^I$'s, the second of eqs. (4.9) is equivalent to:

$$\xi^\mu\partial_\mu\Phi = \xi^\mu\partial_\mu\Phi^\dagger = 0\,. \tag{4.10}$$

We therefore conclude that the most general solution to the EOMs for $\xi_\mu$ can be written as

$$\xi_\mu = \partial_\mu\chi + \frac{i}{2}(\Phi^\dagger\partial_\mu\Phi - \Phi\partial_\mu\Phi^\dagger)\,, \tag{4.11}$$

satisfying the constraints

$$\partial_\mu(2P'\xi^\mu) = \xi^\mu\partial_\mu\Phi = \xi^\mu\partial_\mu\Phi^\dagger = 0\,. \tag{4.12}$$

Note that (4.9) implies (4.10), but the converse is not true when $\Phi(\mathbf{x})$ is not an invertible map. Thus eq. (4.12) contains slightly less information than eq. (4.9). However, if one only considers the fluid flow (that is $\xi^\mu$), the two sets of equations imply the same set of solutions.

---

[19]It is convenient to work in comoving coordinates $(t,\varphi^1,\varphi^2)$, in which eq. (4.4) reads

$$\det\left(\frac{\partial(\mathrm{Re}\bar\Phi,\mathrm{Im}\bar\Phi)}{\partial(t,\varphi^I)}\right) = 0\,, \quad I = 1,2\,. \tag{4.5}$$

Eqs. (4.5) imply that the map $(t,\varphi^I) \to (\mathrm{Re}\bar\Phi,\mathrm{Im}\bar\Phi)$ is non-invertible, for both $I = 1,2$. This condition can be solved in two ways. First, eq. (4.5) is obviously satisfied when $\bar\Phi$ depends on the $\varphi^I$ but not on time; in that case we have $\bar\Phi^\dagger\partial_\mu\bar\Phi \equiv g_I(\varphi)\partial_\mu\varphi^I$. Alternatively, when $\bar\Phi$ depends on time, the noninvertibility conditions imply $\mathrm{Re}\bar\Phi = \mathrm{Re}\bar\Phi\left(f(t,\varphi^1,\varphi^2)\right)$ and $\mathrm{Im}\bar\Phi = \mathrm{Im}\bar\Phi\left(f(t,\varphi^1,\varphi^2)\right)$ for some function $f$; in this second case $\bar\Phi^\dagger\partial_\mu\bar\Phi \equiv \partial_\mu\psi$ is a total derivative. Eq. (4.6) compactly encompasses both cases.

It is finally straightforward to realize that eqs. (4.11) and (4.12) coincide with the EOMs associated with the action:

$$\mathcal{L}_\xi = P(\xi^\mu \xi_\mu), \qquad \xi^\mu = \partial_\mu \chi + \frac{i}{2}(\Phi^\dagger \partial_\mu \Phi - \Phi \partial_\mu \Phi^\dagger), \qquad (4.13)$$

with $\chi$, $\Phi$ and $\Phi^\dagger$ taken as independent fields. Eq. (4.13) is the fluid's action in the Clebsch formulation at leading order in derivatives.

Let us now discuss three spatial dimensions. The most general four-vector can be parametrized (locally) in terms of four scalar fields rather than three as in eq. (4.3) [38]:

$$\xi_\mu = \alpha \partial_\mu \beta + \gamma \partial_\mu \delta. \qquad (4.14)$$

However, as it turns out, it is enough to work with the parametrization (4.7) in terms of three fields. The reason for that is tied to the conservation of the fluid helicity (2.17) and can be seen by considering the equations of motion for $\varphi$, which in three dimensions read:

$$\varepsilon^{\mu\nu\rho\sigma} \partial_\mu \xi_\nu \partial_\rho \varphi^I \partial_\sigma \varphi^J \varepsilon_{IJK} = 0, \quad K = 1, 2, 3. \qquad (4.15)$$

As $\partial_\rho \varphi^I$ is a rank 3 matrix, this equation implies that $\omega_{\mu\nu} = \partial_\mu \xi_\nu - \partial_\nu \xi_\mu$ cannot have maximal rank.[20] In form notation that can be expressed as $\omega \wedge \omega = 0$, which is precisely helicity conservation as expressed in eqs. (2.16), (2.17), and which further implies the fields in the parametrization (4.14) must satisfy

$$d\alpha \wedge d\beta \wedge d\gamma \wedge d\delta = 0. \qquad (4.16)$$

This relation implies that locally there exists a coordinate system $(\tau, \tilde{\varphi}^I)$ such the fields of the parametrization (4.14) depend only on the $\tilde{\varphi}^I$. In this coordinate system $\xi_\mu$ consists of a vanishing $\tau$-component and a three-vector, and thus we may resort to the former Clebsch parametrization (4.3). The rest of the argument proceeds as in the two dimensional case, and we find again that the action (4.13) describes all possible solutions for the density and for the velocity of the perfect fluid.

An action similar to (4.13) was first proposed in [39], and is frequently used in the literature, see e.g. [25, 31, 32]. In the next subsection we will analyze in some detail its properties and symmetries. Before doing that we would like to make some remarks on this derivation.

First, the parametrization (4.3) admits an intuitive interpretation as a minimal modification of the superfluid theory. In a superfluid, all flows have trivial vorticity and may therefore be parametrized by a shift invariant scalar $Tu_\mu = \partial_\mu \chi$ [40], where $\chi$ is thus interpreted as the Goldstone boson of a spontaneously broken $U(1)$ symmetry. The parametrization (4.3) therefore provides the minimal modification of the theory of a superfluid that can account for a nontrivial vorticity.[21]

---

[20]Like in the $d = 2$ case the implications of this equation are most directly analyzed in comoving coordinates $(t, \varphi^1, \varphi^2)$, where the constraint simply reads $\omega_{0I} = 0$ for $I = 1, 2, 3$.

[21]Similar actions are used in the EFT of a non-relativistic rotating superfluid in a vortex lattice state [41–43]. In the vortex lattice however one does not impose the area preserving diffeormophism symmetry (4.20) that we will soon discuss, and thus the action includes additional invariants besides $\xi_\mu$, such as $|\partial_i \Phi|^2$.

The above derivation can be generalized to include higher derivative terms in the action (3.3), as long as such terms are treated perturbatively in the spirit of effective field theory.[22] One finds that the resulting Clebsch action (4.13) is written as a function of $\xi_\mu$ and its derivatives:

$$S_{gen} = \int d^3x \, P_{gen}\left[\partial_\nu, \xi_\mu\right] . \tag{4.17}$$

The EOMs that follow from (4.17) read

$$\partial_\mu S^\mu = S^\mu \partial_\mu \Phi = S^\mu \partial_\mu \Phi^\dagger = 0 \,, \qquad S^\mu = \frac{\delta S_{gen}}{\delta \xi_\mu} \,, \tag{4.18}$$

where we identified the conserved current with the entropy current. Note that in this formulation $S^\mu$ is a conserved current associated with the $U(1)$ shift symmetry of $\chi$, and in general is a function of $\xi^\mu$ and derivatives thereof. The other eqs. in (4.18) imply that at higher order in derivatives eq. (2.10) gets generalized to

$$\omega_{\mu\nu}S^\nu = 0 \,, \qquad \omega_{\mu\nu} = \partial_\mu \xi_\nu - \partial_\nu \xi_\mu = i(\partial_\mu \Phi^\dagger \partial_\nu \Phi - \partial_\nu \Phi^\dagger \partial_\mu \Phi). \tag{4.19}$$

This last result formally coincides with (3.13) obtained in the previous section. The difference is that in the Clebsch formulation the expression of the entropy current in terms of the fields depends on the action, while $\xi_\mu$ and thus the vorticity $\omega$ retain the same form in terms of the Clebsch fields to all orders in the derivative expansion.

Finally, we stress that we have only proven that the EOMs for the hydrodynamic density and velocities that derive from the actions (4.13) and (3.3) are the same, but the dynamics of the two systems are different as a whole. That is obvious in three spatial dimensions, since the comoving system describes three canonical pairs, while the Clebsch Lagrangian only has two since $\Phi$ and $\Phi^\dagger$ are canonically conjugated as we shall see below.[23] Even if less obvious, the two systems are inequivalent also in two spatial dimensions where both the comoving and the Clebsch formulations describe two canonical pairs. The reason for their inequivalence is that, as we shall see, the map between the fields in the two formulations is generically noninvertible even modulo the action of symmetry. That is to be contrasted with known examples of classical dualities, such as that between a scalar and a gauge field in two dimensions, which we shall review in sec. 4.3. In that example, the scalar profile completely

---

[22]In this discussion we neglect potential Wess-Zumino terms, that may not be written in terms of $v_\mu$ and derivatives in $d+1$ dimensions, as the one considered in [44, 45] for baroclinic fluids (fluids with a nontrivial conserved current besides the entropy current).

[23]This is not a contradiction, since only the dynamics for the fluid's density and velocity is equivalent, which require $d+1$ initial conditions in $d$ spatial dimensions. Note that from this perspective, in three dimensions the Clebsch parametrization realizes Euler's equations with the minimal number of fields, while in two (and in all even) dimensions the Clebsch parametrization introduces one too many variables. This variable must get "neutralized", i.e. the structure of the action should be such that only three initial conditions are needed to solve Euler's equations. In this light, the infinite number of conservation laws (2.15) (written purely in terms of the hydrodynamic variables) is perhaps to be expected—see app. A for further details on the role of Casimirs in going from the Hamiltonian to the Lagrangian formulation.

specifies the dual gauge field and vice versa up to the action of the global and of the gauge symmetry, and thus allows mapping all observables on the two sides. In our derivation here, we merely proved that hydrodynamic variables can be expressed equivalently in the Clebsch and the comoving description, but the full dynamics of the two systems in principle includes additional observables.

There is however an exception to the above caveat: flows in which the vorticity volume form $\omega = id\Phi \wedge d\Phi^\dagger$ is everywhere non-degenerate, i.e. fluids with macroscopic vorticity. In that case the relation between the Clebsch fields and the comoving ones in invertible up to the action of symmetries, and we can build a precise map between all observables, not just the hydrodynamic quantities. We discuss further this mathematical curiosity in app. C.1. In this work we will be mostly concerned with small fluctuations around the static flow, for which the vorticity is vanishing. We therefore do not discuss this point further in the main text. We however remark that the fact that for near trivial classical flows, i.e. flows that are intuitively near the ground state, the two descriptions are not in one-to-one correspondence, already hints that in the quantum case the vacua will have radically different properties.

## 4.2 General analysis

Let us now analyze some properties of the system described by (4.13). First, we note that the vector $\xi^\mu$ is inherently invariant under the following infinitesimal transformation

$$
\begin{aligned}
\Phi &\to \Phi + \partial_{\Phi^\dagger} f(\Phi^\dagger, \Phi) \,, \\
\Phi^\dagger &\to \Phi^\dagger - \partial_\Phi f(\Phi^\dagger, \Phi) \,, \\
\chi &\to \chi + \frac{i}{2} \left( 2f - \Phi \partial_\Phi f - \Phi^\dagger \partial_{\Phi^\dagger} f \right) \,.
\end{aligned}
\tag{4.20}
$$

where $f^\dagger = -f$, such that we preserve the identity $(\Phi^\dagger)^\dagger = \Phi$. For $f = \text{const.}$, this symmetry reduces to a $U(1)$ shift of $\chi$. The remaining transformations form a representation of the area preserving diffeomorphism of the $(\Phi, \Phi^\dagger)$ manifold, which we will call $\mathcal{M}_\Phi$. As the Lagrangian (4.17) is a function of $\xi^\mu$, this is trivially a symmetry of the action (4.17) to all order in derivatives. Since hydrodynamic variables are invariant under (4.20), it is this symmetry that ensures that the equations of the fluid can be expressed solely in terms of thermodynamic objects.

The static configuration $\xi_\mu \propto \delta_\mu^0$ admits a large classical degeneracy in terms of the Clebsch fields. This degeneracy can be parametrized in terms of an arbitrary function $\alpha = \alpha(\mathbf{x})$ of the spatial coordinates and an arbitrary function of a single variable $\beta = \beta(\alpha)$ as

$$
\begin{cases}
\chi = Tt + \dfrac{1}{2}\alpha(\mathbf{x})\beta(\alpha(\mathbf{x})) - B(\alpha(\mathbf{x})) \,, \\
\operatorname{Re}\Phi = \dfrac{1}{\sqrt{2}}\alpha(\mathbf{x})\,, \quad \operatorname{Im}\Phi = -\dfrac{1}{\sqrt{2}}\beta(\alpha(\mathbf{x}))\,,
\end{cases}
\qquad \text{where} \quad B(\alpha) = \int d\alpha\, \alpha\, \beta'(\alpha)\,.
\tag{4.21}
$$

For $\Phi = \alpha = \beta = 0$, eq. (4.21) reduces to $\chi \propto t$ up to a constant, and therefore breaks spontaneously the $U(1)$ shift symmetry and time translations to a diagonal subgroup: this

is the superfluid symmetry breaking pattern [40]. Notice however that the $SDiff(\mathcal{M}_\Phi)$ invariance is only partially broken over the large manifold of classical vacua (4.21). That is rendered clear by considering the point $\Phi = 0$, which is invariant for all $f(\Phi^\dagger, \Phi)$'s that are at least quadratic at the origin, so that $SDiff(\mathcal{M}_\Phi)$ is basically unbroken. That is different from what happens in the comoving description, where the field solutions describing the same static background flow are related by, and in one-to-one correspondence with, the elements of $SDiff(\mathcal{M}_\varphi)$, which is fully non-linearly realized. Note that these considerations imply that on a generic flow it is not possible to reconstruct the value of the Clebsch fields in terms of the comoving ones in two spatial dimensions, even if we mod out the action of the symmetry groups, as anticipated in the previous section.

Expanding around the static fluid background (4.21) we obtain the following quadratic action

$$\mathcal{L}^{(2)} = \frac{1}{2}\left(2P' + 4T^2P''\right)\dot{\pi}^2 - P'(\boldsymbol{\nabla}\pi)^2 + 2iTP'\delta\Phi^\dagger\dot{\delta\Phi}\,, \qquad (4.22)$$

where we discarded a total derivative and $\pi$ is the fluctuation of $\chi$. From eq. (4.22) we also see that $\pi$ describes the phonon mode with sound speed $c_s^2 = dp/d\rho$, while $\Phi^\dagger$ and $\Phi$ describe a single mode with trivial momentum independent dispersion relation $\omega(\mathbf{k}) = 0$. Note again that while the existence of a mode with trivial dispersion relation was expected, the existence of this flat direction is not related to the nonlinear realization of a symmetry. In other words, while invariance of the Lagrangian under the group (4.20) forbids a gradient term for $\Phi$, the time independent solutions with $\omega(\mathbf{k}) = 0$ are not obtained from the action of the symmetry on the static solution, unlike in the comoving formulation.

Let us briefly comment on the physical interpretation of the Clebsch variables in eq. (4.13) and the symmetry (4.20). As alluded above, the fields $\Phi$ and $\Phi^\dagger$ are intuitively related to vorticity. Following well known ideas (see e.g. [35, 46]), we make this intuition precise in app. B.2 for two dimensional fluids. There we consider the motion of $N \gg 1$ pointlike vortices in a superfluid, and interpret $\mathrm{Re}\,\Phi$ and $\mathrm{Im}\,\Phi$ as the continuum limit of the vortex labels, while $\chi$ is the superfluid Goldstone (more precisely, we work in the dual formulation in terms of a gauge field, which we review below). We find that the action (4.13) describes the hydrodynamics of such system in the regime where the vortices are sufficiently dense, and thus there effectively exist an invertible map between the spatial coordinates $\mathbf{x}$ and the vortex labels $\Phi$, $\Phi^\dagger$. The area-preserving diffeomorphism symmetry thus emerges as a consequence of the possibility of reshuffling the individual vortices, without changing the macroscopic properties of the fluid. This symmetry is spontaneously broken in a flow with macroscopic vorticity, and thus the Clebsch formulation of the fluid follows by the requirement that this symmetry is nonlinearly realized.

The above interpretation of the Clebsch $SDiff(\mathcal{M}_\Phi)$ symmetry is therefore analogous to that of the $SDiff(\mathcal{M}_\varphi)$ symmetry of the comoving fields that we discussed in sec. 3.1. The difference is that in the present case such physical interpretation is justified only for flows such that $\omega_{\mu\nu}$ is nowhere vanishing, and breaks down near the static background that we are interested in. We remark again that the Clebsch Lagrangian describes all possible solutions

of Euler's equations, irrespectively of whether such solution corresponds or not to a regime in which the Clebsch fields admit a natural physical interpretation.

## 4.3 The non-relativistic and incompressible limits

An important technical advantage of the Clebsch formulation is that it allows for a simple formulation of the incompressible limit. An elegant and straighforward derivation of the latter uses the well known duality between a scalar and a $d-1$-form gauge field. Below we discuss such derivation in two spatial dimensions. The analogous derivation in three dimensions is discussed in app. C.2 - we will simply report the final result at the end of this section.

The idea is to trade the scalar $\chi$ for an Abelian gauge field $A_\mu$, and thus obtain a dual description of the system (4.13). This is done by treating $\xi_\mu$ as an independent field and adding a Lagrange multiplier $A_\mu$ to impose the dependence of $\xi_\mu$ on the fundamental fields:

$$\tilde{\mathcal{L}}_\xi = P(\xi^\mu \xi_\mu) - \frac{1}{2\pi}\left[\xi_\mu - \frac{i}{2}(\Phi^\dagger \partial_\mu \Phi - \Phi \partial_\mu \Phi^\dagger)\right]\varepsilon^{\mu\nu\lambda}\partial_\nu A_\lambda\,. \tag{4.23}$$

Integrating out $A_\mu$ sets $\xi^\mu - \frac{i}{2}(\Phi^\dagger \partial_\mu \Phi - \Phi \partial_\mu \Phi^\dagger) = \partial_\mu \chi$ and we get back the action (4.13) as intended. Integrating out $\xi^\mu$, and discarding a total derivative, we instead get

$$\mathcal{L}_A = F(v_\mu v^\mu) + \frac{i}{2\pi}A_\mu \varepsilon^{\mu\nu\lambda}\partial_\nu \Phi^\dagger \partial_\lambda \Phi\,, \qquad v^\mu = \frac{1}{4\pi}\varepsilon^{\mu\nu\lambda}F_{\nu\lambda} = \frac{1}{2\pi}\varepsilon^{\mu\nu\lambda}\partial_\nu A_\lambda\,, \tag{4.24}$$

where $F$ is simply the Legendre transform of $P$ with respect to $\xi_\mu$,

$$F = P - \xi \cdot v|_{2P'\xi=v}\,. \tag{4.25}$$

In view of eq. (4.1), $F$ is the same function we started with in the comoving description in eq. (3.3). In particular $F = -\rho$ is minus the energy density of the fluid at rest. Note however that there is a crucial difference between the dual map we just illustrated and the procedure carried out in sec. 4.1. Unlike in that case, the map between $\chi$ and the gauge field is always invertible up to the action of the global and gauge symmetry. A gauge theory formulation of the fluid similar to (4.24) was used in [47] to study coastal Kelvin waves.

As usual in particle-vortex duality, the entropy current $S^\mu = v^\mu$, which was a Noether current in the original Clebsch formulation, is now a topological current. The static background (4.21) now corresponds to a constant magnetic field $F^{ij} = \varepsilon^{ij}B = \text{const.}$, while $SDiff(\mathcal{M}_\Phi)$ defined in eq. (4.20) now acts only on $\Phi$:

$$\Phi \to \Phi + \partial_{\Phi^\dagger}f(\Phi^\dagger, \Phi)\,, \qquad \Phi^\dagger \to \Phi^\dagger - \partial_\Phi f(\Phi^\dagger, \Phi)\,, \qquad A_\mu \to A_\mu\,. \tag{4.26}$$

Consider now the non-relativistic limit of the action (4.24). Recalling the discussion that led to eq. (2.21) and the relations $F = -\rho$ and $\sqrt{-v^2} = s$, this limit corresponds to $F$ of the form

$$\begin{aligned} F(v_\mu v^\mu) &= -mc^2\sqrt{-v^2} + F_{NR}\left(\sqrt{-v^2}\right) + O\left(\frac{1}{c^2}\right) \\ &= -mc^2 v_0 + \frac{mv_i v_i}{2v_0} + F_{NR}(v_0) + O\left(\frac{1}{c^2}\right)\,, \end{aligned} \tag{4.27}$$

where $F_{NR}$ does not depend on $c$, we used $-v^2 = v_0^2 - \frac{1}{c^2}v_i v_i$ (restoring powers of $c$), and we assumed for concreteness $v_0 > 0$ on the background.[24] Writing $v^\mu$ in terms of $E^i$ and $B$ according to eq. (4.24) and dropping total derivatives, the action becomes

$$\mathcal{L}_A = F_{NR}(B) + \frac{m\mathbf{E}^2}{4\pi B} + \frac{i}{2\pi}A_\mu \varepsilon^{\mu\nu\lambda}\partial_\nu \Phi^\dagger \partial_\lambda \Phi\,, \tag{4.28}$$

while the mass density is

$$\rho_m = m\frac{B}{2\pi}\,. \tag{4.29}$$

As explained in sec. 2, in the incompressible limit we retain only the field modes with $\omega \ll c_s|\mathbf{k}|$. These include the longitudinal fields $\Phi$, $\Phi^\dagger$, and the Coulomb potential mediated by the electric field, but not the photon. Therefore, to obtain the Lagrangian in the incompressible limit, it is convenient to work in coulomb gauge $\boldsymbol{\nabla} \cdot \mathbf{A} = 0$ and neglect fluctuations of the spatial components of the gauge field when expanding around solution with constant density $\rho_m = \text{const.}$. After discarding total derivatives, replacing the magnetic field with its background expectation value (4.29), and canonically normalizing the fields $\Phi \to \sqrt{\frac{m}{\rho_m}}\Phi$ and $A_0 \to (2\pi)\frac{\sqrt{\rho_m}}{m}A_0$, we obtain

$$\mathcal{L}_3 = \frac{1}{2}\left(\boldsymbol{\nabla}A_0\right)^2 + \frac{i}{\sqrt{\rho_m}}\left(\boldsymbol{\nabla}\Phi^\dagger \wedge \boldsymbol{\nabla}\Phi\right)A_0 + i\Phi^\dagger\dot{\Phi}\,. \tag{4.30}$$

The Gauss law is then given by

$$\boldsymbol{\nabla} \cdot \mathbf{E}(\mathbf{x}) = \frac{i}{\sqrt{\rho_m}}\boldsymbol{\nabla}\Phi^\dagger(\mathbf{x}) \wedge \boldsymbol{\nabla}\Phi(\mathbf{x}) = \frac{1}{\sqrt{\rho_m}}\omega(\mathbf{x}) \tag{4.31}$$

where in particular we see from the right-hand side that the charge density in the electromagnetic picture can be identified with the vorticity in the fluid picture.

Using the Gauss law, we can further integrate out the Coulomb field. This leads to the following non-local Lagrangian

$$\mathcal{L}_\Phi = -\frac{1}{2\rho_m}\left(\boldsymbol{\nabla}\Phi^\dagger \wedge \boldsymbol{\nabla}\Phi\right)\frac{1}{\boldsymbol{\nabla}^2}\left(\boldsymbol{\nabla}\Phi^\dagger \wedge \boldsymbol{\nabla}\Phi\right) + i\Phi^\dagger\dot{\Phi}\,. \tag{4.32}$$

This Lagrangian, with $\Phi$ and $\Phi^\dagger$ dubbed *vorton fields*, was formerly obtained in [4] in a different way. Our discussion shows that it naturally arises as the non-relativistic and incompressible limit of the most general theory invariant under the internal $SDiff(\mathcal{M}_\Phi)$ of the Clebsch fields.

As a consequence of the large internal symmetry group, the theory (4.32) has several peculiar features. First, as noted already before, there is no quadratic kinetic terms for the vorton fields. Additionally, the leading interaction is non-local and can be interpreted as a dipole-dipole potential. This is reminiscent of the analysis of [49], and suggests that the

---

[24]We refer the reader to [48] for a systematic discussion of the constraints imposed by Galilean symmetry beyond leading order in derivatives.

vorton fields should be associated with the density of small vortex-antivortex pairs, whose leading interaction is indeed dipolar. We will make this intuition sharper in the second part of this work, where we will take the action (4.32) as the starting point for the analysis of the quantum perfect fluid.

We conclude this section with the promised result for the Clebsch Lagrangian in the incompressible limit in three spatial dimensions:

$$\mathcal{L}_\Phi = \frac{1}{2\rho_m}\omega^i \frac{1}{\boldsymbol{\nabla}^2}\omega^i + i\Phi^\dagger\dot{\Phi}\,, \qquad \omega^i = i\varepsilon^{ijk}\partial_j\Phi^\dagger\partial_k\Phi\,, \tag{4.33}$$

that we derive in app. C.2. Comments analogous to the ones above apply in this case. We will use the action (4.33) to analyze the quantum perfect fluid in three dimensions in sec. 7.2.

## 4.4 The hurricane solution and the role of symmetries

As shown in eq. (4.21), the static fluid can be represented by the configuration $\chi = Tt$, $\Phi = \Phi^\dagger = 0$. This is a solution thanks to the $\chi \to \chi + c$ shift symmetry, which, combined with ordinary time translations, guarantees the survival of an effective time translation invariance. In this respect the static fluid corresponds to a superfluid state of the $\chi$-shift $U(1)$, with $T$ playing the role of chemical potential. A similar phenomenon and involving the full $SDiff(\mathcal{M}_\Phi)$ is indeed at play over more general stationary solutions, as we now illustrate.

The distinctive property of Euler's equations is that they admit infinitely many stationary solutions [50]. This is a direct consequence of the infinite symmetry of the Lagrangian formulations, comoving or Clebsch: all stationary solutions break spontaneously both time translation and the internal symmetry, while preserving an effective linear combination $H_{eff} = H + Q$, with $Q$ an internal generator. More explicitly, stationary solutions satisfy

$$D_t\Phi_{stationary} = \partial_t\Phi_{stationary} + \delta_Q\Phi_{stationary} = 0\,. \tag{4.34}$$

The variety of stationary solutions is parametrized by the variety of the possible $Q$'s, while the effective Hamiltonian $H_{eff} = H + Q$ characterizes their spectrum of fluctuations.

Let us illustrate these abstract considerations discussing a concrete example: the hurricane solution for the incompressible fluid in $2d$. In spherical coordinates $(r, \theta)$, this corresponds to velocity and vorticity given by

$$v_r = 0\,, v_\theta = v(r) \quad \Longrightarrow \quad \omega(r) = -\rho_m\frac{v(r) + rv'(r)}{r}\,. \tag{4.35}$$

which is easily seen to solve Euler's equation (2.28) in the incompressible limit. Note that the profile $v(r) \propto 1/r$ describes a localized vortex $\omega \propto \delta^2(\mathbf{x})$. On the other hand $v(r) \propto r$ describes a *rigid* flow with constant angular velocity and constant vorticity.

In the Clebsch description (4.32), this solution is obtained by considering the ansatz

$$\Phi = \sqrt{\rho_m}g(r)e^{i(\theta-\nu(r)t)}\,, \tag{4.36}$$

which yields the vorticity

$$\omega(r) = \frac{-2\rho_m}{r} g(r) g'(r) \,, \tag{4.37}$$

from which we identify

$$g(r) = \sqrt{r v(r)} \,. \tag{4.38}$$

The equations of motions then constrain $\nu(r)$ to coincide with the flow's angular velocity:

$$\nu(r) = \frac{v(r)}{r} \,, \tag{4.39}$$

so that the hurricane solution takes the form

$$\Phi = \sqrt{\rho_m r v(r)} \, e^{i\left(\theta - \frac{v(r)}{r} t\right)} \,. \tag{4.40}$$

It is now simple to check that eq. (4.40) satisfies (4.34) with (recall (4.26))

$$\delta_Q \Phi = i \frac{\partial f\left(\Phi \Phi^\dagger\right)}{\partial \Phi^\dagger} \,, \qquad \delta_Q \Phi^\dagger = -i \frac{\partial f\left(\Phi \Phi^\dagger\right)}{\partial \Phi} \,, \tag{4.41}$$

with $f$ the solution of the differential equation

$$f'\left(\rho_m r v(r)\right) = \frac{v(r)}{r} \,, \tag{4.42}$$

which may be solved for generic $v(r)$. For instance, if $v(r) = \alpha r$ for some constant $\alpha$, we have $f(x) = \alpha x$. However, the form of $f$ varies with the velocity profile, which elucidates the relation between the infinity of the class of stationary solutions and the infinity of the $SDiff(\mathcal{M}_\Phi)$ symmetry. An explicit breakdown of $SDiff(\mathcal{M}_\Phi)$ to a finite dimensional subgroup would necessarily reduce the class of stationary solutions to a finite set. In view of that, when considering the quantum theory, we will be driven to consider a UV regulation that preserves as much symmetry as possible.

To conclude this section, notice that the effective time translation generator $H_{eff} = H + Q$ of eq. (4.34) involves a generator $Q$ of an internal non-abelian symmetry. In this situation the modes associated with the (vast) set of broken generators not commuting with $Q$ feature gaps purely controlled by the symmetry algebra. Upon quantization the associated quanta are an instance of the so-called gapped Goldstone discussed in [51–54].

# Part II

# Quantum perfect fluid

## 5 Continuum quantization of the perfect fluid in 2d

### 5.1 The classical incompressible fluid in the Hamiltonian formalism

The action for the incompressible perfect fluid in the Clebsch formulation was given in (4.32). It reads

$$\mathcal{L}_\Phi = i\Phi^\dagger \dot{\Phi} + \frac{1}{2\rho_m} \left(\boldsymbol{\nabla}\Phi^\dagger \wedge \boldsymbol{\nabla}\Phi\right) \frac{1}{-\boldsymbol{\nabla}^2} \left(\boldsymbol{\nabla}\Phi^\dagger \wedge \boldsymbol{\nabla}\Phi\right) \,. \tag{5.1}$$

As already noted, the key feature of the action (5.1) is the absence of a quadratic kinetic term $|\boldsymbol{\nabla}\Phi|^2$ due to the $SDiff(\mathcal{M}_\Phi)$ symmetry, which acts as

$$
\begin{aligned}
\Phi &\to \Phi + i\partial_{\Phi^\dagger} f(\Phi^\dagger, \Phi)\,, \\
\Phi^\dagger &\to \Phi^\dagger - i\partial_\Phi f(\Phi^\dagger, \Phi)\,,
\end{aligned}
\tag{5.2}
$$

for an arbitrary real function $f^\dagger = f$.

In preparation of the discussion of the quantum fluid, let us briefly review the Hamiltonian description of the system (5.1). The Hamiltonian reads

$$
H = \frac{1}{2\rho_m} \int d^2\mathbf{x}\, \omega(\mathbf{x}) \frac{1}{-\boldsymbol{\nabla}^2} \omega(\mathbf{x}) \quad \text{where} \quad \omega(\mathbf{x}) = i\boldsymbol{\nabla}\Phi^\dagger(\mathbf{x}) \wedge \boldsymbol{\nabla}\Phi(\mathbf{x})\,,
\tag{5.3}
$$

and the Poisson brackets are given by

$$
-i\{\Phi^\dagger(\mathbf{x}), \Phi(\mathbf{y})\} = \delta^2(\mathbf{x} - \mathbf{y})\,.
\tag{5.4}
$$

The symmetry (5.2) is then seen to be generated by the following charges

$$
Q_f = \int d^2\mathbf{x}\, f(\Phi(\mathbf{x}), \Phi^\dagger(\mathbf{x}))\,,
\tag{5.5}
$$

with $f$ a real valued function. For future references, we explicitly write the associated algebra

$$
\{Q_f, Q_g\} = iQ_{[f,g]} \qquad [f,g] = \partial_\Phi f \partial_{\Phi^\dagger} g - \partial_{\Phi^\dagger} f \partial_\Phi g
\tag{5.6}
$$

We can take as a basis of the symmetry algebra (real combinations of) monomials of the form:

$$
Q_{(n,m)} = \int d^2\mathbf{x}\, (\Phi^\dagger(x))^n \Phi^m(x)
\tag{5.7}
$$

In particular, $Q_{(1,1)} = \int d^2\mathbf{x}\, \Phi^\dagger(\mathbf{x})\Phi(\mathbf{x})$ generates the $U(1)$ symmetry corresponding to particle number conservation. The algebra then becomes

$$
-i\left\{Q_{(n,m)}, Q_{(k,\ell)}\right\} = (n\ell - km)Q_{(n+k-1,m+\ell-1)}\,.
\tag{5.8}
$$

One should also note that the set of vorticities $\omega(\mathbf{x})$ at each point $\mathbf{x}$ in space closes onto itself under the action of the Poisson brackets, forming the algebra

$$
\{\omega(\mathbf{x}), \omega(\mathbf{y})\} = \int d^2\mathbf{z}\, \boldsymbol{\nabla}_z \delta^2(z - x) \wedge \boldsymbol{\nabla}_z \delta^2(z - y)\omega(\mathbf{z})\,.
\tag{5.9}
$$

This is important as it ensures that the equations of motion for $\omega$, i.e. Euler's equation (2.31), do not depend on additional variables, in particular not the microscopic fields $\Phi(\mathbf{x})$. The Poisson brackets (5.9) define the algebra of volume preserving diffeomorphism $SDiff(\mathcal{M}_x)$ of the spatial manifold $\mathcal{M}_x$. Indeed, one can easily see that that the commutation relation (5.9), when rewritten in momentum space, is identical to the one given in eq. (5.6).

It is important to remark that while $SDiff(\mathcal{M}_\Phi)$ is a symmetry of the theory, this is not the case for $SDiff(\mathcal{M}_x)$.

## 5.2 The spectrum of the quantum incompressible fluid: Infinitely many vortons

Let us now try to quantize the action (5.1) directly in the continuum limit.[25] To this aim we simply promote the brackets (5.4) to commutation relations in the standard way

$$\left[\Phi(\mathbf{x}), \Phi^\dagger(\mathbf{y})\right] = \delta^2(\mathbf{x} - \mathbf{y}),\tag{5.10}$$

so that $\Phi$ and $\Phi^\dagger$ act as, respectively, annihilation and creation operators of a non-relativistic field theory. We additionally introduce a regulator function in the Hamiltonian to take care of UV divergences

$$H = -\frac{1}{2\rho_m} \int d^2\mathbf{x} \left(\boldsymbol{\nabla}\Phi^\dagger \wedge \boldsymbol{\nabla}\Phi\right) \frac{F\left(-\boldsymbol{\nabla}^2/\Lambda^2\right)}{-\boldsymbol{\nabla}^2} \left(\boldsymbol{\nabla}\Phi^\dagger \wedge \boldsymbol{\nabla}\Phi\right),\tag{5.11}$$

where $\Lambda$ is the cutoff scale and $F\left(\mathbf{k}/\Lambda^2\right) = 1 + O\left(\mathbf{k}^2/\Lambda^2\right)$ at small momentum. In the quantum theory however the commutation relation (5.10) introduces ordering ambiguities and we need to specify a certain ordering in the Hamiltonian (5.11). This choice is important as the appearance of a quadratic gradient term, which is protected by symmetry at the classical level, depends on ordering at the quantum level. This indicates that also the fate of the defining $SDiff(\mathcal{M}_\Phi)$ symmetry is decided by operator ordering.

Indeed, at the quantum level, the algebra of the generators (5.7) of the symmetry group is also modified. To illustrate that, let us choose to work with generators $Q_{(n,m)}$ specified by normal ordered products of the form (5.7). We then see the quantum algebra, compared to the classical one, acquires several new singular terms proportional to powers of $\delta^2(0)$:

$$\left[Q_{(n,m)}, Q_{(k,\ell)}\right] = \sum_{r=0}^\infty f_{(n,m),(k,\ell)}^{(n+k-1-r,m+\ell-1-r)} Q_{(n+k-1-r,m+\ell-1-r)} \left[\delta^2(0)\right]^r,\tag{5.12}$$

where the structure constants are given by

$$f_{(n,m),(k,\ell)}^{(n+k-1-r,m+\ell-1-r)} = \left[\frac{k!}{(k-1-r)!}\binom{m}{r+1} - \frac{n!}{(n-1-r)!}\binom{\ell}{r+1}\right],\tag{5.13}$$

which reduces to the former result (5.8) for $r = 0$. Note that the first and second terms in the square parenthesis of eq. (5.13) vanish, respectively, for $r > \min(k-1, m-1)$ and $r > \min(n-1, \ell-1)$. Therefore the sum in eq. (5.12) always truncates after a finite number of terms.

In this section we shall be cavalier about these issues. Whenever necessary, we will assume that the cutoff regulates delta function singularities as $\delta^2(0) \to \Lambda^2$, and we will not worry about the quantum corrections to the symmetry algebra.

We shall however insist that the algebra (5.9) of $SDiff(\mathcal{M}_x)$ generated by the vorticity $\omega = i\partial\Phi^\dagger \wedge \partial\Phi$ – is not modified by the ordering of the fields and is thus unchanged at the quantum level, replacing Poisson brackets by commutators

$$[\omega(\mathbf{x}), \omega(\mathbf{y})] = -i \int d^2\mathbf{z} \, \boldsymbol{\nabla}_z \delta^2(z - x) \wedge \boldsymbol{\nabla}_z \delta^2(z - y)\omega(\mathbf{z}),\tag{5.14}$$

---

[25]The discussion in this section bears some technical similarities with that in app. A of [27].

Writing the Hamiltonian (5.11) solely in terms of the vorticity $\omega$ thus ensures that the Heisenberg picture quantum equations of motion for $\omega$ still do not depend on any additional variables. This demand unambiguously resolves the ordering ambiguity: the Hamiltonian should be understood as written in eq. (5.11), and thus, **not normal ordered**. Somewhat reassuringly, since the combination $\left(\partial \Phi^\dagger \wedge \partial \Phi\right)$ commutes with all the $Q_{(n,m)}$'s, this ordering choice implies that the Hamiltonian still commutes with all the charges. We will provide a more careful justification of this prescription in the next section, where will analyze a discretized model, where the algebra is regulated and where eq. (5.11) is recovered in the continuum limit.

With these preliminaries in order, it is now straightforward to analyze the spectrum of the theory. To obtain the single-particle states spectrum, it is simplest to rewrite the Hamiltonian (5.11) in momentum space:

$$\Phi(\mathbf{x}) = \int \frac{d^2\mathbf{k}}{(2\pi)^2} e^{i\mathbf{k}\cdot\mathbf{x}} \Phi_\mathbf{k} \,. \tag{5.15}$$

We obtain

$$H = \frac{1}{2\rho_m} \int \frac{d^2\mathbf{p}_2 d^2\mathbf{p}_1 d^2\mathbf{k}_2 d^2\mathbf{k}_1}{(2\pi)^6} \delta^2\left(\mathbf{p}_1 + \mathbf{p}_2 - \mathbf{k}_1 - \mathbf{k}_2\right)$$
$$\times \frac{(\mathbf{p}_2 \cdot \mathbf{k}_1)(\mathbf{k}_2 \cdot \mathbf{p}_1) - (\mathbf{p}_2 \cdot \mathbf{p}_1)(\mathbf{k}_2 \cdot \mathbf{k}_1)}{(\mathbf{p}_1 - \mathbf{k}_1)^2} F\left(\frac{(\mathbf{p}_1 - \mathbf{k}_1)^2}{\Lambda^2}\right) \Phi_{\mathbf{p}_2}^\dagger \Phi_{\mathbf{k}_2} \Phi_{\mathbf{p}_1}^\dagger \Phi_{\mathbf{k}_1} \,. \tag{5.16}$$

Upon writing the Hamiltonian in normal ordered form we find

$$H = \frac{\Lambda^2 \tilde{F}(0)}{4\rho_m} \int \frac{d^2\mathbf{p}}{(2\pi)^2} \mathbf{p}^2 \Phi_\mathbf{p}^\dagger \Phi_\mathbf{p} + : H :, \tag{5.17}$$

where we defined the Fourier transform of the regulator function

$$\Lambda^2 \tilde{F}(\Lambda^2 \mathbf{x}^2) = \int \frac{d^2\mathbf{k}}{(2\pi)^2} e^{i\mathbf{k}\cdot\mathbf{x}} F\left(\frac{\mathbf{k}^2}{\Lambda^2}\right) \overset{\mathbf{x}\to 0}{\sim} \Lambda^2. \tag{5.18}$$

and $: H :$ is the normal ordered quartic Hamiltonian, which annihilates single particle states. The dispersion of single vorton states is thus purely controlled by the first (quadratic) term in eq. (5.17) and reads

$$\omega_\mathbf{p} = \frac{\Lambda^2 \tilde{F}(0)}{4\rho_m} \mathbf{p}^2 \,. \tag{5.19}$$

Note that the exact coefficient upfront $\tilde{F}(0)$ is not calculable from the low energy action alone. Yet the momentum dependence and the lack of a gap are a robust feature of the system.

Before discussing the symmetries of the quantum theory, it is worth to briefly recall the properties of the vortons, which were already studied in ref. [4]. Consider single vortons states first. According to eq. (4.31) (Gauss law), the vorticity $\omega(\mathbf{x})/\sqrt{\rho_m}$ can be interpreted as a

charge density. It is then interesting to consider the lowest charge (or vorticity) multipoles of a single vorton. Considering then a normalizable single vorton state

$$|\Psi\rangle \equiv \int d^2\mathbf{x}\,\psi(\mathbf{x})\Phi^\dagger(\mathbf{x})|0\rangle\,, \qquad \int d^2\mathbf{x}|\psi(\mathbf{x})|^2 = 1\,, \tag{5.20}$$

using integration by parts we find for respectively monopole and dipole

$$\frac{1}{\sqrt{\rho_m}}\int d^2\mathbf{x}\,\langle\Psi|\,\omega(\mathbf{x})\,|\Psi\rangle = \frac{1}{\sqrt{\rho_m}}\int d^2\mathbf{x}\boldsymbol{\nabla}_i(\psi^*\boldsymbol{\nabla}_j\psi)\epsilon^{ij} = 0\,, \tag{5.21}$$

and

$$\begin{aligned}
d^i &= \frac{1}{\sqrt{\rho_m}}\int d^2\mathbf{x}\,x^i\,\langle\Psi|\,\omega(\mathbf{x})\,|\Psi\rangle \\
&= \frac{i}{\sqrt{\rho_m}}\int d^2\mathbf{x}\,x^i\psi^*(\mathbf{x})\boldsymbol{\nabla}_i\psi(\mathbf{x})\epsilon^{ij} = \frac{-1}{\sqrt{\rho_m}}\epsilon^{ij}\langle P^j\rangle\,.
\end{aligned} \tag{5.22}$$

The vortons can thus be viewed as vortex-antivortex pairs [4] carrying vanishing total vorticity and a vorticity dipole proportional but orthogonal to the total momentum.[26]

While the quadratic part of eq. (5.17) controls the propagation of the asymptotic free vortons, the quartic part controls their scattering [4]. For instance, for $2 \to 2$ scattering, working at tree level and at small incoming momenta $\mathbf{k}_1$ and $\mathbf{k}_2$, the scattering amplitude is

$$\mathcal{M}(\mathbf{k}_1,\mathbf{k}_2,\mathbf{p}_1,\mathbf{p}_2) = \frac{1}{\rho_m}\frac{(\mathbf{k}_1\wedge\mathbf{p}_1)(\mathbf{k}_2\wedge\mathbf{p}_2)}{(\mathbf{k}_1-\mathbf{p}_1)^2} + \mathbf{k}_1\leftrightarrow\mathbf{k}_2\,. \tag{5.23}$$

We are now ready to discuss the consequences of the infinite symmetry at the basis of our construction. The main result here is that such symmetry implies the existence of an infinite set of independent asymptotic states which are formally composed of multiple vortons, but which behave in practice as single vortons. In order to see that, consider first a two particle state

$$\begin{aligned}
|\psi_2(\mathbf{k})\rangle &= \int d^2\mathbf{x}\,e^{i\mathbf{k}\cdot\mathbf{x}}\int d^2\mathbf{y}\,\Psi_{\mathbf{k}}(\mathbf{y})\Phi^\dagger\Big(\mathbf{x}+\tfrac{1}{2}\mathbf{y}\Big)\Phi^\dagger\Big(\mathbf{x}-\tfrac{1}{2}\mathbf{y}\Big)|0\rangle \\
&= \int \frac{d^2\mathbf{p}}{(2\pi)^2}\tilde{\Psi}_{\mathbf{k}}(\mathbf{p})\Phi^\dagger_{\frac{\mathbf{k}+\mathbf{p}}{2}}\Phi^\dagger_{\frac{\mathbf{k}-\mathbf{p}}{2}}|0\rangle\,,
\end{aligned} \tag{5.24}$$

---

[26]It behoves us to compare eq. (5.19) to what was concluded by Landau in his famous paper on superfluidity. Landau based his study on the quantum mechanical transcription of Euler's equation, with $\rho$ and $\mathbf{v}$ promoted to quantum operators. Upon noticing that the vorticity satisfies a non-abelian algebra, he argued that there should be a discontinuity between states of null vorticity and states with some vorticity, hence the latter should be gapped. By dimensional analysis, the gap, he concluded, should be of order $\Lambda^5/\rho_m$ in $d = 3$, corresponding to $\Lambda^4/\rho_m$ in our $d = 2$ case. We think Landau's argument is incorrect because the algebra is infinite and the commutation relation abelianizes at zero momentum. In fact as eq. (5.22) shows, vortons exhibit vanishing vorticity as $\mathbf{p}\to 0$. Moreover, just by the shift symmetry $\Phi\to\Phi+c$ of the Clebsch lagrangian it is clear that vortons should be gapless. By wrongly concluding that the $\Phi$'s should be gapped, Landau's system reduced to the theory of the derivatively coupled $\chi$ around the background $\chi = \mu t$. That is precisely the EFT of the superfluid [40], the system Landau could have started with in the first place.

where $\mathbf{k}$ is the total momentum and $\Psi(\mathbf{y})$ is the wave-function. Then the Schrödinger equation

$$H|\psi_2(\mathbf{k})\rangle = E_\mathbf{k}|\psi_2(\mathbf{k})\rangle \,, \tag{5.25}$$

results in the following

$$\begin{aligned}
E_\mathbf{k}\Psi_\mathbf{k}(\mathbf{x}) =& \frac{\Lambda^2\mathbf{k}^2}{8\rho_m}\left[\tilde{F}(0) + \tilde{F}(\Lambda^2\mathbf{x}^2)\right]\Psi_\mathbf{k}(\mathbf{x}) - \frac{\Lambda^2\nabla_\mathbf{x}^2}{8\rho_m}\left\{\left[\tilde{F}(0) - \tilde{F}(\Lambda^2\mathbf{x}^2)\right]\Psi_\mathbf{k}(\mathbf{x})\right\} \\
&- \frac{1}{4\rho_m}\left(k^i k^j + \partial_\mathbf{x}^i\partial_\mathbf{x}^j\right)\left[V_{ij}(\mathbf{x})\Psi_\mathbf{k}(\mathbf{x})\right] \,,
\end{aligned} \tag{5.26}$$

where we defined

$$V_{ij}(\mathbf{x}) = \int \frac{d^2\mathbf{q}}{(2\pi)^2}e^{i\mathbf{q}\cdot\mathbf{x}}\frac{q_i q_j - \mathbf{q}^2\delta_{ij}/2}{\mathbf{q}^2}F\left(\frac{\mathbf{q}^2}{\Lambda^2}\right) \,. \tag{5.27}$$

Note that

$$\delta^{ij}V_{ij}(\mathbf{x}) = 0 \quad \text{and} \quad V_{ij}(0) = 0 \,. \tag{5.28}$$

Surprisingly we see that a solution of eq. (5.26) is given by a completely localized wave-function

$$\Psi_\mathbf{k}(\mathbf{x}) \propto \delta^2(\mathbf{x}) \quad \text{and} \quad E_\mathbf{k} = \frac{\Lambda^2\tilde{F}(0)}{4\rho_m}\mathbf{k}^2 \,, \tag{5.29}$$

which is exactly degenerate with the single particle vorton state! Note that this is a very unusual feature, since normally a wave-function localized at such short distances is not under control within EFT (indeed the wave function (5.29) is not normalizable, because of its singular UV property). Yet, we claim that the existence of this two-particle bound state, degenerate with the single-vorton state, is a robust feature of the perfect fluid in the Clebsch formulation. In fact, we claim that there is a similar $n$-vorton bound state for any $n > 0$! This is because, as we explain below, these results originate from the invariance of the Hamiltonian under the charges (5.5). Moreover the regulated model we will present in the next section offers an example where the result is robust with respect to UV effects.

The intuitive reason for the existence of such bound states was anticipated in [4]. At the quantum level, not all symmetry charges (5.5) annihilate the vacuum. In particular, even working in conventions such that all charges are normal ordered, all charges $Q_{(n,0)} = \int d^2\mathbf{x}(\Phi^\dagger(\mathbf{x}))^n$ act nontrivially and generate a state that is degenerate with the vacuum. Taking a wave-function $\Psi_\mathbf{k}(\mathbf{y}) \propto \delta^2(\mathbf{y})$ in eq. (5.24) we create a state that in the $\mathbf{k} \to 0$ limit approaches the degenerate vacuum $Q_{(2,0)}|0\rangle$. The argument we are offering here is just a fast description of Goldstone's theorem,[27] but it simply generalizes the argument for the existence of a single-particle vorton state starting from the charge $Q_{(1,0)}$. By further generalizing the argument, we conclude there exists a similar vorton bound-state for all charges $Q_{(n,0)}$, given they do not annihilate the vacuum.[28] No other universal light states are expected, since in the

---

[27]Indeed $Q_{(n,0)}|0\rangle$ is not normalizable at infinite volume, so that strictly speaking it sits outside the Hilbert space.

[28]We thank Dam Son for asking the mandatory simple question that forced us to figure this out.

normal ordered basis the remaining charges, which all possess at least a destruction operator, form a subalgebra that annihilates the vacuum and is left therefore *unbroken*.[29]

The symmetry algebra also explains the exact degeneracy of the single particle state and two-particle bound state. That is because the state (5.24) is obtained acting on a single particle state $|\psi_1(\mathbf{k})\rangle = \Phi_{\mathbf{k}}^\dagger|0\rangle$ with the charge

$$Q_{(2,1)} = \int d^2\mathbf{x} \left(\Phi^\dagger(\mathbf{x})\right)^2 \Phi(\mathbf{x}) . \tag{5.30}$$

This argument immediately explains the degeneracy of the two-particle bound state with the single-particle state. By considering similar charges we see that all such multi-particle bound states are degenerate with the single-vorton states (and are essentially identical to a single-particle vorton state in all aspects but the particle number charge).

We remark again that the vorton bound states that we just constructed are very unusual from the viewpoint of effective field theory. Indeed we found that by bringing single-particle states very close to each other - a naively illegal operation within EFT - we obtain a low energy state, which can be thought of as a Goldstone boson due to the spontaneous breaking of the symmetry algebra. It is not possible to create such bound states via low energy scatterings of single vortons. That is because these states transform non-trivially under the unbroken symmetry group: for instance the two-vorton bound state is an eigenstate of $Q_{(2,2)}$, which instead annihilates states made of two vortons at finite distance from each other. This situation is reminiscent of the UV/IR mixing phenomena which are observed in certain exotic field theories of fractons [8, 9]. As in those cases, the unusual behaviour is made possible by the infinitely large symmetry group, which forbid operators - such as $\sim |\Phi|^4$ - which would lift the bound states up to the cut-off.

In conclusion, we argued that the perfect quantum fluid in the Clebsch formulation admits infinitely many light vorton bound states, one for each particle number $n > 0$, with energy given by (5.19). Their existence and degeneracy is a robust consequence of the symmetry algebra. Yet, the precise coefficient of their dispersion relation naively depends on all higher derivative operators, thus defying the standard EFT logic.

In the next section we will rederive these results within the lattice model introduced in [4], which reduces to (5.11) in the continuum limit. We will confirm the results of this section on the spectrum and on the degeneracy of the vortons.

## 6  2d fluids as $SU(\infty)$-matrix model

### 6.1  Vorton matrix model

It is simpler to study the theory in finite volume, by going on the torus, so $x^i \in [-\pi r, \pi r]$, for $i = 1, 2$ and $r$ fixed. We further introduce a square lattice discretization $\mathbf{x} = 2\pi r \, \mathbf{m}/N$,

---

[29]Going to a real basis of generators, this implies that there is a Goldstone boson for each two broken generators $\{Q_{(n,0)} + Q_{(n,0)}^\dagger, iQ_{(n,0)} - iQ_{(n,0)}^\dagger\}$, in agreement with the non-relativistic counting rules [54–56].

where $N$ is a large integer, $\mathbf{m} = (m_1, m_2) \in \mathbb{N}^2$, and $a \equiv 2\pi r/N$ is the lattice spacing, such that for $N \to \infty$ we recover the continuum. It will be convenient to consider $N$ odd, so that the $m_i$ cover all integers in the range $[-(N-1)/2, (N-1)/2]$.

It will be simpler to work in momentum space. This is also the latticed torus, like position space, with the components of the wave vector $\mathbf{n} = (n_1, n_2)$ taking integer values in the range $[-(N-1)/2, (N-1)/2]$. The Fourier transform of the vorton fields is thus defined as

$$\Phi(\mathbf{x}) = \frac{1}{(2\pi r)^2} \sum_{\mathbf{n}} \Phi^{\mathbf{n}} e^{i\frac{\mathbf{n}\cdot\mathbf{x}}{r}}, \qquad \Phi^{\dagger}(\mathbf{x}) = \frac{1}{(2\pi r)^2} \sum_{\mathbf{n}} \Phi^{\dagger}_{\mathbf{n}} e^{-i\frac{\mathbf{n}\cdot\mathbf{x}}{r}}, \tag{6.1}$$

where the sum runs over the above mentioned range of $\mathbf{n}$.[30] Notice that, consistently with the above, the momentum modes $\Phi^{\mathbf{n}}$ and $\Phi^{\dagger}_{\mathbf{n}}$ are invariant under the shift $\mathbf{n} \sim \mathbf{n} + (N, 0)$, $\mathbf{n} \sim \mathbf{n} + (0, N)$. Like for the space coordinates, also the physical momentum $\mathbf{p} = \frac{1}{r}\mathbf{n}$, has then entries $p^i$ taking discrete values in the range $p^i \in [-(N-1)/2r, (N-1)/2r] \sim [-\pi/a, \pi/a]$. The lattice version of the canonical commutation relations (5.10) is

$$[\Phi^{\mathbf{m}}, \Phi^{\dagger}_{\mathbf{n}}] = (2\pi r)^2 \delta^{\mathbf{m}}_{\mathbf{n}}. \tag{6.3}$$

All we have to do then is to write a lattice Hamiltonian that reduces to (5.11) in the limit $N \to \infty$. To this aim note that eq. (5.11) can be written purely in terms of the vorticity.

The obvious guess for the lattice Hamiltonian is simply

$$H = \frac{1}{2\rho_m (2\pi)^2} \sum_{\mathbf{n}} \frac{F(\mathbf{n}^2/\bar{\Lambda}^2)}{\mathbf{n}^2} \omega_{\mathbf{n}} \omega_{-\mathbf{n}}, \qquad \bar{\Lambda} \equiv r\Lambda, \tag{6.4}$$

where the vorticity is obtained by discretizing $\omega(\mathbf{x}) = i\boldsymbol{\nabla}\Phi^{\dagger}(\mathbf{x}) \wedge \boldsymbol{\nabla}\Phi(\mathbf{x})$:

$$\hat{\omega}(\mathbf{x}) = i\Delta_1 \Phi^{\dagger}(\mathbf{x})\Delta_2\Phi(\mathbf{x}) - i\Delta_2\Phi^{\dagger}(\mathbf{x})\Delta_1\Phi(\mathbf{x}). \tag{6.5}$$

Here $\Delta$ is the lattice derivative, defined as

$$\Delta_k\Phi(\mathbf{x}) = \frac{1}{2a}\left[\Phi(\mathbf{x} + a\hat{n}_k) - \Phi(\mathbf{x} - a\hat{n}_k)\right], \qquad \hat{n}_k = \begin{cases} (1,0) & \text{for } k = 1 \\ (0,1) & \text{for } k = 2 \end{cases}. \tag{6.6}$$

Therefore in momentum space we obtain

$$\hat{\omega}_{\mathbf{n}} = \frac{iN^2}{(2\pi r)^4} \sum_{\mathbf{m}} \Phi^{\dagger}_{\mathbf{m}+\mathbf{n}}\Phi^{\mathbf{m}} \left[\sin\left(2\pi\frac{m_1 + n_1}{N}\right)\sin\left(2\pi\frac{m_2}{N}\right) - \sin\left(2\pi\frac{m_2 + n_2}{N}\right)\sin\left(2\pi\frac{m_1}{N}\right)\right]. \tag{6.7}$$

Where $\omega(\mathbf{x})$ and its Fourier transform $\omega_{\mathbf{n}}$ are related as in eq. (6.1).

---

[30]This normalization for the Fourier transform is convenient because in the continous limit one recovers the standard relation

$$\Phi^{\mathbf{n}} = \frac{(2\pi r)^2}{N^2} \sum_{i} \phi(\mathbf{x}_i) e^{-i\frac{\mathbf{n}\cdot\mathbf{x}_i}{r}} \to \int d^2x \, \Phi(\mathbf{x}) e^{-i\frac{\mathbf{n}\cdot\mathbf{x}}{r}} \tag{6.2}$$

In the continuum limit, this expression reduces to

$$\hat{\omega}_{\mathbf{n}} \xrightarrow{N \to \infty} \omega_{\mathbf{n}} = \frac{i}{(2\pi)^2 r^4} \sum_{\mathbf{m}} (\mathbf{n} \wedge \mathbf{m}) \Phi^{\dagger}_{\mathbf{m}+\mathbf{n}} \Phi^{\mathbf{m}}, \tag{6.8}$$

and the commutation relations of $\omega_{\mathbf{n}}$ reproduces the algebra (5.14) of $SDiff(\mathcal{M}_x)$, which in momentum space reads

$$[\omega_{\mathbf{n}}, \omega_{\mathbf{m}}] = \frac{i}{r^2} (\mathbf{n} \wedge \mathbf{m}) \, \omega_{\mathbf{n}+\mathbf{m}} \,. \tag{6.9}$$

There are however some issues with these expressions, in particular (6.5). At the technical level, the issue is that the commutator $[\omega(\mathbf{x}), \omega(\mathbf{y})]$ cannot be expressed anymore just in terms of the vorticity itself at finite $N$,[31] and reduces to the $SDiff(\mathcal{M}_x)$ algebra only for $N \to \infty$. This in turn implies that the equation of motion for the vorticity operator depends explicitly on the microscopic variables $\Phi$, $\Phi^{\dagger}$, unlike the continuum theory where Euler's equation (2.31) can be written purely in terms of $\omega(\mathbf{x})$. Relatedly, but at the more conceptual level, the lattice Hamiltonian (6.4) is not invariant under the action of all the charges (5.7). The shift symmetry $\Phi \to \Phi + c$, generated by $\{Q_{(1,0)}, Q_{(0,1)}\}$, and the $SL(2, \mathbb{R})$ subgroup, generated by the charges $\{Q_{(2,0)}, Q_{(0,2)}, Q_{(1,1)}\}$, are the only surviving internal symmetries of the model.

In principle, we could work with the naive model described above. However, as we discussed in sec. 4.4, the infinitely many conserved charges of the Clebsch theory ensure the existence of the many stationary flows that characterize what we commonly mean by a (classical) fluid. Therefore, already at the classical level, the naive model above, which only possesses a finite symmetry group, may at most mimic the behaviour of a fluid for a finite range of scales and times, which we expect to become larger and larger as we increase the cutoff. While it would be interesting to quantify these considerations, in the present work we take a different approach. Below we construct a different lattice model, with infinitely many conserved charges, that may be thought as deformations of the continuum ones (5.7).

To this aim, we use that the algebra of area-preserving diffeomorphism $SDiff(\mathcal{M}_x)$ on the torus can be seen as the limit $N \to \infty$ of the $SU(N)$ algebra [58]. The key insight of [4] is that this mathematical fact can be used to define a different discretized model, where the commutator algebra of the vorticity is closed and coincides with $SU(N)$. It turns out that this model is also invariant under the action of a modified version of the charges (5.7).

Let us briefly review the basic facts about $SU(N)$ that are needed for our construction. Like for the latticized model we shall focus on the case of odd $N$. For our purposes it is convenient (in fact necessary) to work in the 't Hooft basis of the $SU(N)$ algebra (see [57–61]), in which the generators are labelled by a non-null vector $\mathbf{n} = (n_1, n_2)$ with integer entries in the range $[-(N-1)/2, (N-1)/2]$. Notice that this set consists indeed of $N^2 - 1$ elements.

---

[31]Mathematically eq. (6.7) has the same structure $\Phi^{\dagger}_I \Phi_J F_{IJK}$ as the Jordan-Schwinger realization of a Lie algebra via harmonic oscillators shown in eq. (A.10). For the algebra to close the structure constants $F_{IJK}$ should satisfy the Jacobi identity, but the coefficient function in 6.7 fails this test. A similar remark already appeared in [57].

In the fundamental representation [61] the generators are $N \times N$ traceless matrices satisfying

$$T_{\mathbf{n}}T_{\mathbf{m}} = \omega^{\frac{\mathbf{n} \wedge \mathbf{m}}{2}}T_{\mathbf{n+m}}, \tag{6.10}$$

where $\omega = e^{2\pi i/N}$, and

$$Tr(T_{\mathbf{n}}T_{\mathbf{m}}) = N\delta_{\mathbf{n+m},0}. \tag{6.11}$$

The explicit form of the $T_{\mathbf{n}}$ can be found in the above mentioned papers, but is immaterial for this discussion. What matters is eq. (6.10), which implies the commutator is

$$[T_{\mathbf{n}}, T_{\mathbf{m}}] = 2i \sin\left(\frac{\pi}{N}\mathbf{n} \wedge \mathbf{m}\right)T_{\mathbf{n+m}}. \tag{6.12}$$

Now, rescaling $T_{\mathbf{n}} = 2\pi\tilde{T}_{\mathbf{n}}/N$ and formally taking the limit $N \to \infty$ we obtain $[\tilde{T}_{\mathbf{n}}, \tilde{T}_{\mathbf{m}}] = i(\mathbf{n} \wedge \mathbf{m})\tilde{T}_{\mathbf{n+m}}$, which coincides with the $SDiff(\mathcal{M}_x)$ algebra (5.14) (in momentum space).

In order to encompass also the zero mode of the field, we will actually need to extend the 't Hooft construction to $U(N)$. That is simply accomplished by extending the 't Hooft basis to include the identity matrix, with label $\mathbf{n} = (0,0)$, i.e. $T_{\mathbf{0}} = \mathbb{1}_{N \times N}$. With this identification in place, it is simple to check that eqs. (6.10), (6.11) and (6.12) hold unchanged for $U(N)$.

We can now leverage these results to formulate a discretized model suitable to our purposes. First, starting from the canonically conjugated Fourier modes in eq. (6.1), we build complex matrices $\Phi_{ij}$ and $\Phi_{ij}^{\dagger}$ as

$$\begin{aligned}
\bar{\Phi}_{ij} &= \sum_{\mathbf{n}}(T_{\mathbf{n}})_{ij}\Phi^{\mathbf{n}}, \\
\bar{\Phi}_{ij}^{\dagger} &= \sum_{\mathbf{n}}\Phi_{\mathbf{n}}^{\dagger}(T_{-\mathbf{n}})_{ij},
\end{aligned} \tag{6.13}$$

where the sums run over all the elements of $U(N)$ including the $U(1)$ factor, corresponding to 0-mode $\Phi^{\mathbf{0}}$. We then identify the vorticity with the, suitably normalized, $SU(N)$ charges[32]

$$\tilde{\omega}_{\mathbf{n}} = \frac{1}{(2\pi)^3 r^4}\text{Tr}(\bar{\Phi}^{\dagger}[T_{\mathbf{n}}, \bar{\Phi}]) = \frac{1}{(2\pi)^3 r^4}\text{Tr}(T_{\mathbf{n}}[\bar{\Phi}, \bar{\Phi}^{\dagger}]). \tag{6.14}$$

where we used the cyclicity of the trace in the last step. Eq. (6.14) implies that $\tilde{\omega}_{\mathbf{0}} = 0$, corresponding to vorticity being a total derivative. By construction, $\tilde{\omega}_{\mathbf{n}}$ generates the adjoint $SU(N)$ action on $\Phi$ and $\Phi^{\dagger}$:

$$e^{i\sum_{\mathbf{n}}\alpha_{\mathbf{n}}\tilde{\omega}_{\mathbf{n}}}\bar{\Phi}e^{-i\sum_{\mathbf{n}}\alpha_{\mathbf{n}}\tilde{\omega}_{\mathbf{n}}} = U\bar{\Phi}U^{\dagger} \tag{6.15}$$

with $\alpha_{\mathbf{n}}$ the $SU(N)$ Lie parameter and $U$ the corresponding $SU(N)$ element in the fundamental representation. Notice that vorticity acts trivially on the $SU(N)$ singlet zero-mode $\Phi^{\mathbf{0}}$. Using (6.12), we can write (6.14) explicitly as

$$\begin{aligned}
\tilde{\omega}_{\mathbf{n}} &= \frac{i}{(2\pi)^2 r^4}f_{\mathbf{nm}}{}^{\mathbf{k}}\Phi_{\mathbf{k}}^{\dagger}\Phi^{\mathbf{m}} \\
&= \frac{i}{(2\pi)^2 r^4}\sum_{\mathbf{m}}\frac{N}{\pi}\sin\left(\frac{\pi}{N}\mathbf{n} \wedge \mathbf{m}\right)\Phi_{\mathbf{m+n}}^{\dagger}\Phi^{\mathbf{m}}.
\end{aligned} \tag{6.16}$$

_____________________

[32]Recall the Jordan-Schwinger construction and that, from eq. 6.3, the $\{\Phi^{\mathbf{n}}, \Phi_{\mathbf{n}}^{\dagger}\}$ are a set of independent ladder operators.

It also follows immediately that the commutator of the vorticity yields the $SU(N)$ algebra

$$[\tilde{\omega}_\mathbf{n}, \tilde{\omega}_\mathbf{m}] = \frac{i}{r^2} \frac{N}{\pi} \sin\left(\frac{\pi}{N} \mathbf{n} \wedge \mathbf{m}\right) \tilde{\omega}_{\mathbf{n}+\mathbf{m}}. \tag{6.17}$$

Eqs. (6.16) and (6.17) reduce to the continuum expressions (6.9) for $|\mathbf{n}|, |\mathbf{m}| \ll \sqrt{N}$. This suggests to define a regulated Hamiltonian by simply making the replacement $\omega \to \tilde{\omega}$ in (6.4):

$$H = \frac{1}{2\rho_m(2\pi)^2} \sum_\mathbf{n} \frac{F(\mathbf{n}^2/\bar{N}^2)}{\mathbf{n}^2} \tilde{\omega}_\mathbf{n} \tilde{\omega}_{-\mathbf{n}}. \tag{6.18}$$

Indeed in order for the dynamics to approximate that of the perfect fluid we should also choose $F(\mathbf{n})$ to have support in the range of $\mathbf{n}$ where the $SU(N)$ algebra (6.17) approximates the $SDiff(\mathcal{M}_x)$ algebra, that is for $|\mathbf{n}| \lesssim \bar{N}/2 \lesssim \sqrt{N}/2$. That way eq. (6.18) reduces to the continuum Hamiltonian (5.11) when acting on states with $|\mathbf{n}| \ll \bar{N}$. We then get back the fluid Hamiltonian (5.11) in the limit $N \to \infty$ for states with low momentum. When restoring dimensional units, the wave number cut-off corresponds to a physical momentum cut-off $\Lambda = \bar{N}/r$. This construction corresponds to the $SU(N)$ matrix model of [4]. It should however be stressed that, even though the Hamiltonians expressed in term of the regulated vorticity coincide in the two constructions, ref. [4] was based on the comoving formulation in such a way that $\omega_\mathbf{n}$ was not represented in terms of vorton fields. The description in terms of vorton variables was there obtained sort of inductively in a second step. Here we have instead followed a deductive procedure, starting from the Clebsch formulation, which from the start features the vorton variables.

In the next subsection we will study the spectrum of the model (6.18) in detail, while below we comment on its symmetries. However, before doing that, we would like to remark that the regulated vorticity (6.16) matches the continuum result (6.8) only for field configurations that are smooth on lengths $\lesssim O(\sqrt{N})$ lattice spacing. Equivalently: when expressed in position space, (6.16) is dominated by fields at relative lattice distances ranging up to $O(\sqrt{N})$. That is significantly more non-local than the naive discretization (6.7). In practice, as already hinted above and as amply discussed in [4], the role of regulator length is played by $a\sqrt{N}$. This makes the matrix model at hand quite peculiar compared to more standard constructions where only fields separated by a few lattice steps are coupled.

Two facts determine the symmetries of the system: 1) the Hamiltonian (6.18) is purely a function of the $SU(N)$ generators, 2) the vorton matrices $\bar{\Phi}$, $\bar{\Phi}^\dagger$ transform as $SU(N)$ adjoints, see eq. (6.15). It then follows that any $SU(N)$ singlet built out of $\bar{\Phi}$ and $\bar{\Phi}^\dagger$ commutes with $\tilde{\omega}_\mathbf{n}$, hence with the Hamiltonian. The singlets are now simply the traces of any product of $\bar{\Phi}$ and $\bar{\Phi}^\dagger$. Like in sec. 5 we can pick a basis of normal ordered operators:

$$\begin{aligned}
\tilde{Q}_{(n,m)} &= \frac{1}{(2\pi r)^{2n+2m-2}N} \mathrm{Tr}\left((\bar{\Phi}^\dagger)^n(\bar{\Phi})^m\right) \\
&= \frac{1}{(2\pi r)^{2n+2m-2}} \sum_{\mathbf{k}_1 ... \mathbf{k}_{n+m}} \delta^0_{\mathbf{k}_{tot}} \left(\Phi^\dagger_{-\mathbf{k}_1} ... \Phi^\dagger_{-\mathbf{k}_n} \Phi^{\mathbf{k}_{n+1}} ... \Phi^{\mathbf{k}_{n+m}}\right) \prod_{i<j} \omega^{-\frac{1}{2}\mathbf{k}_i \wedge \mathbf{k}_j},
\end{aligned} \tag{6.19}$$

where, in the second line, we used the properties of the 't Hooft matrices. Note that the elements $\tilde{Q}_{(1,0)}$ and $\tilde{Q}_{(0,1)}$ coincide with the field zero modes $\Phi_{\mathbf{0}}^{\dagger}$ and $\Phi^{\mathbf{0}}$, that do not appear in the Hamiltonian and thus trivially commute with it. For $N \to \infty$ we can, at least naively, take $\omega = e^{2\pi i/N} \to 1$ finding that the lattice charges reduce to those of the continuum theory shown in eq. (5.7):

$$
\begin{aligned}
\tilde{Q}_{(n,m)} \to Q_{(n,m)} &= \frac{1}{(2\pi r)^{2n+2m-2}} \sum_{\mathbf{k}_1...\mathbf{k}_{n+m}} \delta^0_{\mathbf{k}_{tot}} \left( \Phi^{\dagger}_{-\mathbf{k}_1}...\Phi^{\dagger}_{-\mathbf{k}_n} \Phi^{\mathbf{k}_{n+1}}...\Phi^{\mathbf{k}_{n+m}} \right) \\
&= \int d\mathbf{x} (\Phi^{\dagger}(\mathbf{x}))^n \Phi^m(\mathbf{x}) .
\end{aligned}
\tag{6.20}
$$

Stating things more precisely: $\tilde{Q}_{(n,m)}$ reduces to the continuum result (5.7) if we formally restrict the sum to momenta $\mathbf{n} \ll \sqrt{N}$, and therefore by considering matrix elements of $\tilde{Q}_{(n,m)}$ over states supported on such low momentum range. The exact expression of $\tilde{Q}_{(n,m)}$ is however different since the sum in eq. (6.19) includes large momenta. One can verify directly that, also at finite $N$, that $H$ is unaffected by the transformations generated by $\tilde{Q}_{(n,m)}$ :

$$
\begin{aligned}
\Phi^{\mathbf{n}} &\to \Phi^{\mathbf{n}} + (2\pi r)^2 \mathrm{Tr}((T_{-\mathbf{n}})_{ij} \partial_{\bar{\Phi}^{\dagger}_{ji}} \tilde{Q}_{(n,m)}) = \Phi^{\mathbf{n}} + (2\pi r)^2 \partial_{\Phi^{\dagger}_{\mathbf{n}}} \tilde{Q}_{(n,m)} , \\
\Phi^{\dagger}_{\mathbf{n}} &\to \Phi^{\dagger}_{\mathbf{n}} - (2\pi r)^2 \mathrm{Tr}((T_{\mathbf{n}})_{ij} \partial_{\bar{\Phi}_{ji}} \tilde{Q}_{(n,m)}) = \Phi^{\dagger}_{\mathbf{n}} - (2\pi r)^2 \partial_{\Phi^{\mathbf{n}}} \tilde{Q}_{(n,m)} .
\end{aligned}
\tag{6.21}
$$

This discussion holds unchanged at the quantum and at the classical level. The invariance under the charges (6.19) is what makes the Hamiltonian (6.18) technically natural.

Rather embarassingly, we have been unable to identify the symmetry group generated by the $\tilde{Q}_{(n,m)}$, either at the classical or at the quantum level. Let us explain the issue, starting from the simplest case: the $SU(2)$ matrix model. Note that this limiting case is equivalent to a different formulation of the rigid body as we explain in app A.

For $N = 2$, neglecting the zero modes that are anyhow decoupled, there are three independent traces: $\mathrm{Tr}(\bar{\Phi}^2)$, $\mathrm{Tr}((\bar{\Phi}^{\dagger})^2)$ and $\mathrm{Tr}(\bar{\Phi}\bar{\Phi}^{\dagger})$. Their algebra closes and coincides with the $SL(2, \mathbb{R})$ algebra. Thanks to the properties of Pauli matrices, all other traces with higher power of the fields can be decomposed into product of these $SL(2, \mathbb{R})$ generators, and therefore do not generate independent transformations and do not form a group when exponentiated. The algebra generated by the higher traces, and more in general the algebra generated by the products of some symmetry charges, is the universal enveloping algebra (see e.g. [62]).

For $N > 2$ there are $N^2 - 1$ independent traces. These always include the $SL(2, \mathbb{R})$ generators, but also traces with higher powers of the fields. For instance, for $N = 3$ for the 5 additional generators we could take $\mathrm{Tr}(\bar{\Phi}^3)$, $\mathrm{Tr}((\bar{\Phi}^{\dagger})^3)$, $\mathrm{Tr}(\bar{\Phi}^{\dagger}\bar{\Phi}^2)$, $\mathrm{Tr}((\bar{\Phi}^{\dagger})^2\bar{\Phi})$ and $\mathrm{Tr}((\bar{\Phi}^{\dagger})^2\bar{\Phi}^2)$. The Poisson brackets (or the commutator) between an operator with $n$ fields and one with $m$ fields result into an object with $n + m - 2$ fundamental fields (unless it vanishes). Therefore it is clear that any algebra generated by a finite number of traces with $n > 2$ fields is not closed. It might perhaps be possible to construct a closed algebra considering non-polynomial functions of the fields, but we were not able to achieve that.

## 6.2 Spectrum of the matrix model

In view of the infinite symmetry identified in the previous section we expect the spectrum of our system to feature severe degeneracies. As the conserved charges (6.19) come in all powers of the ladder operators $\Phi_{\mathbf{n}}^\dagger$, $\Phi^{\mathbf{n}}$, the degeneracies will be among states with different vorton number. We will still organize our discussion starting from the states with fixed vorton number.

In order to proceed, it is convenient to normal order the Hamiltonian:

$$
\begin{aligned}
H = &\frac{2N^2}{(2\pi r)^6 \rho_m} \sum_{\mathbf{n}} \Phi_{\mathbf{n}}^\dagger \Phi^{\mathbf{n}} \sum_{\mathbf{m}\neq 0} \frac{F(\mathbf{m}^2/\bar{N}^2)}{\mathbf{m}^2} \sin^2\left(\frac{\pi}{N}(\mathbf{m}\wedge\mathbf{n})\right) \\
&+ \frac{2N^2}{(2\pi r)^8 \rho_m} \sum_{\mathbf{n}} \frac{F(\mathbf{n}^2/\bar{N}^2)}{\mathbf{n}^2} : \omega_{\mathbf{n}}\omega_{-\mathbf{n}} : \, .
\end{aligned}
\tag{6.22}
$$

We can now work out the spectrum starting from the Fock vacuum, which satisfies $\Phi^{\mathbf{n}}|0\rangle = 0$ $\forall\,\mathbf{n}$ and which obviously has vanishing energy. Consider then the action of the conserved charges (6.19) on $|0\rangle$. While the vast majority of them annihilates the vacuum, the subset $\tilde{Q}_{(n,0)}$ $\forall\,n$, acts non-trivially producing multivorton states that are exactly degenerate with the vacuum. Notice that for $n > N$ the $\tilde{Q}_{(n,0)}$ are not independent and can be written in terms of products of the $\tilde{Q}_{(n,0)}$ with $n \leq N$. We thus conclude that the set of independent states with zero energy is given by

$$
\prod_{n=1}^{n=N} (\tilde{Q}_{(n,0)})^{q_n}|0\rangle\,, \qquad q_n \in \mathbb{N}\,.
\tag{6.23}
$$

The infinity of this degeneracy is associated to the non-compactness of the symmetry algebra, which already for $N = 2$ coincides with $SL(2,R)$. The existence of an infinite degeneracy of ground states at finite volume sets our system apart from ordinary systems with internal global symmetry which at finite volume normally have a single vacuum, albeit with level splittings controlled by the inverse power of the volume. However, in practice, this property may not be an issue once one takes the infinite volume limit. On the one hand, the charges are not local operators, so that all these states sit outside the Hilbert space that is constructed by the algebra of local observables. On the other hand, we can always consider, as it is standard in ordinary condensed matter systems, the presence of symmetry breaking effects at the boundary, which fully lifts the degeneracy. In particular the zero vorton state $|0\rangle$ may result as the true and unique ground state. But of course, even if the degeneracy of vacua does not survive the infinite volume limit, the existence of the symmetry still has consequences, encapsulated by Goldstone's theorem, as we shall see below.

Consider now the generic single vorton state $|\mathbf{n}\rangle = \Phi_{\mathbf{n}}^\dagger|0\rangle$. One easily sees it is an

eigenstate of the Hamiltonian with energy

$$
\begin{aligned}
E_{\mathbf{n}} &= \frac{2N^2}{(2\pi r)^4 \rho_m} \sum_{\mathbf{m} \neq 0} \frac{F(\mathbf{m}^2/\bar{N}^2)}{\mathbf{m}^2} \sin^2\left(\frac{\pi}{N}(\mathbf{m} \wedge \mathbf{n})\right) \\
&= \frac{C\bar{N}^2}{(2\pi r)^4 \rho_m} \mathbf{n}^2 \left[1 + O\left(\frac{\bar{N}^2 \mathbf{n}^2}{N^2}\right)\right] = \frac{C\Lambda^2 \mathbf{p}^2}{16\pi^2 \rho_m} \left[1 + O\left(\frac{\bar{N}^4}{N^2}\frac{\mathbf{p}^2}{\Lambda^2}\right)\right],
\end{aligned}
\tag{6.24}
$$

where in the second line we expanded for $|\mathbf{n}| \ll \bar{N} \lesssim \sqrt{N}$ and used that $F$ is supported at $\mathbf{m} \lesssim \bar{N}$, while $C$ is an $O(1)$ parameter that depends on the precise form of $F$. We see that for $N \to \infty$, with $\bar{N}$ fixed we recover the result (5.19). On the other hand if we keep the ratio $\bar{N}/\sqrt{N}$ fixed we recover the same continuum theory up to higher derivative terms controlled by $\bar{\Lambda}$.

Sticking to the finite $N$ case, we recall that $\Phi_{\mathbf{n}}^\dagger$ decomposes as the $SU(N)$ singlet, $\Phi_{\mathbf{0}}^\dagger$, and adjoint $\Phi_{\mathbf{n} \neq \mathbf{0}}^\dagger$. The state $\Phi_{\mathbf{0}}^\dagger|0\rangle$ is degenerate with the vacuum and coincides with special case $\tilde{Q}_{(1,0)}|0\rangle$ in the class of eq. (6.23). Focussing instead on the genuine vorton adjoint states $\Phi_{\mathbf{n} \neq \mathbf{0}}^\dagger|0\rangle$, we can again, like for the vacuum, construct an infinite class of states with precisely the same energy. The charges that serve that purpose are the $\tilde{Q}_{(m,1)}$ with $m \geq 2$. Notice, as before that for $m > N$ the $\tilde{Q}_{(m,1)}$ are not independent and can be written in terms of products of one $\tilde{Q}_{(n,1)}$ and in principle several $\tilde{Q}_{(k_i,0)}$ all with $n$ and $k_i$ less or equal to $N$. Very much like for eq. (6.23) the set of states that are exactly degenerate with a single vorton are given by

$$
|q, \mathbf{n}\rangle \equiv \tilde{Q}_{(q,1)} \Phi_{\mathbf{n}}^\dagger|0\rangle = \frac{1}{(2\pi r)^{2q-2} N} \mathrm{Tr}(T_{\mathbf{n}}(\bar{\Phi}^\dagger)^q)|0\rangle, \qquad 2 \leq q \leq N
\tag{6.25}
$$

together with those obtained by acting with products of $\tilde{Q}_{(n,0)}$ (where in the last equality we have used the definition of the charges and the commutation relations). There is however a crucial difference between the states in eq. (6.25) and those obtained by further action of the $\tilde{Q}_{(n,0)}$. As we shall momentarily show explicitly (for the case $q = 2$) the former class of states is associated with the action of a *local* operator on the vacuum. Stated equivalently, it is obtained by acting with an unbroken charge, $\tilde{Q}_{(n,1)}$, on a single particle state, which is itself associated with the action of a local operator. Instead the $\tilde{Q}_{(n,0)}$ that generate all the other states correspond to integrals with weight 1 over the whole volume, see (6.20), and thus map to non-local operators in the infinite volume limit. These other states are obviously associated with the degenerate vacua we already mentioned before and are therefore outside the Hilbert space at infinite volume.

We can now check that the states (6.25) not only correspond to the action of local operators but they also map, at infinite $N$, to the degenerate multi vorton states we found already in the continuum. We work out explicitly the case $q = 2$, but one can easily extend the result to arbitrary $q$. One finds

$$
|2, \mathbf{n}\rangle = \frac{1}{(2\pi r)^2} \sum_{\mathbf{m}} \cos\left(\frac{\pi}{N}\mathbf{m} \wedge \mathbf{n}\right) \Phi_{\mathbf{n}/2+\mathbf{m}}^\dagger \Phi_{\mathbf{n}/2-\mathbf{m}}^\dagger|0\rangle,
\tag{6.26}
$$

which, by taking the naive $N \to \infty$ limit and by setting the cosine to 1, matches precisely the continuum state $\int d^2\mathbf{x}\, e^{i\mathbf{P}\cdot\mathbf{x}}(\Phi^\dagger(\mathbf{x}))^2|0\rangle$ upon identifying $\mathbf{p} \equiv \mathbf{n}/r$. Notice, however, that the full expression of the wave-function at finite $N$ also includes contributions from single-particle modes with large momentum, and is thus UV sensitive. The difference with the continuum is due to the modified form of the charges.

We can continue the procedure by considering the general class of two vorton states $\Phi^\dagger_\mathbf{n}\Phi^\dagger_\mathbf{m}|0\rangle$. Given $\tilde{Q}_{(1,0)} = \Phi_\mathbf{0}$, the states $(\Phi^\dagger_\mathbf{0})^2|0\rangle$ and $\Phi^\dagger_\mathbf{0}\Phi^\dagger_{\mathbf{n}\neq\mathbf{0}}|0\rangle$ correspond to the previously encountered degenerate vacuum and single vorton state on top of a degenerate vacuum. The remaining case of $\mathbf{n}, \mathbf{m} \neq 0$, corresponds to the tensor product $\mathbf{Adj} \otimes \mathbf{Adj}$ which, taking into account the Bose symmetry of vortons, decomposes into $\mathbf{1} \oplus \mathbf{Adj} + \mathbf{R}_1 \oplus \mathbf{R}_2$, with $\mathbf{R}_{1,2}$ $SU(N)$ irreps of size $\sim N^2$ (see discussion in ref. [4]). Again the singlet is a degenerate vacuum associated with the action of $\tilde{Q}_{(2,0)}$ while the adjoint is the state in eq. (6.26). The remaining $\mathbf{R}_{1,2}$ genuinely correspond to two vorton states once the infinite volume limit is taken. The Hamiltonian is not trivially diagonalized on these states and time evolution is more effectively described in terms of an $S$-matrix. Equivalent states, and identical scattering amplitudes in the infinite volume limit, can here be obtained by acting on these states with the conserved charges $\tilde{Q}_{(q,1)}$. However a detailed description does not seem enlightening at this point.

### 6.3  Connection with the comoving coordinates

As emphasized before, the energy spectrum of the theory is determined by the algebra of the vorticity, irrespectively of its realization in terms of microscopic fields. This gives us the opportunity to relate our discussion to the comoving coordinates' description of the fluid. This is possible because the Hamiltonian's building block is the vorticity (see eq. (6.4)), and the spectrum of the theory ultimately relies on the commutation relations (5.14) and follows directly from group theory as was described in [4]. The realization of the vorticity in terms of the fields $\Phi$, $\Phi^\dagger$ makes the analysis of the spectrum simple as we detailed in the previous sec.s, but it is not necessary. A different, perhaps less intuitive, realization instead gives us the comoving fluid, which differs from the Clebsch formulation only in the degeneracy of the eigenstates.

The story was developed in [4] and roughly goes as follow. As emphasized in Part I of this work the main difference between the Clebsch and the comoving formulation is that in the latter the internal symmetry group $SDiff(\mathcal{M}_\varphi)$ (working again on a torus for simplicity) is spontaneously broken to the trivial group. When dealing with a mechanical system describing the (classical) spontaneous breaking $G \to \emptyset$ for some internal symmetry group $G$, the Peter-Weyl theorem dictates the structure of the Hilbert space of the theory.

More in detail, the comoving coordinate formulation describes a mechanical system where the fields $\varphi_i(x)$, $i = 1, 2$ describe an area preserving map of the torus $T^2$ onto itself.[33] The physical configurations of this mechanical system therefore span the $SDiff(T^2)$ manifold.

---

[33]Again, we decide to work in finite volume with periodic boundary condition for simplicity and to be able to implement the $SU(N)$ regularization.

This also means that there are two ways to act with the $SDiff(T^2)$ group, either on the physical coordinates $\mathbf{x}$ (the right action) or on the comoving coordinates $\varphi$ (the left action). While the left action is a symmetry as explained in sec. 3, the right action is not: it just spans the manifold of possible configurations – as the action (3.16) makes clear. The Hamiltonian is purely a function of the right charges, which in this case coincide with the vorticity. This is analogous to the rotating rigid body, and more in general to any mechanical system describing a coset $G \to \emptyset$ – the Lie algebra of the symmetry group $G$ (left action) is isomorphic the algebra of the right charges, i.e. the conjugated momenta in terms of which the Hamiltonian reads as a simple quadratic form.

Resorting to the $SU(N)$ regularization (6.17) of the vorticity algebra, corresponds to replacing the fluid with a mechanical model whose configuration space coincides with the $SU(N)$ group manifold. Indeed it more precisely corresponds to a system on $SU(N)/Z_N$. That is because on the $SU(N)$ group manifold (discrete) translations only commute modulo the action of the center $Z_N$ [4]. A proper realization of translations is therefore obtained by working on $SU(N)/Z_N$. In this situation, the Hilbert space takes the structure dictated by the Peter-Weyl theorem and it decomposes into the direct sum of blocks of the form $(r, r)$ where $r$ is any representation of $SU(N)/Z_N$. The latter consists of the subset of $SU(N)$ representations that can be written as tensor products of multiple adjoints. In particular the fundamental and antifundamental representations are barred. Thus each $(r, r)$ block consists of $d_r^2$ states, each with a $d_r$ degeneracy. The ground state is unique, while vorton states naturally correspond to the adjoint, and are $N^2 - 1$ times degenerate. In the continuum limit we thus recover an infinite vorton degeneracy. Notice though that the Hilbert space of the comoving fluid is not a Fock space. Moreover at finite $N$ the degeneracies of the two discretized models do not match.

Thus, while it appears in a different form, the infinite degeneracy of the vortons is a property of both the comoving and the Clebsch description, and thus seems a robust feature of the quantum perfect fluid. We will find that this remains true also in the different formulations that we will consider in sec. 7.

Note that, instead, the ground state is drastically different in the two formulations, and perhaps surprisingly, it's unique only in the comoving formulation, despite the spontaneous symmetry breaking in the classical theory. This is analogous to Coleman's mechanism preventing spontaneous symmetry breaking in relativistic two-dimensional models, and was anticipated in [1]. We also comment that the fact that the Clebsch and the comoving fluid have a different structure of the vacuum, but admit similarly degenerate vorton states in the continuum limit, is qualitatively similar to the discussion at the end of sec. 4.1 and in app. C.1. There we argue that for fluid flows with nowhere vanishing vorticity the comoving coordinates and the Clebsch fields are completely equivalent, while they are not around the static flow (even if they yield the same equations for the fluid density and velocity).

# 7 More quantum fluids

## 7.1 Fermionic vortons in 2d

The Hamiltonian of the incompressible quantum fluid is simply a function of the vorticity $\omega(\mathbf{x})$. Thus any realization of the vorticity algebra in terms of fundamental fields yields the same energy spectrum, if not the same multiplet degeneracies.

An interesting possibility suggested in [4] is to write the vorticity in terms of fermionic degrees of freedom. In this section we will consider such fermionic vorton theory. We will show in particular that the fermionic model possesses as well an infinite set of symmetries, though the charges have a more intricate structure than in the bosonic model. The $SU(N)$ regularization introduced in the previous section will play a key role in their understanding.

The Hamiltonian keeps the same structure as before

$$H = \int \frac{d^2p}{(2\pi)^2} \frac{F(\mathbf{p}^2/\Lambda^2)}{\mathbf{p}^2} \omega_{\mathbf{p}}^f \omega_{-\mathbf{p}}^f \tag{7.1}$$

where the $f$ index means that we replace the charges $\omega_{\mathbf{p}}$ with

$$\omega_{\mathbf{p}}^f = i \int \frac{d^2k}{(2\pi)^2} (\mathbf{p} \wedge \mathbf{k}) \psi_{\mathbf{p+k}}^\dagger \psi^{\mathbf{k}} \,. \tag{7.2}$$

Here $\psi^{\mathbf{k}}$ and $\psi_{\mathbf{p}}^\dagger$ are quantum fields obeying the anticommutation relations

$$[\psi^{\mathbf{k}}, \psi_{\mathbf{p}}^\dagger]_+ = (2\pi)^2 \delta^2(\mathbf{p} - \mathbf{k}) \,. \tag{7.3}$$

It is straightforward to check that the operators $\omega_{\mathbf{p}}^f$ still satisfy the $SDiff(\mathcal{M}_x)$ algebra[34]

$$[\omega_{\mathbf{p}}^f, \omega_{\mathbf{k}}^f] = i(\mathbf{p} \wedge \mathbf{k}) \omega_{\mathbf{p+k}}^f \,. \tag{7.4}$$

Note that this model cannot arise from a canonical relativistic fluid Lagrangian like those considered in part I of this work, but it nonetheless provides a nontrivial, purely quantum-mechanical, realization of the Hamiltonian of the perfect incompressible fluid.

In light of our considerations, the spectrum of the theory is unchanged. It is indeed simple to verify that single-particle states $\psi_{\mathbf{p}}^\dagger|0\rangle$ have the vorton dispersion relation (5.19).

It is less trivial to understand the symmetry of the Hamiltonian, and hence the degeneracy of the fermionic vorton states. Indeed, as in the bosonic case, canonical transformations of $\psi(x)$,

$$\psi(x) \to \psi(x) + [\psi(x), \int d^2y f(\psi(y), \psi^\dagger(y))]$$

$$\psi^\dagger(x) \to \psi^\dagger(x) + [\psi^\dagger(x), \int d^2y f(\psi(y), \psi^\dagger(y))] \tag{7.5}$$

---

[34]As emphasized in [4], both the expression $\omega_{\mathbf{p}} = i \int \frac{d^2k}{(2\pi)^2}(\mathbf{p} \wedge \mathbf{k})\phi_{\mathbf{p+k}}^\dagger \phi^{\mathbf{k}}$ and (7.2) can be seen as different Jordan-Schwinger representations of the area preserving diffeomorphism algebra, using the adjoint representation. This is made clear in the $SU(N)$ matrix model [4].

leave the charges $\omega^f(\mathbf{x})$ invariant. However, the most general function of the Grassmanian variables is $f = 1 + a\psi + a^*\psi^\dagger + b\psi^\dagger\psi$, where $a$ is Grassmanian parameter. The (super-)symmetry generated by these charges corresponds to shifts by a Grassmanian parameter and to $U(1)$ particle-number. That amounts to a finite number of parameters, in stark contrast to the infinity of the corresponding symmetry in the bosonic case.

The above result is surprising, as it would appear that the goal of constraining the action to the very specific form of vorton QFT, is achieved in the fermionic case with much less symmetry than in the bosonic case. The story is however not complete. Indeed, unlike the bosonic theory, the fermionic theory possesses symmetry transformations involving derivatives. That is more easily seen in the $SU(N)$ regulated theory, which is defined by simply replacing the vorton fields of sec. 6.1 with their fermionic counterparts. The matrix model formulation allows for a systematic construction of all the charges.

As in the bosonic case, the Hamiltonian is built out of a representation of the $U(N)$ generators by fermionic bilinears

$$\tilde{\omega}^f_{\mathbf{n}} = i \sum_{\mathbf{m}} \frac{N}{\pi} \sin\left(\frac{\pi}{N}\mathbf{n} \wedge \mathbf{m}\right) \psi^\dagger_{\mathbf{m}+\mathbf{n}}\psi^{\mathbf{m}} . \tag{7.6}$$

The field $\psi_{\mathbf{n}}$ decomposes into the $SU(N)$ single zero mode $\psi_{\mathbf{0}}$ and and $SU(N)$ adjoint $\psi_{\mathbf{n}\neq\mathbf{0}}$. As in the bosonic case, the symmetries of the Hamiltonian are obtained considering all the $SU(N)$ singlets. It is convenient to revert to the matrix notation:

$$\begin{aligned}
\Psi_{ij} &= \sum_{\mathbf{n}} (T_{\mathbf{n}})_{ij}\psi^{\mathbf{n}} \\
\Psi^\dagger_{ij} &= \sum_{\mathbf{n}} \psi^\dagger_{\mathbf{n}}(T_{-\mathbf{n}})_{ij} .
\end{aligned} \tag{7.7}$$

Under a finite $U(N)$ transformation $U = \exp\left(i\sum_{\mathbf{n}} \alpha^{\mathbf{n}}\tilde{\omega}^f_{\mathbf{n}}\right)$ generated by exponentiating the action of the discretized vorticity $\Psi$ transforms as

$$\Psi \to U\Psi U^\dagger \tag{7.8}$$

The singlets are then simply given by the trace of any product of the matrices $\Psi$ and $\Psi^\dagger$. However, due to the anticommuting nature of the fields, some of the traces vanishes. For instance, at the quadratic level we find $\mathrm{Tr}\left[\Psi^2\right] = \mathrm{Tr}\left[(\Psi^\dagger)^2\right] = 0$, and thus the $SL(2,\mathbb{R})$ symmetry present in the bosonic theory is reduced to the $U(1)$ symmetry generated by

$$Tr[\Psi^\dagger\Psi] = \psi^\dagger_{\mathbf{k}}\psi^{\mathbf{k}} , \tag{7.9}$$

as we found directly in the continuum. However traces with a higher number of fields are generically non-vanishing and may be written using the $SU(N)$ commutator[35]. At cubic order

---

[35] As in the bosonic case, the zero mode single component $\psi_{\mathbf{0}}$ is dealt with trivially, so our focus is on the adjoint component, which is also the one with implications at infinite volume.

we find

$$\mathrm{Tr}(\Psi^3) - h.c. = f_{\mathbf{nmk}}\psi^{\mathbf{n}}\psi^{\mathbf{m}}\psi^{\mathbf{k}} - h.c.,$$
$$\mathrm{Tr}(\Psi^2\Psi^\dagger) - h.c = f_{\mathbf{nmk}}\psi^{\mathbf{n}}\psi^{\mathbf{m}}\psi^\dagger_{\mathbf{k}} - h.c. \tag{7.10}$$

where $f_{\mathbf{nmk}}$ are the structure constants of $SU(N)$. In the $N \to \infty$ limit, they can be rewritten in position space as

$$\int d^2\mathbf{x}\varepsilon^{ij}\partial_i\psi(\mathbf{x})\partial_j\psi(\mathbf{x})\psi(\mathbf{x}) - h.c., \quad \text{and} \quad \int d^2\mathbf{x}\varepsilon^{ij}\partial_i\psi(\mathbf{x})\partial_j\psi(\mathbf{x})\psi^\dagger(\mathbf{x}) - h.c. \tag{7.11}$$

Contrary to the bosonic case, these tranformations involve derivatives of the fields, but they are nonetheless local.

Consider now invariants that involve only powers of $\Psi^\dagger$. In a normal ordered basis these are the only conserved charges that act non-trivially on the Fock vacuum $|0\rangle$. They control the degeneracy of the ground state and the occurrence of gapless modes. The results below should be compared to the bosonic case where there is an infinite number of such charges acting non-trivially on the vacuum.

First, using the cyclicity of the trace, it is easy to see that

$$Tr((\Psi^\dagger)^n) = (-1)^{n-1}\mathrm{Tr}((\Psi^\dagger)^n), \tag{7.12}$$

such that only traces with an odd number of fermions are non-vanishing. This is interesting as it implies that the gapless vortons should all be fermionic. Now we know that at finite $N$,

$$\mathrm{Tr}(T_{\mathbf{n}_1}...T_{\mathbf{n}_n}) = \omega^{\frac{1}{2}\sum_{i<j}\mathbf{n}_i\wedge\mathbf{n}_j}N\delta_{\sum_i\mathbf{n}_i,\mathbf{0}} \tag{7.13}$$

If we take the limit $N \to \infty$, this naively becomes $\mathrm{Tr}(T_{\mathbf{n}_1}...T_{\mathbf{n}_n}) = (1/Na^2)(2\pi)^2\delta^2(\sum_i\mathbf{k}_i)$ with $\mathbf{k}_i \equiv \mathbf{n}_i/r$. This is what we used in the bosonic vorton theory to argue that the charges (6.19) reduce to their continuum counterpart (5.7).

However, in the fermionic model, only the fully antisymmetric combination of all the permutations survives. More explicitly, we have

$$\mathrm{Tr}((\Psi^\dagger)^n) = \psi^\dagger_{-\mathbf{k}_1}...\psi^\dagger_{-\mathbf{k}_n}\mathrm{Tr}(T_{\mathbf{k}_1}...T_{\mathbf{k}_n})$$
$$\propto \psi^\dagger_{-\mathbf{k}_1}...\psi^\dagger_{-\mathbf{k}_n}\mathrm{Tr}(T_{[\mathbf{k}_1}...T_{\mathbf{k}_n]}) \tag{7.14}$$

Therefore in order to obtain a non vanishing result we must expand the prefactor $\omega^{\frac{1}{2}\sum_{i<j}\mathbf{n}_i\wedge\mathbf{n}_j}$ in (7.13) to high enough powers of momentum to produce an antisymmetric combination. Following this program, we obtain the following charges with respectively 3, 5, 7 and 9

fermions in the continuum theory:

$$
\begin{aligned}
Q_{(3,0)} &= \int d^2\mathbf{x}\,\psi^\dagger(\mathbf{x})\cdot\partial_x\psi^\dagger(\mathbf{x})\partial_y\psi^\dagger(\mathbf{x})\\
Q_{(5,0)} &= \int d^2\mathbf{x}\,\partial_x\psi^\dagger(\mathbf{x})\partial_y\psi^\dagger(\mathbf{x})\cdot\partial_x^2\psi^\dagger(\mathbf{x})\partial_x\partial_y\psi^\dagger(\mathbf{x})\partial_y^2\psi^\dagger(\mathbf{x})\\
Q_{(7,0)} &= \int d^2\mathbf{x}\,\psi^\dagger(\mathbf{x})\cdot\partial_x\psi^\dagger(\mathbf{x})\partial_y\psi^\dagger(\mathbf{x})\cdot\partial_x^3\psi^\dagger(\mathbf{x})\partial_x^2\partial_y\psi^\dagger(\mathbf{x})\partial_x\partial_y^2\psi^\dagger(\mathbf{x})\partial_y^3\psi^\dagger(\mathbf{x})\\
Q_{(9,0)} &= \int d^2\mathbf{x}\,\partial_x\psi^\dagger(\mathbf{x})\partial_y\psi^\dagger(\mathbf{x})\cdot\partial_x^2\psi^\dagger(\mathbf{x})\partial_x\partial_y\psi^\dagger(\mathbf{x})\partial_y^2\psi^\dagger(\mathbf{x})\times\\
&\qquad\qquad\qquad\times\partial_x^3\psi^\dagger(\mathbf{x})\partial_x^2\partial_y\psi^\dagger(\mathbf{x})\partial_x\partial_y^2\psi^\dagger(\mathbf{x})\partial_y^3\psi^\dagger(\mathbf{x})\,.
\end{aligned}
\tag{7.15}
$$

Note that these are all scalars under rotations due to the Grassmanian nature of the fields. Proceeding, one can find a similar symmetry charge for each odd number of fields.

We stress that, as in the bosonic model, there is at most one charge made out of $n$ creation operators, both in the $SU(N)$ regulated theory and in the continuum one. The only difference is that, for fermions, in the limit $N\to\infty$ we cannot simply replace $\omega\to1$ as we did in the previous section, since the leading term in the expansion in powers of momentum always vanishes (but for $Q_{(1,0)}$). Therefore, in the continuum bosonic vorton theory there are no symmetry charges involving derivatives and, similarly, in the fermionic continuum theory there are no additional charges made out of $3,5,7,9$ powers of $\Psi^\dagger$ and more derivatives than those given in (7.15).

Using these charges we therefore conclude that both the ground-state of the fluid at finite volume, and the vortons, are infinitely degenerate. The only significant difference compared to the bosonic Clebsch formulation is that the vorton states are now fermionic.

## 7.2 The perfect quantum fluid in 3d

Another technical advantage of the Clebsch formulation over the comoving one is that it allows to straightforwardly generalize the analysis of sec. 5 to the 3d case, as we now discuss.

Let us start from the action (4.33) in the incompressible limit. Upon canonically normalizing the fields, the action reads

$$
\mathcal{L}_\Phi = i\Phi^\dagger\dot\Phi - \frac{1}{2\rho_m}\left(\varepsilon^{ijk}\partial_j\Phi^\dagger\partial_k\Phi\right)\frac{1}{\boldsymbol{\nabla}^2}\left(\varepsilon^{ilm}\partial_l\Phi^\dagger\partial_m\Phi\right)\,.
\tag{7.16}
$$

Like in the 2d case, the action is invariant under the $SDiff(\mathcal{M}_\Phi)$ group, generated by the 3d analogue of eq. (5.5).

The analysis of the quantum theory is almost identical to the two dimensional case. The canonical commutation relations are

$$
\left[\Phi(\mathbf{x}),\Phi^\dagger(\mathbf{y})\right] = \delta^2(\mathbf{x}-\mathbf{y})\,,
\tag{7.17}
$$

so that $\Phi$ and $\Phi^\dagger$ act again as, respectively, annihilation and creation operators of a non-relativistic field theory. Introducing a regulator function as in eq. (5.11), the Hamiltonian reads

$$H = -\frac{1}{2\rho_m} \int d^3x \left( \varepsilon^{ijk} \partial_j \Phi^\dagger \partial_k \Phi \right) \frac{F\left(-\boldsymbol{\nabla}^2/\Lambda^2\right)}{-\boldsymbol{\nabla}^2} \left( \varepsilon^{ijk} \partial_j \Phi^\dagger \partial_k \Phi \right) . \tag{7.18}$$

As in sec. 5, we resolve the ordering ambiguities by demanding that the Hamiltonian is written purely in terms of the vorticity $\omega^i = i\varepsilon^{ijk}\varepsilon^{ijk}\partial_j\Phi^\dagger\partial_k\Phi$, and thus commutes with the charges (5.7). Therefore, upon normal ordering the fields, we obtain again that single-particle states have a quadratic dispersion relation

$$\omega_{\mathbf{p}} = \frac{\Lambda^3 \tilde{F}(0)}{4\rho_m} \mathbf{p}^2 , \tag{7.19}$$

where $\tilde{F}$ is defined in full analogy to eq. (5.18). Invariance under the charges (5.7) further implies that the dispersion relation (7.19) is also satisfied by infinitely many degenerate "bound-states", one for each number of vortons $n > 1$. The particles forming such bound state have completely overlapping wave-functions (in the continuum). The existence of these gapless states is associated to the spontaneous breaking of the symmetry.

Because of the equivalence of the Hamiltonians between the Clebsch and the comoving formulation, the vorton states (7.19) are the only light states of the incompressible fluid also in the comoving formulation. We did not study that case in detail, but we expect that the comoving three-dimensional fluid admits a unique ground state as in two dimensions,[36] and that the vortons are again infinitely degenerate.

## 7.3 Comparison with other works

We believe that the results of [4] and this work provide a complete assessment of the structure of the perfect quantum fluid(s). The most physically relevant prediction is the existence of an infinitely degenerate light state, the vorton. This state and its degeneracy were not predicted before. Nonetheless, the question of how to quantize the perfect fluid has appeared in a number of previous works. We compare our results to some of them below.

Some of the struggles to make sense of the perfect quantum fluid are reviewed in [25], which mostly focuses on the Clebsch variables formulation. In the comoving formulation, to the best of our knowledge, the question of how to make sense of the quantum perfect fluid was first raised in [1]. There the authors identified the origin of the peculiar dispersion relation of the transverse mode in the nonlinear action of the $SDiff(\mathcal{M}_\varphi)$ symmetry group, as reviewed in sec. 3. As a first attempt at making sense of the quantum theory, the authors introduced a symmetry breaking term giving a sound speed $c_T$ to the transverse mode, thus lifting the degeneracy implied by the classical dispersion relation $\omega_{\mathbf{k}} = 0$ and allowing for the construction of a Fock space. It was found however that several physical observables, including various cross sections, become singular in the limit $c_T \to 0$; more precisely, the

---

[36]This is motivated by the discussion in [1], where it was argued that a mechanism analogous to Coleman's theorem forbids symmetry breaking in the quantum formulation of the comoving fluid.

theory analyzed in [1] is arbitrarily strongly coupled at a scale which approaches zero in the limit $c_T \to 0$. This is unsurprising in light of the analysis of [4], that we reviewed in sec. 6.3: in the limit $c_T \to 0$ the Hilbert space is not a Fock space, but rather arranges in linear representations of the symmetry group - the vortons' being the only light states.

The question was then rivisited in [2]. There it was proposed that the comoving fluid EFT should only be used to compute correlation functions of $SDiff(\mathcal{M}_\varphi)$ invariant operators, such as the density and velocity of the fluid. The authors computed a variety of such correlators in the *naive* vacuum at one loop, finding, remarkably, that the EFT expansion remains valid in the naively expected regime and that the UV divergences are consistently renormalized via counterterms invariant under the symmetries.

We disagree with the proposal of [2]: we argued that the perfect quantum fluid admits states that are charged under the symmetry group, the vortons. The EFT can also be used to compute observables with these states in the external legs, such as their scattering amplitude as considered in [4], not just correlation functions of invariant operators. Nonetheless, the fact that the vorton states are charged under the symmetry group implies that their number is conserved, and therefore they are never created acting on the vacuum with invariant operators. In other words, due to the non-relativistic nature of the vortons it is possible to work consistently in the zero vorton sector of the theory, in which they can be ignored and the fluid's dynamics is essentially equivalent to that of a standard superfluid. We believe that is the reason why the authors of [2] did not find any issues with strong coupling nor hints of the existence of vorton states in their explicit calculations. One plausible interpretation of the calculations in [2] is that they describe correlators on the ground state of the transverse variables, where all effects of vorticity are absent and all that remains is the compressional mode. On such a state the fluid is equivalent to a superfluid.

More recently, [46] studied the two-dimensional perfect quantum fluid using the Clebsch description. However, the regime of interest of the analysis of [46] is very different from our focus: the author studied flows with large and everywhere nonvanishing vorticity $\omega$, physically describing a superfluid bucket with many pointlike vortices moving around chaotically. In that regime the transverse mode dispersion relation is non-degenerate already at the classical level, $\omega(\mathbf{k}) \neq 0$, and thus there is no conceptual subtlety in describing the quantum theory.[37] The main result of [46] is that quantum effects are responsible for a higher derivative correction of the form $\propto (\boldsymbol{\nabla} \log \omega)^2$ in the Hamiltonian (5.11) of the incompressible fluid, which may

---

[37]To make this concrete one can consider a *stationary* regime, where the time evolution of the comoving fields consists of a $SDiff(\mathcal{M}_\varphi)$ transformation as in (4.34): $\dot{\varphi}^I = f^I(\varphi)$ with $\partial_I f^I = 0$ in terms of the comoving fields. As emphasized in sec. 4.4, the fact that the unbroken time translation $H_{eff} = H + Q_f$ involves a linear combination of the generators of a non linearly realized non-abelian internal symmetry, implies that the associated Goldstone bosons have a non-vanishing gap fully controlled by the algebra [51–54]. More explicitly, considering a Goldstone fluctuation $\varphi^I + \pi^I$ with $\pi^I(x) = \xi^I(\varphi(x))$ an infinitesimal $SDiff(\mathcal{M}_\varphi)$, one has that the gap is controlled by the equation

$$\mathcal{D}_t \xi^I = \delta_t \xi^I - \delta_f \xi^I = f^J \partial_J \xi^I - \xi^J \partial_J f^I = \{f, \xi\}^I \,, \tag{7.20}$$

upon suitable diagonalization of the Lie product.

otherwise be treated classically. Note that this term indeed makes sense only for macroscopic vorticity $\omega \neq 0$.

## 8 An attempt at a physical realization: 2d positronium

We have seen that the quantum perfect fluid displays highly unusual features, including UV/IR mixing and an infinitely degenerate spectrum. While these properties make the system at hand non-trivial and interesting from a theoretical point of view, it is unclear if and how the quantum perfect fluid may be realized experimentally.

Perhaps, a less ambitious goal is to forget about the infinite degeneracy discussed in sec. 5 and just ask the following question: can we construct a system possessing asymptotic states with the same dispersion relation and scattering amplitude as the vortons?

The most obvious way to tentatively realize the vorton theory is through the dual electromagnetic description of the perfect fluid discussed in section 4.3. In that formulation the fluid at rest corresponds to a homogeneous background magnetic field $B$ (see (4.24)) while vorticity corresponds to electric charge density (see (4.31)). Furthermore, as discussed in section 5.2, the vortons here correspond to neutral particles carrying a momentum dependent electric dipole $d^i = -\epsilon^{ij}p^j/\sqrt{\rho_m}$. Intuitively, the latter properties are matched by bound states of oppositely charged particles in 2+1 QED when considering the effect of the homogeneous magnetic field. Indeed such a neutral bound state, when moving with velocity $v^i$, will experience, in its rest frame, an electric field $E^i = -\epsilon^{ij}v^jB \propto -\epsilon^{ij}p^j$. That will cause the creation of an electric dipole precisely analogous to that of the vorton.

In this section we shall make this analogy more precise by studying the simplest case, where the role of the vorton is played by a particle anti-particle bound state, the 2+1 QED analogue of positronium.

As we shall see the analogy works only up to a point. When considering scattering at low momentum we do find that the non-local part of the amplitude, associated with the long range nature of the electromagnetic interaction, nicely matches the result of the vorton theory. However this case also features hard contact terms in the amplitude, which arise from distances of the order of the Bohr radius. These effects are not present in our vorton theory. This mismatch is perhaps not so surprising given that 2+1 QED does not seem to possess the infinite symmetry of the quantum fluid. In that view one could perhaps say that our attempt here is useless. We nonetheless have considered it worth of a discussion, because it sort of concretely illustrates how special the vorton system is compared to an ordinary QFT.

### 8.1 Hamiltonian description for positronium

Let us consider two non-relativistic particles of mass $m$ and opposite electric charge $e$ in $(2+1)$-electrodynamics. As we are in two spatial dimension, the electric coupling $e$ has mass dimension $\frac{1}{2}$ and the magnetic field is a scalar. Indicating by $\mathbf{r}_1, \mathbf{r}_2$ and $\mathbf{p}_1, \mathbf{p}_2$ the particle

positions and canonical momenta, the Hamiltonian reads

$$H_{positronium} = \frac{(\mathbf{p}_1 + e\mathbf{A}(r_1))^2}{2m} + \frac{(\mathbf{p}_2 - e\mathbf{A}(r_2))^2}{2m} + \frac{e^2}{2\pi} \log \left( \frac{|\mathbf{r}_1 - \mathbf{r}_2|}{a_0} \right) . \qquad (8.1)$$

The constant $a_0$ only induces a constant shift in the energy and thus it is not physical. We can fix it to coincide with the Bohr radius $a_0 = (2me^2)^{-\frac{1}{2}}$. Choosing the symmetric gauge, the (magnetic) background gauge field is

$$A^i(\mathbf{x}) = -\frac{1}{2} B \epsilon^{ij} r^j . \qquad (8.2)$$

It is convenient to work in the center of mass frame, with $\mathbf{X} = \frac{\mathbf{r}_1 + \mathbf{r}_2}{2}$ and $\mathbf{x} = \mathbf{r}_1 - \mathbf{r}_2$, and similarly for the momenta. Because of the presence of the magnetic field, the center of mass coordinate and the relative one do not decouple, so that the Hamiltonian depends explicitly on $\mathbf{X}$. To remedy this state of affairs we make a change of canonical momenta

$$p'^i = p^i + \frac{e}{2} B \epsilon^{ij} X^j, \quad P'^i = P^i - \frac{e}{2} B \epsilon^{ij} x^j , \qquad (8.3)$$

which preserves the canonical commutation relations:

$$[x^i, p'^j] = [X^i, P'^j] = i\hbar \delta^{ij} , \qquad (8.4)$$

with all the other commutators vanishing. The Hamiltonian now reads,

$$H_{positronium} = \frac{1}{2m} \left( \mathbf{P}'^2 + 2\mathbf{p}'^2 + \frac{1}{2\pi a_0^2} \log \left( \frac{|\mathbf{x}|}{a_0} \right) + \frac{\mathbf{x}^2}{r_L^4} - 2\frac{\mathbf{x} \wedge \mathbf{P}'}{r_L^2} \right), \qquad (8.5)$$

with $r_L = (eB)^{-1/2}$ the Landau radius. The Hamiltonian does not depend anymore on $\mathbf{X}$, but the kinetic energy of the center of mass mixes $\mathbf{P}'$ and $\mathbf{x}$.

The dynamics strongly depends on the relative size of $a_0$ and $r_L$. In the regime where $a_0 \gg r_L$, magnetic effects dominate: the spectrum is approximately given by the Landau levels of the individual particles, while their mutual Coulombic attraction is just a small perturbation. In this regime there is nothing resembling a vorton state. We will thus focus on the case $a_0 \lesssim r_L$. Here the Coulomb potential gives a dominant contribution and the resulting bound states can plausibly be interpreted as vortons. Indeed that is clearly the case for $a_0 \ll r_L$, where magnetic effects can be treated as a perturbation on the lowest lying positronium states for which the wave function is localized in a region where $x \lesssim a_0$ (see eq.(8.5)). In what follows we shall assume that $a_0$ is comparable to, but sufficiently smaller than, $r_L$ for the lowest level to resemble a bound positronium. The case of a single scale $a_0 \sim r_L$ also makes for a straightforward matching with the vorton theory which is also characterized by a single length scale $1/\Lambda$.

As the Hamiltonian commutes with $\mathbf{P}'$, we can label the eigenstates by the continuous $\mathbf{P}'$ and by the discrete levels associated with the relative motion, i.e. with $\mathbf{p}'$ and $\mathbf{x}$. After having constructed such eigenstates we will be considering positronium scattering in the low

momentum regime, which corresponds to the regime of momenta below $\Lambda$ in the vorton EFT. We will thus focus on $|\mathbf{P}'| \ll r_L^{-1}, a_0^{-1}$. In this limit, the dipole term $V_d \equiv -\mathbf{x} \wedge \mathbf{P}'/(mr_L^2)$ can be treated as a small perturbation. In first approximation, neglecting $V_d$, the potential is spherically symmetric, which is best studied by going to polar coordinates $(r, \theta)$ and by expanding the wave-function in spherical harmonics

$$\psi(x, X) = e^{i\mathbf{P}' \cdot \mathbf{X}} \sum_{l,n} c_{l,n} \psi_{l,n}(\mathbf{x}), \quad \text{with } \psi_{l,n}(\mathbf{x}) = \frac{1}{a_0\sqrt{2\pi}} e^{il\theta} f_{l,n}(r/a_0). \tag{8.6}$$

with $l$ the angular momentum and $n$ the orbital quantum number. At $\mathbf{P}' = 0$ the Schrödinger equation for the angular momentum eigenstates then reads

$$\frac{1}{m}\left[ -\left( \frac{\partial^2}{\partial r^2} + \frac{1}{r}\frac{\partial}{\partial r} \right) + \frac{1}{r^2}l^2 + \frac{1}{\pi a_0^2}\ln\left( \frac{r}{a_0} \right) + \frac{1}{4}\frac{r^2}{r_L^4} \right] f_{n,l}(r/a_0) = \epsilon_{n,l} f_{n,l}(r/a_0), \tag{8.7}$$

where the eigenvalues can be written as

$$\epsilon_{n,l} = \frac{2}{ma_0^2}\hat{\epsilon}_{n,l}(y) = e^2 \hat{\epsilon}_{n,l}(y) \qquad y \equiv \frac{r_L}{a_0}, \tag{8.8}$$

with $\hat{\epsilon}_{n,l}$ dimensionless functions of $y \equiv r_L/a_0$.

Because of the logarithmic term, the Schrödinger equation can only be solved numerically. However, as our considerations will not depend on the exact values of the $\hat{\epsilon}_l$, we will spare the reader the details of the numerical analysis. The main feature of the spectrum in the regime of interest ($y \gtrsim 1$), is the $O(e^2) \sim 1/ma_0^2$ separation between the lowest levels.

Let us now consider the case of small but finite $\mathbf{P}'$. For that purpose we write the Hamiltonian as $H_{positronium} = H_0 + V_d$ and treat $V_d$ as a perturbation. In each subspace at fixed $\mathbf{P}'$, the effect of $V_d$ is the same as that of an external electric field $E_i \propto \epsilon_{ij}P'_j$ coupled to the atomic electric dipole: its lowest order effect is to mix states that differ by one unit of angular momentum. By inspecting eq. (8.5) one straightforwardly concludes that the perturbative expansion is controlled for $|\mathbf{P}'|a_0/y^2 \ll 1$. We will restrict our study to the groundstate $n = 0, l = 0$. At first order in the perturbation, the ground state becomes

$$|\psi_{\mathbf{P}'}\rangle = |\psi_{n=0,l=0}\rangle + \sum_n c_1^{(n)} |\psi_{n,l=1}\rangle + \sum_n c_{-1}^{(n)} |\psi_{n,l=-1}\rangle \tag{8.9}$$

where the coefficients $c_1^{(n)}$ and $c_{-1}^{(n)}$ are given by

$$c_{\pm 1}^{(n)} = \frac{\langle \psi_{n,\pm 1}| V_d |\psi_{0,0}\rangle}{\epsilon_{0,0} - \epsilon_{n,1}} = -\frac{1}{r_L^2(\hat{\epsilon}_{0,0} - \hat{\epsilon}_{n,1})}(P^2 \pm iP^1)\chi_{\pm 1,0}^{(n)}. \tag{8.10}$$

Here we defined

$$\chi_{\pm 1,0}^{(n)} = \int dr r^2 f_{n,\pm 1}^*\left( \frac{r}{a_0} \right) f_{0,0}\left( \frac{r}{a_0} \right) \equiv a_0^3 \hat{\chi}_{\pm 1,0}^{(n)}, \tag{8.11}$$

where $\hat{\chi}_{l,l'}^{(n)} = O(1)$ for arbitrary $y \gtrsim 1$. We checked numerically that the sum over $n$ converges, so that perturbation theory works as expected.

The energy of the ground state is only affected at second order in $V_d$, and at that order reads

$$E_{\mathbf{P}'} = e^2 \hat{\epsilon}_0(y) + c(y) \frac{\mathbf{P}'^2}{2m} + \mathcal{O}\left(\frac{a_0^2}{y^8} \frac{\mathbf{P}'^4}{m}\right), \tag{8.12}$$

where the coefficient of the kinetic energy is

$$c(y) = 1 + \frac{2}{y^4} \sum_n \frac{|\hat{\chi}_{1,0}^{(n)}|^2 \hat{\epsilon}_0}{(\hat{\epsilon}_{n,1} - \hat{\epsilon}_{0,0})^2}. \tag{8.13}$$

The dipole moment arises instead at first order and is expectedly $\propto \epsilon^{ij} P'^j$:

$$\begin{aligned}
\mathbf{d}^i &= e \langle \psi_{\mathbf{P}'} | \mathbf{x}^i | \psi_{\mathbf{P}'} \rangle \\
&= \sum_n \frac{\sqrt{2} a_0 |\hat{\chi}_{1,0}^{(n)}|^2}{\sqrt{m} \, y^2 (\hat{\epsilon}_{0,0} - \hat{\epsilon}_{n,1})} \epsilon^{ij} P'^j + \mathcal{O}\left(\frac{a_0^3}{y^6 \sqrt{m}} \mathbf{P}'^3\right).
\end{aligned} \tag{8.14}$$

We can now compare our findings to the vorton EFT. First of all we note that the dispersion relation (8.12) involves a $O(\mathbf{P}'^0)$ term, which is instead absent in the gapless result (5.19) of the vorton. However this is not a significant difference; indeed, because of the non-relativistic nature of the fluid theory in the incompressible limit, we may always perform a field redefinition $\Phi \to e^{-imt}\Phi$ in the action (4.32) and obtain a gapped dispersion relation.[38] Up to this subtlety, we obtain for small $\mathbf{P}'$ a quadratic dispersion relation like for the vorton. The dispersion relations are then matched for

$$m = 2c(y) \left(\frac{\rho_m}{\tilde{F}(0)\Lambda^2}\right) \sim \frac{\rho_m}{\Lambda^2}. \tag{8.15}$$

Similarly, eq. (8.14) matches the vorton result in eq. (5.22) provided we make the identification

$$\rho_m = \frac{my^4}{2a_0^2} \left(\sum_n \frac{|\hat{\chi}_{10}^{(n)}|^2}{(\hat{\epsilon}_{0,0} - \hat{\epsilon}_{n,1})^2}\right)^{-2} \sim \frac{my^4}{a_0^2}. \tag{8.16}$$

We stress that while in the vorton case the energy and the dipole are respectively quadratic and linear in $\mathbf{P}'$, in the case of positronium the result also involves higher powers of $\mathbf{P}'^2(a_0/y^2)^2$. The absence of these corrections in the vorton theory is due to the exact $SDiff(\mathcal{M}_\Phi)$ symmetry. Therefore, even if these corrections are small at low momentum, their presence in eqs. (8.12) and (8.14) is a first indication of the difference between the positronium system and the vorton model. On the other hand also the $SU(N)$ completion of the vorton model discussed in sec. 6 features higher derivative terms, in association with the "deformation", not the breaking, of the symmetry.

---

[38]More precisely, we obtain a gapped dispersion relation according to a different Hamiltonian which is a linear combination of $H$ and the unbroken $U(1)$ charge $Q_{(1,1)}$.

Now eqs. (8.15) and (8.16) imply that the vorton QFT cut-off is matched by

$$\Lambda^2 = \frac{c(y)y^4}{\tilde{F}(0)a_0^2} \left( \sum_n \frac{|\hat{\chi}_{10}^{(n)}|^2}{(\hat{\epsilon}_{0,0} - \hat{\epsilon}_{n,1})^2} \right)^{-2} \sim \frac{y^4}{a_0^2} \, . \tag{8.17}$$

Notice that $y^2/a_0$ also coincides with the scale of $\mathbf{P}'$ beyond which $V_d$ can no longer be treated as a small perturbation. What expectedly happens in this regime is that the Lorentz force pulls the electron and positron apart at distances larger that the Bohr radius, in such a way that the state no longer resembles a bound positronium. This is in full analogy with the behaviour of adjoint states of momentum $p \gtrsim \Lambda$ in the $SU(N)$ regulated vorton theory analyzed in detail in [4].

## 8.2   Positronium scattering

After having matched the properties of single positronium and single vorton, let us now consider $2 \rightarrow 2$ positronium scattering, focussing on the low momentum region $p \ll y^2/a_0$. Indicating the position of the electron and positron of the two positronia respectively by $(\mathbf{x}_1, \mathbf{x}_2)$ and $(\mathbf{y}_1, \mathbf{y}_2)$ the relevant Hamiltonian takes the form

$$H = H_{positronium,\mathbf{x}} + H_{positronium,\mathbf{y}} + V_{int} \tag{8.18}$$

where the first two terms consist of two copies of the two body Hamiltonian (8.5) while the interaction terms is purely due to the residual Coulomb potential. The latter reads

$$\begin{aligned} V_{int} =& \frac{e^2}{4\pi} \log \left( \frac{|\mathbf{x}_1 - \mathbf{y}_2|^2 |\mathbf{x}_2 - \mathbf{y}_1|^2}{|\mathbf{x}_1 - \mathbf{y}_1|^2 |\mathbf{x}_2 - \mathbf{y}_2|^2} \right) \\ =& \mathbf{d}_{y,i} \frac{1}{2\pi} \left( \frac{\delta_{ij}}{(\mathbf{Y} - \mathbf{X})^2} - 2\frac{(\mathbf{Y}_i - \mathbf{X}_i)(\mathbf{Y}_j - \mathbf{X}_j)}{(\mathbf{Y} - \mathbf{X})^4} - \pi\delta_{ij}\delta^2(\mathbf{X} - \mathbf{Y}) \right) \mathbf{d}_{x,j} + ... \end{aligned} \tag{8.19}$$

where in the second line, aiming at the low momentum regime, we have performed the multipole expansion at the lowest order with $\mathbf{d}_x = e(\mathbf{x}_1 - \mathbf{x}_2)$ and $\mathbf{d}_y = e(\mathbf{y}_1 - \mathbf{y}_2)$, and the positronia centers of masses are $\mathbf{X} = (\mathbf{x}_1 + \mathbf{x}_2)/2$ and $\mathbf{Y} = (\mathbf{y}_1 + \mathbf{y}_2)/2$. As $V_{int}$ decays at large distances, the above split is suited to describe asymptotic states consisting of the bound states of $H_{positronium,\mathbf{x}}$ and $H_{positronium,\mathbf{y}}$.

While the full Hamiltonian $H$ is valid for any configurations of the four charges, the splitting of the six coulomb interaction terms into $H_{positronium,\mathbf{x}/\mathbf{y}}$ and $V_{int}$ is arbitrary. It only becomes meaningful when considering scattering events where the asymptotic states are the bound states $(\mathbf{x}_1, \mathbf{x}_2)$ and $(\mathbf{y}_1, \mathbf{y}_2)$. Of course, during the scattering, recombinations can happen, so that the proper splitting of $H$ may be different for incoming and outgoing states. Keeping track of the relevant splitting, one can still employ the Lippman-Schwinger methodology and compute scattering amplitudes in the Born-approximation, with minor modifications of the procedure. The precise treatment can be found in chapter 16 of [63] for example.

For bound states of all distinguishable particles, each option for the asymptotic states is described by a unique splitting of the type in eq. (8.18). On the other hand, in the extreme

case of identical pairs of constituents with the same statistics, for which the bound states are identical bosons, the asymptotic states are obviously symmetrized linear combinations of the wave functions associated to 4 equivalent splittings. They thus take the form

$$\psi_{\mathbf{P}_A \mathbf{P}_B}(x_1, x_2, y_1, y_2) = \frac{1}{\sqrt{4}} \big( \psi_{\mathbf{P}_A}(x_1, x_2) \psi_{\mathbf{P}_B}(y_1, y_2) + \psi_{\mathbf{P}_A}(y_1, x_2) \psi_{\mathbf{P}_B}(x_1, y_2) $$
$$+ \psi_{\mathbf{P}_A}(x_1, y_2) \psi_{\mathbf{P}_B}(y_1, x_2) + \psi_{\mathbf{P}_A}(y_1, y_2) \psi_{\mathbf{P}_B}(x_1, x_2) \big) \tag{8.20}$$

where $\psi_{\mathbf{P}}(\mathbf{x}_1, \mathbf{x}_2)$ is the wavefunction for a single postronium as defined in equation 8.6.

The S-matrix $\left\langle \psi^{(-)}_{\mathbf{P}_C \mathbf{P}_D} \middle| \psi^{(+)}_{\mathbf{P}_A \mathbf{P}_B} \right\rangle$ amounts then to the sum of 16 terms, 8 with particle recombination and 8 without. The 16 terms, and each subgroup of 8 amplitudes, can also be grouped in pairs whose elements are related by a permutation of the initial, or final, momenta. These elements correspond diagrammaticaly to the t- and the u-channel. The amplitudes without recombination arise from both long- and short-range effects. On the other hand, those with recombination are purely controlled by short range effects, i.e. by the overlap of the incoming bound states over a region of the order of the Bohr radius. In view of that, their contribution to the action is expected to be local, i.e. polynomial at low momentum. Moreover these contributions cannot be dealt with in perturbation theory as recombination is a genuinely non-perturbative effect. We will thus focus on the other class of terms, whose long range part is under perturbative control, and can also stand out because of its non-polynomial behaviour.

The computation of this class of terms, as one can easily see, works with the same splitting (see eq. (8.18)) for the in and out states, which simplifies the computation. Working in the Born approximation, we get

$$(2\pi)^2 \delta^2(\mathbf{P}_A + \mathbf{P}_B - \mathbf{P}_C - \mathbf{P}_D) \mathcal{M}_{AB \to CD} \simeq \langle \psi_{\mathbf{P}_C} \psi_{\mathbf{P}_D} | V_{int} | \psi_{\mathbf{P}_A} \psi_{\mathbf{P}_B} \rangle \tag{8.21}$$

where $\mathcal{M}_{AB \to CD}$ consists of the sum of a $t$-channel

$$\mathcal{M}^t_{AB \to CD} = \frac{2a_0^2}{my^4} \left( \sum_n \frac{|\hat{\chi}_{10}^{(n)}|^2}{(\hat{\epsilon}_{0,0} - \hat{\epsilon}_{n,1})} \right)^2 \left( \frac{(\mathbf{P}_A \wedge \mathbf{P}_C)(\mathbf{P}_B \wedge \mathbf{P}_D)}{(\mathbf{P}_A - \mathbf{P}_C)^2} \right) + \dots, \tag{8.22}$$

and of a $u$-channel obtained from $\mathcal{M}^t_{AB \to CD}$ by exchanging $C \leftrightarrow D$. The dots stand for higher powers of momenta. This result matches precisely equation (5.23) with the identification (8.16), to leading order in the momentum. Notice also that while the amplitude vanishes at zero momentum, compatibly with the weakness of the interaction and hence with the applicability of the Born approximation, the individual amplitudes for the two channels are non local. Technically that means that, given that the non-perturbative and strong short distance contribution is purely local, the partial waves in each channel at arbitrarily high angular-momentum are nicely dominated by this perturbative vorton-like contribution. But, alas, summing the leading $\mathcal{M}^t_{AB \to CD}$ and $\mathcal{M}^u_{AB \to CD}$, one somewhat magically obtains a polynomial result

$$\mathcal{M}^{tot}_{AB \to CD} = \frac{a_0^2}{2my^4} \left( \sum_n \frac{|\hat{\chi}_{10}^{(n)}|^2}{(\hat{\epsilon}_{0,0} - \hat{\epsilon}_{n,1})} \right)^2 \left( (\mathbf{P}_A - \mathbf{P}_B)^2 - (\mathbf{P}_C + \mathbf{P}_D)^2 \right) + \dots. \tag{8.23}$$

This means that in the context of indistinguishable particles of the same statistics, there is no separation into long range and short range effects. Therefore, one should also include the effects from contact interactions, even at leading order. These effects come from a regime where the comparison with the vorton theory breaks down and thus the positronium is not a good realization of that theory. Notice however that if one considers distinguishable constituents or constituents consisting of one pair of identical bosons and one pair of identical fermions, in such a way that the bound states are fermions, the accidental cancellation disappears and a non-local long range contribution survives. Needless to say, this surving contribution matches perfectly the vorton result.[39]

Nevertheless, in all cases, the vorton theory and the positronium differ from each other because the latter feature contact non derivative terms. We have already argued that these control the channels with recombination, but we would like here show more directly that such contact terms arise also when going beyond the Born approximation in the channel without recombination. In this case, the elastic correction to the t-channel is given by

$$\delta \mathcal{M}_{AB \to CD} = \langle \psi_{\mathbf{P}_C} \psi_{\mathbf{P}_D} | V_{int} \frac{1}{E_{\mathbf{P}_A} + E_{\mathbf{P}_B} - H_0} V_{int} | \psi_{\mathbf{P}_A} \psi_{\mathbf{P}_B} \rangle \ . \tag{8.24}$$

We can formally compute (8.24) by inserting a complete set of states in between the two insertions of the interaction potential. This computation is represented pictorially in figure 1. Proceeding in this way, we see that in the resolution of the identity we also receive contributions from states made of two positroniums with large relative momenta, and thus we cannot resort to the expansion (8.19) to evaluate the corresponding matrix elements. The contributions to the scattering amplitude (8.24) from the exchange of these states is not suppressed by the small momentum of the external positroniums, and is thus more important than the naive leading order result (8.22). For instance, in the limit $\mathbf{P}_A = \mathbf{P}_B = \mathbf{P}_C = \mathbf{P}_D = 0$, we find that the correction to the amplitude reads:

$$\delta \mathcal{M} = \int \frac{d^2 \mathbf{P}}{(2\pi)^2} \frac{1}{2(\epsilon_0 - E_{\mathbf{P}})} \frac{16 e^4}{\mathbf{P}^4} \left| \int d\mathbf{x} \sin \left( \frac{\mathbf{P} \cdot \mathbf{x}}{2} \right) \psi_0(\mathbf{x}) \psi_{\mathbf{P}}^*(\mathbf{x}) \right|^4 \tag{8.25}$$

where $\mathbf{P}$ is the relative momentum of the intermediate state. Here $\psi_0(\mathbf{x}) = \frac{1}{a_0 \sqrt{2\pi}} f_0(|\mathbf{x}|/a_0)$ is the wavefunction for a dipole with no momenta (8.6) while $\psi_{\mathbf{P}}(\mathbf{x})$ is the wavefunction for a dipole with momenta $\mathbf{P}$ . The correction to the amplitude was computed using the exact interaction potential, as the integral is dominated by momenta of order $\frac{1}{a_0}$ such that the long distance, small momenta approximation for the dipole wavefunctions and for the interaction is not valid. Crucially, eq. (8.25) is not vanishing, thus indicating a breakdown of the perturbative expansion and the presence of a contact term in the amplitude.

---

[39]Note also that the scattering between vortons with different $U(1)$ charges corresponds to the scattering of distinguishable bosons, resulting in an amplitude that is genuinely nonlocal.

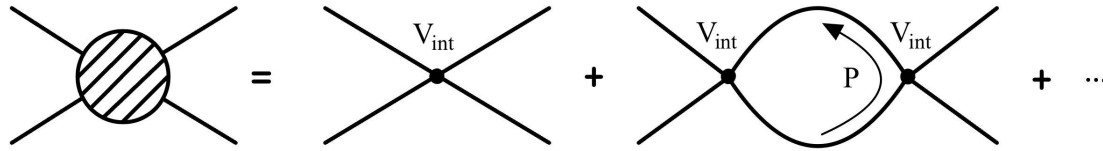

**Figure 1**: Diagrammatic representation of the scattering amplitude of two positronium in the t-channel up to second order in the Born approximation.

The estimate above shows explicitly, and somewhat expectedly, that the scattering amplitude for positronia contains contact terms, whose precise computation sits outside perturbation theory, and which start at zero powers of momentum. In an EFT description of positronium interaction these correspond to contact interaction like $(\Phi^\dagger)^2\Phi^2$, $\Phi^\dagger\nabla_i\Phi^\dagger\Phi\nabla_i\Phi$ etc., which are not present in the vorton theory. We can summarize our findings with the following non-relativistic EFT for positronium field $\Phi_p$:

$$
\begin{aligned}
\mathcal{L}_{positronium} = {}& i\Phi_p^\dagger\dot\Phi_p - \frac{c}{2m}|\boldsymbol{\nabla}\Phi_p|^2 - e^2\hat\epsilon_0|\Phi_p|^2 \\
& - \frac{\lambda}{4}|\Phi_p|^4 + \frac{1}{4\rho_m}\left(\boldsymbol{\nabla}\Phi_p^\dagger\wedge\boldsymbol{\nabla}\Phi_p\right)\frac{1}{-\boldsymbol{\nabla}^2}\left(\boldsymbol{\nabla}\Phi_p^\dagger\wedge\boldsymbol{\nabla}\Phi_p\right) + \dots,
\end{aligned}
\tag{8.26}
$$

where all interactions are taken to be normal ordered, and where the dots stand for local terms involving derivatives. As anticipated, despite the similar dipolar structure of the last interaction term, the theory of positronium does not possess the same symmetries as the vorton EFT (5.1). In particular, it includes a contact interaction $|\Phi_p|^4$, which is forbidden by the $\mathcal{M}_\Phi$ diffeomorphism symmetry in the vorton EFT.

Some comments are in order. As formerly noted, the dipole-dipole contribution to the 2-to-2 amplitude between identical scalar vortons is local, i.e. it contributes only to a finite number of partial waves. One might wonder if it is possible to isolate a purely non-local effect, that distinguishes the dipolar term from contact interactions, by looking at higher-point amplitudes. However, we note that the quartic interaction $\frac{\lambda}{4}|\Phi_p|^4$ in $2+1$ dimensions is marginally irrelevant, and therefore induces a logarithmic running of both the dipolar term and $\lambda$ itself. Such logarithmic terms are the dominant non-local contributions to all the scattering amplitudes. Therefore, there is no amplitude or partial wave whose value is dominated by the last term in eq. (8.26) at low momentum.

As we alluded before, the situation is somewhat better for fermionic positronium states (with no spin degrees of freedom). In that case, denoting with $\Psi_p$ the fermionic field, the leading contact interaction in the non-relativistic EFT involves two derivatives and is given by $(\Psi_p^\dagger\boldsymbol{\nabla}\Psi_p^\dagger)\cdot(\Psi_p\boldsymbol{\nabla}\Psi_p)$, due to the Grassmanian nature of the field. Therefore, for fermionic positronium bound states there is no marginal coupling and the dipolar interaction contributes to a non-local term in the 2-to-2 scattering amplitude. At low energies, the dipolar-interaction yields the dominant contribution to partial waves with sufficiently large angular momentum ($l>1$), that are unaffected by the contact term.

We remark that the similarity between the fermionic vorton's dispersion relation and scattering amplitude and the corresponding quantities for positronium is accidental. In particular, also in the fermionic positronium system there is no emergent symmetry that can be identified with the vorton's $SDiff(\mathcal{M}_\Psi)$ or one of its subgroups.

Of course, both in the bosonic and fermionic theory, we could cancel the leading contact interactions by including additional short-range potentials in the microscopic model (8.5), such as a Yukawa potential, and tuning their coefficients so as to cancel the quartic and higher couplings of the low-energy EFT. The vorton theory (5.11) amounts to an infinite number of tunings from this perspective.

## 9  Outlook

The results presented in ref. [4] and in this work affirmatively address the question: does the the universality class of the quantum mechanical zero temperature fluid exist? As discussed, at the theoretical level such universality class is well-defined, with its lowest energy excitations described by a novel quasi-particle, the vorton. However, the unusual features of the vorton theory make it challenging, at best, to envision how such a state could be realized in nature.

Note that the Clebsch system appears to provide a more natural framework for exploring potential realizations of the perfect quantum fluid. As discussed in sec. 8, vortons have a natural physical interpretation within this picture. Furthermore, unlike in the comoving system, the existence of low energy modes has a robust origin in the spontaneous breaking of the internal symmetry, as in more standard examples such as superfluids and solids. Yet, the symmetry group is infinite dimensional and implies the non-decoupling of UV modes, suggesting that the quantum theory described here is unlikely to be realized experimentally *as is*. Nonetheless, our analysis opens several intriguing directions for future research, which we summarize below.

First, at a conceptual level, our findings bear similarities to recent studies on fractons and exotic field theories [8, 9, 27], which also exhibit UV/IR mixing phenomena. A key distinction, however, lies in the lattice formulation of the perfect quantum fluid, that we discuss in Sec. 6, which is inherently non-local on the lattice scale. More precisely, in this formulation vorticity involves all the variables within a fixed physical distance and hence, as the continuum limit is taken, on infinitely many lattice sites. In the same section, we also present an alternative model with a local (i.e. essentially nearest neighbour) lattice construction, characterized by an $SL(2, R)$ symmetry group. This model shares some features of the vorton theory, including the presence of a bound state formed by two quasi-particles with completely overlapping wave functions. Its simpler lattice formulation may facilitate an experimental realization, warranting further detailed investigations.

Second, this work primarily focused on fluctuations around a static flow. Another interesting avenue is the study of small fluctuations around highly vortical flows. As explained in the main text (see secs. 4.4 and 7.3), this is a conceptually and technically more straightforward setup than the one analyzed here, since there are no pathological modes with trivial

dispersion. Quantum corrections therefore provide small modifications to Euler equations. Some of these correction were analyzed in [46], where it was also suggested that it might be possible to observe such quantum effects in metastable states consisting of a large number of particles confined on a plane and in the presence of a uniform magnetic field—a setup similar to that of the quantum Hall effect. It should be simple to apply some of our techniques, in particular those of sec. 5, to highly vortical flows, and thus complement and extend the analysis of [46].

Finally and most importantly, experimentally observed fluids occur at finite temperature and are dissipative; in that setup, typical observables of interest are correlation functions of the energy-momentum tensor and conserved currents, rather than energy levels. The EFT for the dissipative fluid is formulated on the Schwinger-Keldysh contour [18, 19], and includes additional couplings that make the vorticity mode diffusive; these lift the pathological dispersion relation and, as well known, lead to the diffusion pole $\omega(\mathbf{k}) \simeq -iD\mathbf{k}^2$ in correlation functions, where $D$ is the diffusion constant. It is natural to ask how our work connects with that setup. It is in particular interesting that the EFT of dissipative fluids reduces, in the dissipationless limit, to the comoving description discussed in section 3. On the other hand, we have shown in the present work that the Clebsch description is better suited for quantization. It would thus be interesting to investigate whether finite temperature dissipative hydrodynamic can also be reformulated in a Clebsch-like description. Possibly that could be the case for highly vortical flows where the comoving and Clebsch description appear equivalent in the limit where dissipation is negligible. Could some of the ideas developed here prove relevant to the study of dissipative fluids, particularly in cases with small or vanishing diffusion constants $D$?

### Acknowledgements

We thank N. Arkani-Hamed, A. Cappelli, L. Delacrétaz, S. Dubovsky, V. Gorbenko, M. Mirbabayi, A. Nicolis, R. Penco, J. Penedones, S. Sibiryakov, D.T. Son and A. Zhabin for useful discussions. During this work GC, was supported by the Simons Foundation grant 994296 (Simons Collaboration on Confinement and QCD Strings) and by the BSF grant 2018068. The work of BH was performed in part at the Kavli Institute for Theoretical Physics under Grant No. NSF PHY-1748958. EF and RR are partially supported by the Swiss National Science Foundation under contract 200020-213104 and through the National Center of Competence in Research SwissMAP. RR acknowledges the hospitality of the Perimeter Institute for Theoretical Physics, of the Center for Cosmology and Particle Physics at NYU and of the Theory Division of CERN. RR also acknowledges support from the Simons Collaboration on Confinement and QCD Strings.

# Appendix

## A    Inequivalent rigid bodies

The perfect fluid's equations of motion are usually referred to as Euler's equations. Amusingly, these share many common features with another set of equations named after Euler: those describing the motion of a rigid body at fixed center of mass position. In particular, for both of these problems it is possible to write different, inequivalent, actions that produce the same equations of motions for the velocity fields. This is a well known fact to fluid dynamicists, but it is nonetheless a somewhat unfamiliar feature for high energy theorists. In this appendix we review this fact for Euler's equations describing a rigid body.

Let us first review the most basic facts about the rigid body. A sufficiently generic rigid body breaks spontaneously the rotational group completely. The angles that parametrize its motion can be thought as the Goldstone bosons $\{\pi^a\}$ for the spontaneous breaking of the $SO(3)$ group. Therefore, to write an action, we introduce an arbitrary parametrization of the coset $SO(3)/1$:

$$U = e^{i\pi^a T_a}\,, \tag{A.1}$$

where $T_{a=1,2,3}$ are the generators of $SO(3)$. The standard Euler angles correspond to a different choice of the parametrization $U$.

As well know, we can write $SU(2)$ invariant Lagrangians in terms of the components of the angular velocity $\boldsymbol{\Omega} = (\Omega_1, \Omega_2, \Omega_3)$, obtained from the matrix $U$ as:

$$\Omega_a = -i\mathrm{Tr}\left[U^{-1}T_a\dot{U}\right]\,. \tag{A.2}$$

In particular, the most general $SU(2)$ invariant action for the $\pi^a$'s to second order in derivatives and invariant under time reversal is

$$L = \frac{1}{2}\boldsymbol{\Omega} \cdot \mathbf{I} \cdot \boldsymbol{\Omega}\,, \tag{A.3}$$

where $\mathbf{I} = \mathrm{diag}\left(I_{11}, I_{22}, I_{33}\right)$ is the inertia tensor. From the variation of the action (A.3) we derive Euler's equations of motion (EOMs):

$$\mathbf{I} \cdot \dot{\boldsymbol{\Omega}} + \boldsymbol{\Omega} \wedge (\mathbf{I} \cdot \boldsymbol{\Omega}) = 0\,. \tag{A.4}$$

The components of the angular momentum are constant in time and read

$$J_a = U_{ab}^{-1} I_{bc}\Omega_c\,. \tag{A.5}$$

It is also simple to derive the Hamiltonian, see e.g. [4]:

$$H = \frac{1}{2}\mathbf{R} \cdot \mathbf{I}^{-1} \cdot \mathbf{R} \quad \text{where} \quad \mathbf{R} = \mathbf{I} \cdot \boldsymbol{\Omega}\,. \tag{A.6}$$

We will refer to the $\mathbf{R}$ as right momenta; note that they are distinguished from the angular momentum, $J_a = U_{ab}^{-1}R_b$. The Poisson brackets of both the $R_a$ and the $J_a$ admit a $SU(2)$-group structure

$$\{R_a, R_b\} = \varepsilon_{abc}R_c\,, \qquad \{J_a, J_b\} = \varepsilon_{abc}J_c \tag{A.7}$$

from which the EOMs (A.4) follow using $\dot{\mathbf{R}} = \{H, \mathbf{R}\}$. Note that the conservation of the angular momentum Casimir $J^2 = \mathbf{R} \cdot \mathbf{R} = \text{const.}$ follows immediately from the $SU(2)$ group structure of eq. (A.7). The conservation of the angular momentum instead follows from the Poisson brackets:

$$\{R_a, J_b\} = 0 \quad \Longrightarrow \quad \{H, J_a\} = 0\,, \tag{A.8}$$

so that the $J_a$'a generate the $SU(2)$ internal symmetry.

The feature of the rigid body we are interested in is that its EOMs (A.4), as well as the Hamiltonian (A.6), are written purely in terms of the angular velocity without any reference to the internal angles of the solid. In other words, we may disregard the matrix $U$ in eq. (A.1) and simply solve for the motion of the velocities $\boldsymbol{\Omega}$, regarded now as *fundamental* variables, rather than defined by eq. (A.2). However, note that if we disregard the angles, we cannot construct the components of the angular momentum (A.5), but only the Casimir $J^2$. In other words, we may consider a system characterized just by the angular velocities, but in which there is no analogue of the angles $\pi^a$. Nevertheless, such a system is governed by the equations of motion for a rigid body, as follows from the Poisson brackets (A.7) acting on the Hamiltonian (A.6).

We now ask then: can we write an action for such a system? In other words, can we write a self-consistent Lagrangian for the velocities $\boldsymbol{\Omega}$ whose extremization results in eq. (A.4), without any reference to the angles (A.1) or any other additional variables?[40] Interestingly, the answer is no. This is because the EOMs are first-order in derivatives and therefore require the $\Omega_a$ to form conjugate pairs, but this is impossible for three variables. Equivalently, the Poisson brackets in eq. (A.7) cannot be inverted due to the existence of a conserved Casimir $J^2 = \mathbf{R} \cdot \mathbf{R} = \text{const.}$—see the discussion at the end of this section for further details on the role of Casimirs.

The need of introducing coordinates also has a physical interpretation. Indeed, as we saw above, the $R$-charges commute with the Casimir $J^2 = \mathbf{R} \cdot \mathbf{R}$. This means that, if we work purely in terms of $R_a$'s, we cannot couple the system to external agents that change $J^2$. In other words, we need some sort of internal coordinates to inject angular momentum into the system.

The absence of any reference to the angles in the EOMs and the Hamiltonian however makes it possible to write other actions, in terms of different fundamental fields, that lead to the same EOMs (A.4) for the angular velocity. To see this, let us introduce complex fields

---

[40]Here we refer to standard unconstrained extremization. It is possible to consider modified principles that do not require introducing additional fields [44]. We will not discuss these here, since their significance once we consider a quantum theory is unclear to us.

$\{\phi_A, \phi_A^\dagger\}$ in an arbitrary representation $r$ of the $SU(2)$ group. Then we can consider the Lagrangian:

$$L = i\phi_A^\dagger \dot{\phi}_A - \frac{1}{2}\left(\phi^\dagger T_a \phi\right) I_{ab}^{-1} \left(\phi^\dagger T_b \phi\right), \tag{A.9}$$

where $T_{a=1,2,3} = \{(T_a)_{AB}, \ A, B = 1, \ldots, 2r+1\}$ form a representation of the $SU(2)$ group in the $r$-representation. From this action we immediately derive the Poisson brackets $\{\phi_A, \phi_B^\dagger\} = \delta_{AB}$ and the Hamiltonian (A.6) with the identification

$$R_a = \phi^\dagger T_a \phi. \tag{A.10}$$

The $SU(2)$ structure of the Poisson brackets (A.7) and the EOMs (A.4) then follow.

To illustrate in detail the differences with the Euler's angles, consider the symmetry of the system (A.9) for fields in the adjoint representation. This is equivalent to taking $\phi = \phi_a T_a$ to be a traceless complex hermitian $3 \times 3$ matrix. In this case the right momenta read

$$R_a = \mathrm{Tr}\left(\phi^\dagger \left[T_a, \phi\right]\right) = \mathrm{Tr}\left(T_a \left[\phi, \phi^\dagger\right]\right). \tag{A.11}$$

It is then easy to see that the action (A.9) is invariant under $SL(2,R)$ transformations acting on the fields as

$$\begin{pmatrix} \mathrm{Re}\phi \\ \mathrm{Im}\phi \end{pmatrix} \quad \rightarrow \quad \begin{pmatrix} a & b \\ c & d \end{pmatrix} \begin{pmatrix} \mathrm{Re}\phi \\ \mathrm{Im}\phi \end{pmatrix} \quad \text{with} \quad ad - bc = 1. \tag{A.12}$$

In fact, eq. (A.9) is the most general Lagrangian to quartic order in the fields which is compatible with the internal $SL(2,R)$ symmetry, as well as time reversal, which acts as $\phi \leftrightarrow \phi^\dagger$ to preserve the kinetic term. This forbids terms linear in the $R_a$'s. More generally, the symmetry (A.12) is compatible with any action written in terms of the $R_a$'s plus the kinetic term. Therefore the $SL(2,R)$ symmetry (A.12) is as constraining as the $SU(2)$ symmetry of the standard rigid body. Note also that the $SL(2,R)$ Casimir coincides with the $SU(2)$ one in the Euler's angle description $\mathbf{R} \cdot \mathbf{R}$.[41]

We also remark that in the system (A.9) all the configurations such that $[\phi, \phi^\dagger] = 0$ are degenerate and have zero energy. Unlike the standard rigid body, for which all configurations at rest are related by rotational symmetry, the classical *vacua* of the system (A.9) are not all obtained from the same state via the action of the $SL(2,R)$ symmetry group, that acts linearly on the fields. Instead, this degeneracy is a dynamical consequence of the structure of the action (which is *indirectly* a consequence of the symmetry).

We finally reiterate that none of the two Lagrangian descriptions that we introduced is equivalent to the set of EOMs (A.4). Indeed, the eqs (A.4) require only three initial conditions to be solved, while the system (A.3) admits three canonical pairs, and (A.9) has

---

[41]This is generically true for any real $SU(2)$ representation, *i.e.* $r$ integer: there is an $SL(2,\mathbb{R}) \times O(2r+1) \subset Sp(4r+2,\mathbb{R})$ dual pair structure, where the representations of $SL(2,\mathbb{R})$ appearing in the Hilbert space are determined by those of $O(2r+1)$, and vice-versa. As a consequence, the Casimirs can be written in terms of one another.

$2r + 1$ canonical pairs, where $r$ is the spin of the representation of the internal variables. The statement is simply that the dynamics of the angular velocities, which is a subset of the full dynamics in both cases, is specified by the EOMs (A.4).

In summary, even if the EOMs (A.4) are written just in terms of the angular velocities, there is no way to associate them with an action purely written in terms of the $\Omega_a$'s. Rather, if we insist on having a Lagrangian description, we obtain inequivalent systems characterized by different internal variables, that are needed to reproduce the symplectic structure (A.7). In the standard rigid body formulation (A.3), the variables are just the angles and are needed to realize nonlinearly the $SU(2)$ symmetry. In the alternative formulation (A.7), the symmetry group is non-compact and linearly realized. In both cases, the symmetry group does not act on the velocities, but ensures the conservation of the Casimir $\mathbf{R} \cdot \mathbf{R}$.

Let us close this section by giving some more technical details. In doing so, we hope to expose some of the fundamental group structure underlying the rigid body and incompressible fluid. As detailed above, the rigid body dynamics are entirely captured by the dynamics of the charges $\boldsymbol{R}$. These parameterize the *phase space*, isomorphic to $\mathbb{R}^3$. The dynamics follow upon (1) picking Poisson brackets and (2) choosing a Hamiltonian. The fundamental Poisson brackets are taken to be governed by the $SU(2)$ algebra, $\{R_a, R_b\} = f_{ab}{}^c R_c = \epsilon_{abc} R_c$. For general functions $A = A(R)$ and $B = B(R)$ on phase space, the brackets take the form

$$\big\{ A(R), B(R) \big\} = R_c \, f_{ab}{}^c \, \frac{\partial A}{\partial R_a} \frac{\partial B}{\partial R_b} \equiv \left\langle R, \left[ \frac{\partial A}{\partial R}, \frac{\partial B}{\partial R} \right] \right\rangle, \tag{A.13}$$

where $[ \ , \ ]$ is the commutator on $\mathfrak{g} = \mathfrak{su}(2)$. Brackets like this, determined by the structure constants of an underlying Lie algebra $\mathfrak{g}$, are known as Lie-Poisson brackets, *e.g.* [64, 65]. The incompressible fluid flow is another Lie-Poisson system, goverened by the group of volume preserving diffeomorphisms $G = SDiff(M)$ [64, 66].

For a Lie-Poisson system there is generally an obstruction to inverting the Poisson brackets to obtain the symplectic form. This is due to Casimirs $C_k$: these (Poisson) commute with the phase space coordinates, $\{C_k, \mathbf{R}\} = 0$, and therefore with *any* function on phase space. As such, they give a zero mode and render the brackets $\{ \ , \ \}$ degenerate. This means that dynamics, independent of the Hamiltonian, will be constrained to surfaces of constant Casimir values. For the rigid body, we have the $SU(2)$ Casimir $J^2 = \mathbf{R}^2$, which foliates the phase space $\mathbb{R}^3$ into spheres upon which dynamics takes place. Note that the sphere is two-, namely even-, dimensional: on these surfaces the brackets are non-degenerate, giving a well-defined symplectic form. For example, in standard spherical coordinates, we can take $\{J \cos\theta, \phi\} = 1$.[42] As long as we only interested in the dynamics at a fixed value of $\mathbf{R}^2$ =const., we may indeed write an action in terms of $\theta$ and $\phi$ only—this is the well known $SU(2)/U(1)$ coadjoint orbit, describing the spin of non-relativistic particles and two-dimensional charged particles in a uniform magnetic field in the lowest Landau level, see e.g. [67–69] for details.

By viewing the $R_a$ as composite objects, we can construct theories with an enlarged phase space that give rise to an effective description of the $R_a$ alone. The process of going

---

[42]This description is valid away from the poles; for those, we need other coordinate patches.

from a "microscopic" description to an effective description, involving the $\mathbf{R}$ alone, is called a *phase space reduction*. For example, the Euler angle description of the rigid body, with its six-dimensional phase space $T^*SU(2)$,[43] gets reduced to the effective phase space $\mathbb{R}^3$ parameterized by the $R_a$.[44] The opposite procedure, of constructing an enlarged phase space which contains the effective space of interest, is sometimes called "inflating" the phase space [65], in contrast to the process of reduction.

## B  Microscopic derivations of fluid Lagrangians

### B.1  Derivation of the comoving action from point-like particles

In this section we review the derivation of [25] of the comoving coordinates' action for fluids in the non-relativistic limit. We consider $N \gg 1$ Galilean particles, whose action reads

$$S = \int dt \left[ \sum_{k=1}^{N} \frac{m}{2} \dot{\mathbf{X}}_k^2 - V(\{\mathbf{X}_k\}) \right] . \tag{B.1}$$

Here $k = 1, 2, \ldots, N$ is the particles' label and $V = V(\{\vec{X}_k\})$ is the potential. We assume that all the particles interact with each other in the same way, and thus $V$ is invariant under reshuffling of the indices $k = 1, \ldots, N$. The hydrodynamic limit corresponds to $N \to \infty$ limit, so that the particles form a space-filling continuum. We may therefore promote the index $k$ to a $d$ continuous variables $\vec{\varphi}$: $\mathbf{X} = \mathbf{X}(t, \vec{\varphi})$. Importantly, we assume that the map $\mathbf{X} \leftrightarrow \vec{\varphi}$ is invertible. This means there exists a function $\vec{\chi}(t, \vec{r})$ such that

$$\mathbf{X}(t, \vec{\chi}(t, \mathbf{r})) = \mathbf{r}, \qquad \vec{\chi}(t, \mathbf{X}(t, \vec{\varphi})) = \vec{\varphi}. \tag{B.2}$$

In the continuum limit, invariance under reshuffling of the particles is equivalent to requirement that the potential $V$ remains unchanged under the action of volume preserving diffeomorphisms

$$\vec{\phi} \to \vec{f}(\vec{\phi}) \quad \text{s.t.} \quad \det\left( \frac{\partial \vec{f}}{\partial \vec{\phi}} \right) = 1. \tag{B.3}$$

Assuming locality, this implies that to leading order in derivatives we must have

$$V = V(\det W), \qquad W_j^I = \left. \frac{\partial \varphi^I}{\partial X^j} \right|_{\vec{\varphi} = \vec{\chi}(t, \vec{X})} . \tag{B.4}$$

---

[43]This is the usual cotangent bundle formulation $T^*\mathcal{C}$, making use of canonical $q$'s and $p$'s, where the configuration space is identified with the rotation group, $\mathcal{C} = G = SU(2)$. Another realization, as explained above, is to construct Clebsch-like variables for the $R_a$, which essentially corresponds to choosing the spin-1, adjoint representation for $R_a = \phi^\dagger T_a \phi$. For general group $G$ we could take $R_a = f_{ab}{}^c p_c q^b = p_c (T_a^{\text{adj}})^c{}_b q^b$ (or use $(\phi, \phi^\dagger)$ in place of $(q, p)$).

[44]The effective phase space is the dual of the Lie algebra, $\mathfrak{g}^* = \mathfrak{so}^*(3) \simeq \mathbb{R}^3$. The spheres of constant Casimir discussed above are the coadjoint orbits under the $G$-action on $\mathfrak{g}^*$.

Therefore, in the hydrodynamic limit the action (B.1) obviously is

$$S = \int dt d^d \varphi \left[ \frac{m}{2} \dot{\mathbf{X}}^2 - V(\det W) \right] . \tag{B.5}$$

Switching to the Lagrangian formulation, eq. (B.5) becomes

$$S = \int dt d^2 x \left[ \frac{m}{2} \frac{v^i v^i}{v^0} - v^0 V(v^0) \right] . \tag{B.6}$$

The equivalence between eq. (B.5) and (B.6) can be seen inverting the map between $\vec{\phi}$ and $\vec{X}$ using

$$\varphi^I(t, \mathbf{X}(t, \vec{\varphi})) = \text{const.} \quad \Longrightarrow \quad \dot{\varphi}^I = -\partial_i \varphi^I \dot{X}^i = -W_i^I \dot{X}^i . \tag{B.7}$$

It is then simple to check that eq. (B.6) corresponds to the non-relativistic limit of eq. (3.3) using $F = -\epsilon$ and recalling eq. (2.21) (discarding a total derivative).

## B.2  Derivation of the Clebsch action from point-like vortices

A derivation similar to the one in the previous section can be used to derive the Clebsch action (4.13) from the classical dynamics of a superfluid in the presence of a large number of point particle vortices in two dimensions. This derivation is inspired by [35, 46].

As well known, the dynamics of vortices in a superfluid is equivalent to that of charge 1 particles in the Lowest Landau Level, interacting with an electromagnetic field [70, 71]. To lowest order in derivatives, the action therefore reads

$$S = - \int d^3 x \, G(F_{\mu\nu}) - \sum_p \int dt \left[ A_i \dot{X}_p^i + A_0(\mathbf{X}_p) \right] , \tag{B.8}$$

where $G$ is an arbitrary function (constrained by the spacetime symmetry) and the superfluid density is equivalent to a large background magnetic field

$$\langle A_i \rangle = -\varepsilon_{ij} x^j B / 2 \quad \Longrightarrow \quad \langle F_{ij} \rangle = \varepsilon_{ij} B , \tag{B.9}$$

where we work in Landau gauge.

Similarly to the previous section, we suppose that the vortices form a space-filling continuum and promote the label index to a continuum variable: $p \to \vec{\phi}$. We thus obtain

$$S = - \int d^3 x \, G(F_{\mu\nu}) - \int dt d^2 \phi \left[ A_i(t, \mathbf{X}) \dot{X}^i + A_0(t, \mathbf{X}) \right] , \tag{B.10}$$

where $\mathbf{X} = \mathbf{X}(t, \vec{\phi})$. The key assumption is that the map $\mathbf{X}(t, \vec{\phi})$ can be inverted. This means that the vorticity vector $\omega^\mu$ is nowhere singular. Under this assumption, we have

$$d^2 \phi = d^2 x \left| \frac{\partial \phi}{\partial X} \right| = d^2 x \, \varepsilon^{0ij} \partial_i \phi^A \partial_j \phi^B \varepsilon_{AB} / 2 , \qquad \dot{X}^i = -\frac{\partial X^i}{\partial \phi^K} \dot{\phi}^K . \tag{B.11}$$

Using these, after some algebra we arrive at

$$S = -\int d^3x \left[ G(F_{\mu\nu}) + A_\mu \varepsilon^{\mu\nu\rho} \partial_\nu \phi^A \partial_\rho \phi^B \varepsilon_{AB}/2 \right] , \tag{B.12}$$

which is just the action (4.24) with the identification $G = -F$, $\phi^1 = \sqrt{2}\text{Re}\,\Phi$ and $\phi^2 = \sqrt{2}\text{Im}\,\Phi$. This shows that the Clebsch formulation of the perfect fluid emerges in the description of configurations with macroscopic vorticity, such that the vorticity volume form $\omega = id\Phi \wedge d\Phi^\dagger$ is nowhere singular and thus the map between spatial coordinates and vortex labels $\phi^A$ is invertible.

Let us finally comment that it is straightforward to extend this derivation to account for higher derivative corrections to the microscopic vortex action (B.8). A particularly relevant case concerns with the contribution of the vortex fugacities, the zero-point energy of the vortices, which diverges logarithmically with the vortex size $a$ as $\sim \sum_p B \log a^{-1}$, and results in a contribution $\propto \omega \log(\omega a^2)$ in the fluid action (B.12) - see [35] for a detailed discussion.

## C  More on the Clebsch formulation of the perfect fluid

### C.1  Comoving and Clebsch fields in two dimensions

In sec. 4.1 we proved that the Clebsch and the comoving description describe, to arbitrary order in the derivative expansion, the same hydrodynamic flows. Yet, as in the case of the rigid body reviewed in sec. A, the two systems are not equivalent when their full dynamics is taken into account. This is true even if the systems (3.3) and (4.13) have the same number of degrees of freedom, that is the number of canonical pairs, in two dimensions.

The subtlety lies in the fact that the Clebsch parametrization is sometimes highly redundant, and thus for each configuration $\{\varphi^1, \varphi^2\}$, there may be more than one configuration of the fields $\{\chi, \Phi, \Phi^\dagger\}$ that yields the same velocity profile. To see this, it is enough to consider the solution corresponding to the static flow in both descriptions. In the Clebsch formulation this is given by (4.21) and is vastly degenerate. As we have emphasized in sec. 4.2, this degeneracy is not due to a symmetry, rather it follows from the structure of the action. The static flow is instead described by $\varphi^I = \alpha x^I$ up to the action of the symmetry group in the comoving description. This situation is to be contrasted with the standard duality between a scalar and a gauge field. There the scalar profile completely specifies the gauge field up to gauge transformations, and viceversa the electromagnetic field specifies the scalar up to the action of the shift symmetry.

There is however a large class of flows for which the map between the Clebsch and the comoving fields is invertible up to the action of the respective symmetry groups. Those are the flows for which the vorticity volume form

$$\omega = id\Phi \wedge d\Phi^\dagger \neq 0 \tag{C.1}$$

is non-degenerate. When $\omega$ is non-degenerate, we can construct an invertible map $\mathbf{x}(t, \Phi, \Phi^\dagger)$, representing the trajectory of the vortices; this map is unambiguous up to the action of the

symmetry (4.20). At the same time, for all physical flows there exists an invertible map $\mathbf{x}(t, \varphi^I)$ in terms of the comoving coordinates, specified by $S^\mu = \frac{1}{2}\varepsilon_{IJ}\varepsilon^{\mu\nu\rho}\partial_\nu\varphi^I\partial_\rho\varphi^J$ up to the area-preserving diff. symmetry. Combining the two we obtain an invertible map $\Phi = \Phi(t, \varphi^I)$ between $\Phi$, $\Phi^\dagger$ and the comoving fields, modulo the action of the symmetries. When such a map exists, it is possible to map all the observables from one formulation to the other following the derivation in sec. 4.1.

To appreciate this map in greater detail, it is useful to note that both the comoving and the Clebsch descriptions admit two infinite sets of conserved currents.

Let us discuss first those that follow from the comoving Lagrangian (3.3). The first infinite set corresponds to the Noether currents associated with the $SDiff(\mathcal{M}_\varphi)$ symmetry (3.1). These are derived considering an infinitesimal transformations of the dynamical fields $\delta\varphi^I \propto \varepsilon^{IJ}\partial_J f$ for an arbitrary function $f(\varphi^I)$, and read

$$J_f^\mu = \varepsilon^{\mu\nu\rho}2F'v_\nu\partial_\rho\varphi^I\partial_I f(\varphi)\,, \tag{C.2}$$

Note that these currents cannot be written purely in terms of the density and velocity of the fluid, but explicitly depend on the microscopic fields due to the non-Abelian structure of the symmetry group. The other infinite set of currents corresponds instead to

$$j_f^\mu = v^\mu f(\varphi) = \frac{1}{2}\varepsilon_{IJ}\varepsilon^{\mu\nu\rho}\partial_\nu\varphi^I\partial_\rho\varphi^J f(\varphi)\,, \tag{C.3}$$

for any function $f$ of the comoving currents. These currents are topoligcally conserved as an immediate consequence of (3.4) and (3.7). For $f = 1$, eq. (C.3) reduces to the entropy current. The currents (C.3) do not admit any obvious physical interpretation at this stage.

In the Clebsch formulation we also have two similar sets of currents. First, the Noether currents associated with the area-preserving diff. transformations (4.20) are given by

$$j_f^\mu = 2P'(\xi^2)\xi^\mu f(\Phi, \Phi^\dagger) = S^\mu f(\Phi, \Phi^\dagger)\,. \tag{C.4}$$

for any arbitrary real function $f$. In addition to the Noether currents, the theory (4.13) also admits a set of trivially conserved topological currents

$$J_f^\mu = \varepsilon^{\mu\nu\rho}\xi_\nu\partial_\rho f(\Phi, \Phi^\dagger)\,. \tag{C.5}$$

Conservation of these follows from the trivial identity $\varepsilon^{\mu\nu\rho}\partial_\mu\xi_\nu\partial_\rho\Phi = \varepsilon^{\mu\nu\rho}\partial_\mu\xi_\nu\partial_\rho\Phi^\dagger = 0$.

Using the correspondence $Tu^\mu = 2F'v^\mu = \xi^\mu$ and $S^\mu = v^\mu = 2P'\xi^\mu$ and the existence of an invertible map $\varphi^I(\Phi, \Phi^\dagger)$ for flows such that eq. (C.1) holds, we immediately see that the comoving Noether currents (C.2) map to the topological ones (C.5) in the Clebsch formulation. Viceversa, the Clebsch Noether currents (C.4) become the trivially conserved vectors (C.3) in the comoving description.

In the tab. 1 we summarize the comparison between the comoving and the Clebsch formulations of the two-dimensional fluid. The first entries in tab. 1, that do not refer to the currents discussed above, are true also in three spatial dimensions.

| | Clebsch | Comoving |
|---|---|---|
| Lagrangian | $P(\xi^2)$ | $F(v^2)$ |
| fundamental fields | $\chi, \Phi, \Phi^\dagger$ | $\varphi_1, \varphi_2$ |
| vorticity $\omega^\mu = \epsilon^{\mu\nu\rho}\partial_\nu\xi_\rho$, with | $\xi^\mu = \partial_\mu\chi + \frac{i}{2}(\Phi^\dagger\partial_\mu\Phi - \Phi\partial_\mu\Phi^\dagger)$ | $\xi^\mu = 2F'v^\mu$ |
| entropy current | $v^\mu = 2P'\xi^\mu$ | $v^\mu = \frac{1}{2}\varepsilon_{IJ}\varepsilon^{\mu\nu\rho}\partial_\nu\varphi^I\partial_\rho\varphi^J$ |
| energy density | $\rho = 2\xi^2 P'(\xi^2) - P(\xi^2)$ | $\rho = -F(v^2)$ |
| pressure | $p = P(\xi^2)$ | $p = F(v^2) - 2F'(v^2)v^2$ |
| symmetry | $SDiff(\mathcal{M}_\Phi)$ | $SDiff(\mathcal{M}_\varphi)$ |
| Noether Currents | $j_f^\mu = v^\mu f(\Phi, \Phi^\dagger)$ | $J_f^\mu = \varepsilon^{\mu\nu\rho}\xi_\nu\partial_\rho\varphi^I\partial_I f(\varphi)$ |
| Topological Currents | $J_f^\mu = \varepsilon^{\mu\nu\rho}\xi_\nu\partial_\rho f(\Phi, \Phi^\dagger)$ | $j_f^\mu = v^\mu f(\varphi)$ |
| Solution for fluid at rest | $\chi = \mu t, \quad \Phi^\dagger = \Phi = 0$ | $\varphi^I = \alpha\mathbf{x}^I$ |
| spontaneously breaks | shifts: $\chi \to \chi + c_\chi, \Phi \to \Phi + c_\Phi$ | $SDiff(\mathcal{M}_\varphi)$ |

**Table 1**: Comparison of the two theories describing a perfect fluid. Both theories realize Euler's equations as the conservation of the stress-energy tensor. First, all hydrodynamical quantities are expressed in terms of the variables of the theories. Second, the dynamically and topologically conserved quantities are displayed in both cases. Finally the classical solution for a fluid at rest is displayed in both cases. The solution is not invariant under the full symmetry group of each theory, however the breaking is significantly different in each case. While the solution only breaks shifts of the fields in the Clebsch description, the full symmetry group is broken in the comoving description.

As we have seen in sec. 3.2, the conservation of the Noether currents (C.2) of the comoving Lagrangian is associated with Kelvin's theorem. A similar remark holds for the topological currents (C.5) of the Clebsch description, provided we work in a background such that $\omega^\mu$ is nowhere vanishing. We instead found that the conservation of the currents (C.3) and (C.4) do not lead to novel interesting conservation laws for the hydrodynamic variables.[45]

We finally comment that, while the considerations of this section are mathematically

---

[45]Consider for instance the charges associated to the $SDiff(\mathcal{M}_\Phi)$ symmetry of the Clebsch fields. Working in the gauge description introduced in section 4.3, these read

$$Q_f = \int d^2x\, v^0 f(\Phi, \Phi^\dagger), \tag{C.6}$$

for an arbitrary function $f$ of the vorton fields, where the entropy current is $v^\mu = \frac{1}{2\pi}\varepsilon^{\mu\nu\rho}\partial_\nu A_\rho$. Choosing a basis $f(\Phi, \Phi^\dagger) = \delta(\Phi(\mathbf{x}) - \Phi_0)$, the charges read

$$Q_\Phi^0 = \int d^2x v^0\delta(\Phi(\mathbf{x}) - \Phi_0) = v^0 \left|\frac{\partial\Phi}{\partial x}\right|^{-1}_{\Phi=\Phi_0}, \tag{C.7}$$

where here we have importantly assumed that the map $\mathbf{x} \to \{\Phi^\dagger(\mathbf{x}), \Phi(\mathbf{x})\}$ is invertible. Integrating eq. (C.7)

amusing, it is unclear to us if they have any physical meaning. First, as remarked above, the existence of a map between the two fluid formulations that we discussed appears to be an accident of two dimensions. Second, the relation discussed above becomes singular when working around the static flow, which is the starting point of our analysis of the quantum fluid. As we argue in Part II of this work, the two Lagrangian formulations of the fluid will lead to two inequivalent quantum fluids, with the same Hamiltonian spectrum but different Hilbert spaces (i.e. different degeneracies of the Hamiltonian eigenstates).

## C.2 The incompressible limit in three spatial dimensions

In this section we derive the incompressible limit of the three dimensional fluid. To this aim, we proceed similarly to the two-dimensional case analyzed in sec. 4.3 and dualize the scalar $\chi$ first in terms of a 2-form gauge field $A_{\mu\nu}$. This is done considering a Lagrange multiplier in the form

$$\tilde{\mathcal{L}}_\xi = P(\xi^\mu \xi_\mu) - \frac{1}{4\pi}\left[\xi_\mu - \frac{i}{2}(\Phi^\dagger \partial_\mu \Phi - \Phi \partial_\mu \Phi^\dagger)\right]\varepsilon^{\mu\nu\rho\sigma}\partial_\nu A_{\rho\sigma}, . \tag{C.9}$$

Integrating out $A_{\mu\nu}$ sets $\xi^\mu - \frac{i}{2}(\Phi^\dagger \partial_\mu \Phi - \Phi \partial_\mu \Phi^\dagger) = \partial_\mu \chi$ as before. We instead integrate out $\xi^\mu$ and, discarding a total derivative, we obtain

$$\mathcal{L}_A = F\left(v_\mu v^\mu\right) + \frac{i}{4\pi}A_{\mu\nu}\varepsilon^{\mu\nu\rho\sigma}\partial_\rho \Phi^\dagger \partial_\sigma \Phi, \qquad v^\mu = \frac{1}{12\pi}\varepsilon^{\mu\nu\rho\sigma}H_{\nu\rho\sigma}, \tag{C.10}$$

where we introduced the gauge-field strength

$$H_{\nu\rho\sigma} = \partial_\nu A_{\rho\sigma} + \partial_\rho A_{\sigma\nu} + \partial_\sigma A_{\nu\rho}, \tag{C.11}$$

and $F = -\epsilon$ is simply the Legendre transform of $P$. As in two dimensions, the entropy current $v^\mu$ in the dual formulation is a topological current. The static background $v^\mu \propto \delta_0^\mu$ therefore corresponds to a constant expectation value for the *magnetic* field $\langle H_{ijk}\rangle \propto \varepsilon_{ijk}$.

We now take the non-relativistic limit as in eq. (4.27). Discarding a total derivative, we find

$$\mathcal{L}_A = F_{NR}\left(B\right) + \frac{mf_{ij}^2}{8\pi B} + \frac{i}{4\pi}A_{\mu\nu}\varepsilon^{\mu\nu\rho\sigma}\partial_\rho \Phi^\dagger \partial_\sigma \Phi, \tag{C.12}$$

where $F_{NR}(B)$ is minus the non-relativistic energy density, and $B$ is the magnetic field $H_{ijk} = \varepsilon_{ijk}B$, such that the mass density is

$$\rho_m = m\frac{B}{2\pi}, \tag{C.13}$$

---

over an arbitrary surface $S$ in $\Phi$ space, we find the conservations of the following charges:

$$\frac{d}{dt}\int_S d^2\Phi_0 v^0 \left|\frac{\partial \Phi}{\partial x}\right|^{-1} = \int_{S(t)} d^2x v^0 = 0, \tag{C.8}$$

where in physical space the surface $S(t)$ is obtained evolving an arbitrary surface at $t = 0$ with the vorticity vector $\omega^\mu/\sqrt{-\omega^2}$. Since the equations of motion (4.19) imply $\omega^\mu \propto S^\mu$, eq. (C.8) states that the particle number is conserved on surfaces that move parallel to the particle flow $S^\mu$. This statement is true for any conserved current; it follows from the fact that $\mathcal{L}_v(\star S) = 0$ for any $v \propto S$, which is a consequence of the trivial identity $S \wedge S = 0$ and current conservation $d \star S = 0$.

and we defined the *magnetostatic* field as

$$f_{ij} = H_{i0j} = \partial_i A_{0j} - \partial_j A_{0i} - \dot{A}_{ij}\,. \tag{C.14}$$

The incompressible limit is obtained setting $\rho_m = \text{const.}$. This is equivalent to neglecting the fluctuations of the purely spatial components of the gauge field $A_{ij}$ as in two dimensions. We rename for simplicity

$$a_i \equiv A_{0i}\,. \tag{C.15}$$

The field $a_i$ was dubbed hydrophoton in [36]. Then we obtain

$$\mathcal{L}_3 = \frac{m^2}{16\pi^2\rho_m}\left(\partial_i a_j - \partial_j a_i\right)^2 + \frac{i}{2\pi}a_i\varepsilon^{ijk}\partial_j\Phi^\dagger\partial_k\Phi + i\frac{\rho_m}{m}\Phi^\dagger\dot{\Phi}\,. \tag{C.16}$$

Integrating out the magnetostatic field $a_i$, one finally obtains the non-local Lagrangian (4.33) in the main text.

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
