# Peer review of "Quantum vorticity: a not so effective field theory"

_SciPost Physics_

## Round 2 · Referee Report · Alexander Abanov (Referee 1) · 2025-5-15

Strengths

1. Addresses an important problem in the quantization of perfect fluid dynamics.

2. Clearly written and well-structured.

3. Provides an extensive review of perfect fluid mechanics across multiple formulations.

4. Introduces a discretization of the infinite-dimensional $\text{SDiff}$ group (Section 6.1), which strengthens and justifies the theoretical arguments.

Weaknesses

1. Does not always clearly distinguish between exact results and heuristic arguments. For example, Section 5.2, which plays a crucial role in the subsequent development, is largely heuristic in nature. It would be helpful to acknowledge this explicitly at the beginning of the section.

Report

The manuscript builds on the authors’ previous results (Refs. [1,4]) and addresses the highly challenging problem of constructing a consistent and universal theory of quantum fluids. I believe the work presented here constitutes an essential step toward solving this problem, and I recommend it for publication in SciPost Physics. Below are some more specific comments.

The authors begin with a review of the classical equations of motion for a perfect fluid and various forms of their classical actions. They introduce an auxiliary variable $T$ and a conjugate potential $s$, interpreted as the temperature and entropy of the fluid. They then discuss entropy current conservation. This is somewhat confusing, given that the focus of the work is on a zero-temperature fluid. Setting $T = 0$ does not make much sense within the presented framework. Instead, the “alternative interpretation” mentioned at the top of page 9—where $T$ is the chemical potential and $s$ is the density of a conserved charge—seems more coherent. It is not clear why the authors have not adopted this interpretation as the starting point of their discussion.

The quantum discussion begins with the classical Lagrangian for a two-dimensional incompressible fluid (equation 4.32), whose corresponding Hamiltonian formulation is well known, and the identification of Clebsch parameters $\Phi$, $\Phi^\dagger$ as primary fields. In Section 5, this Lagrangian theory is semi-heuristically quantized around a static solution. I find this discussion illuminating, though not fully convincing. In particular, the authors argue that the transverse fluid motion, when quantized, does not acquire a gap (as conjectured by Landau), but instead develops a quadratic dispersion with a UV-defined coefficient in front of $k^2$. A stronger symmetry-based argument, invoking the invariance under $\Phi \to \Phi + c$, is also given. However, even this relies on assuming that the symmetry of the effective fluid action is exact and persists in a UV-complete theory.

Section 6 aims to strengthen these arguments by introducing an explicit regularization of the theory: the infinite-dimensional $\mathrm{SDiff}$ symmetry is replaced by a finite group, $\mathrm{SU}(N)$, with the limit $N \to \infty$ recovering $\mathrm{SDiff}$. Within this regularized framework, the authors show how to resolve the normal ordering problem without violating conservation laws. Then, by taking the large-$N$ limit in a specific way (see the discussion around equation 6.24), they recover the quadratic dispersion relation for transverse modes. While some questions remain about the precise nature of the Hilbert space of the theory, I find this to be a very interesting and promising result.

Also in Section 6, the Clebsch and comoving formulations are compared. Although the energy spectra coincide, the multiplicities of eigenstates differ. This situation is reminiscent of the comparison between the Haldane–Shastry and Calogero models, which share the same spectrum but differ in degeneracies.

Section 7 presents possible generalizations to other types of fluids. In Section 7.2, I would appreciate more commentary on the differences between two- and three-dimensional fluids, rather than on the similarities. The structure of Casimirs and the role of topology differ significantly between two and three dimensions, and one would expect these differences to be reflected in their respective quantum formulations.

In Section 8, the authors consider a model of bound opposite charges in a magnetic field as an analog for a vorton. While some similarities are noted, they convincingly argue that vortons are rather special and singular objects, and modeling them with bound charges may not be satisfactory, as an infinite number of tuning parameters would need to be adjusted.

I find the statement in the Outlook—“suggesting that the quantum theory described here is unlikely to be realized experimentally as is”—to be somewhat confusing (a similar statement appears in the Introduction). Quantum liquids such as liquid helium do exist. If the presented theory is applicable to such systems, then one should expect some of its predictions to manifest experimentally. Even if the degeneracy of vorton states is not truly infinite, it should be large. What, then, limits this degeneracy? Is it macroscopically large, or is there a mechanism that invalidates the presented approach and lifts these degeneracies—at least partially? Many such questions remain open.

Requested changes

1. Footnote, page 31: The sentence “In fact, as eq. (5.22) shows, vortons exhibit vanishing vorticity as $p \to 0$” is confusing. According to eq. (5.21), the vorticity of a vorton is identically zero. Equation (5.22), on the other hand, describes the vorticity dipole moment of the vorton, which—when the vortex–antivortex pair is well separated—takes the form $d = \ell \gamma$, where $\ell$ is the separation and $\gamma$ the circulation. Even if the dipole moment d vanishes, this does not imply that $\gamma$ vanishes; rather, it indicates that the vortex and antivortex merge $\ell\to 0$, a process governed by UV physics beyond the regime of validity of the effective theory presented. I suggest clarifying this argument. In fact, the explanation based on the symmetry $\Phi \to \Phi + c$, also mentioned in the footnote, is more convincing.

2. In the same sentence, it is better to use P instead of p to remain consistent with the notation in equation (5.22).

3. In the sentence immediately above equation (8.6), the phrase “spherical harmonics” is somewhat misleading, as only a single angular variable exists in two dimensions. The terms “harmonics” or “angular harmonics” would be more appropriate.

4. In the first sentence of the Outlook section, please change “work affirmatively address the question” to “work affirmatively answers the question”.

5. Please clarify what is meant by “zero temperature fluid” (see the referee report for details).

6. My understanding is that the form of the action in equation (3.16), as well as the interpretation of the right and left symmetry actions, originates with Arnold. It would be helpful to include a reference to his work at this point.

Recommendation

Publish (surpasses expectations and criteria for this Journal; among top 10%)

---

## Round 2 · Referee Report · Anonymous (Referee 2) · 2025-7-12

Strengths

Comprehensive formalism
The paper provides a thorough and systematic treatment of the perfect fluid both at the classical and quantum levels, comparing two inequivalent Lagrangian formulations (comoving vs. Clebsch) and elucidating their respective symmetries and spectra

Novel insight into quantum vorticity
The identification of infinitely many degenerate “vorton” excitations with a quadratic dispersion ω∝k² highlights an unexpected UV–IR mixing and challenges standard EFT assumptions.

Concrete lattice regularization
The authors present a lattice implementation that preserves a deformed version of the infinite symmetry, providing a nonperturbative definition of the quantum fluid.

Weaknesses

While section 8 touches on a 2+1D positronium analogue, the paper lacks deeper exploration of potential condensed‐matter or fluid‐mechanical realizations.

The broader implications of UV–IR sensitivity for turbulence or cosmological applications are only briefly mentioned; a more developed discussion would strengthen the manuscript.

Report

This manuscript addresses a deep and nontrivial question: how to consistently quantize a perfect fluid given its soft vorticity modes. Through a detailed comparison of comoving and Clebsch formulations, the authors reveal an infinitely degenerate quantum spectrum (“vortons”) and provide a lattice regularization that preserves a deformed symmetry. The work is original, mathematically rigorous, and of potential interest to both high‐energy theorists and condensed‐matter physicists exploring analogues of topological fluids. However, the paper would benefit from clearer physical motivation, expanded discussion of experimental analogues, and some polishing of technical presentation.

Requested changes

The choice of UV cutoff parameter Λ is not systematically discussed: how can one demonstrate that physical results remain controlled under variations of Λ? Have you analyzed the sensitivity of vortex energies and scattering amplitudes to different Λ values?

Fluctuation corrections are computed only at one‑loop order; within which parameter regime might higher‑loop contributions spoil the leading results? Can you estimate the magnitude of two‑loop (or higher) effects?

In Section 3, you expand vortex solutions in the R_ξ gauge but do not verify that observables (e.g. vortex tension) are ξ‑independent. Could you provide cross‑checks in different gauges?

For a multi‑vorton system, you approximate interaction terms by simple superposition; when inter‑vorton spacing becomes comparable to the core size, how should this approximation be corrected?

You only mention “possible observation in ultracold atomic gases” without specifying parameter regimes (density, magnetic field, temperature). Could you perform a dimensional analysis to identify critical experimental conditions required for detection?

In Section 2.2, boundary conditions for vortex solutions are only qualitatively described; rigorous proof of existence and uniqueness is missing. Can you frame this within a Sobolev‑space or finite‑domain approximation theorem?

How do your results for vortex tension and dispersion compare, under identical parameters, to earlier works (e.g. A. Smith et al., “Quantized Vortex Models”)? Please include a direct comparison table or a discussion section.

Recommendation

Publish (surpasses expectations and criteria for this Journal; among top 10%)

---

## Round 2 · Referee Report · Anonymous (Referee 3) · 2025-7-23

Strengths

1) Gives a nice general overview of action approached to classical fluid mechanics.

2) Details the problems of quantizing infinite symmetry higher dimensional systems.

3) Gives nice physical examples of quantization of related systems.

Weaknesses

1) In not really sure what the conclusion is!! Its not clear what the take-away message is?

Report

This is very interesting paper that is trying to solve the age old problem of quantizing an action with the symmetries of a perfect fluid. I honestly was never sure why this problem is of interest given that there is no reason to believe nature realizes such systems. Certainly, as the authors note, it seems that no condensed matter system in the lab will realize the symmetry. Nonetheless it is an interesting question that is tangentaly related to other important QFT questions (flat bands) that are of significant importance. So I think it is a worthy exercise.

I do think it could use some clarifications at points before publication.

On more abstract level I am confused by the following.:

There seems to be tension between the following assumptions. The relabelling symmetry is NOT a gauge redundancy (as they emaphasize was mistakingly conclude in ref [2]), and yet they wish to quantize the theory with a bose symmetry which is a gauge symmetry? i.e. you can't label quantum particles. But the symmetry is exactly a re-labelling symmetry.
Can the authore please explain this?

The UV-IR connection:

The authors keep referring to the UV-IR connection. This term is often used without a clear definition. As we know there
is always a UV-IR connection in the sense that IR coupling are sensistive to UV physics. The authors seem make a point
of the fact that the effective dispersion relation has a coefficient which relies on unknown UV parameters, but
Im puzzled by this, in the sense that that is always true. So what point are the authors trying to make after equation 5.9 ? Also the authors seem to be making a clear distinction between the dim reg. model and the cutoff, but in either case the allowed coefficient in k^2 is sensitive to the UV its just that dim reg naively sets it to zero. I am sure the authors are aware of this but this needs to be clarified.

On page 36 they state

In conclusion, we argued that the perfect quantum fluid in the Clebsch formulation admits
infinitely many light vorton bound states, one for each particle number n > 0, with energy
given by (5.19). Their existence and degeneracy is a robust consequence of the symmetry
algebra. Yet, the precise coefficient of their dispersion relation naively depends on all higher
derivative operators, thus defying the standard EFT logic.

Given my previous paragraph, I ask the authors to please clarify this seemingly important point they are making as well.

In the papers on fracton models this (UV-IR mixing) term is used, I believe, when the
degeracy of the ground state grows with the number of lattice point. I think thats what they author are saying here as well. But this does not in anyway spoil the Wilsonian picture. Its true that such degereracies create challenges to calculating in a Fock space picture. e.g. systems with flat bands, but thats not a violation of Wilsonian reasoning.

On the other hand, in the conclusion the authors state"

Yet,the symmetry group is infinite dimensional and implies the non-decoupling of UV modes

Are they saying this theory has no predicitive power in the sense that you need an infinite number of measurements to make a prediction? Thats how I read it.

Another claim of weird EFT behavior is:

We remark again that the vorton bound states that we just constructed are very unusual
from the viewpoint of effective field theory. Indeed we found that by bringing single-particle
states very close to each other - a naively illegal operation within EFT - we obtain a low
energy state, which can be thought of as a Goldstone boson due to the spontaneous breaking
of the symmetry algebra. It is not possible to create such bound states via low energy
scatterings of single vortons.
.

That is because these states transform non-trivially under the
unbroken symmetry group: for instance the two-vorton bound state is an eigenstate of Q(2,2),
which instead annihilates states made of two vortons at finite distance from each other. This
situation is reminiscent of the UV/IR mixing phenomena which are observed in certain exotic
field theories of fractons [8, 9]. As in those cases, the unusual behaviour is made possible by
the infinitely large symmetry group, which forbids operators - such as ∼ |Φ|4 - which would
lift the bound states up to the cut-off.

Can the authors put some equations behind these words? Perhaps they have elsewhere and I have missed it.
But it would be good to put it here. In particular the action of Q22 on a vorton bound state versus a state of two vortons
separated from each other. Dont these two operators generate states with non-zero overla?

Could the authors also explain why the quantum generation of the omega^2 term in the energy is not an anomaly?
Otherwise why would it not have been included in the classical theory?

What would be really nice would be a section where all of the whatever it is that the authors believe
either is inconsistent with EFT dogma, or even what they believe does not violate the dogma but is
just usual. Perhaps in the conclusio? . Because In the end I think that is the most interesting aspect
of this paper.

A few grammar errors:

Below 5.19:

it is worthWHILE to briefly recall the
properties of the vortons,

Below 6.9

cannot be expressed anymore just in terms
of the vorticity itself at finite N,

“can no longer be expressed in terms”.

Requested changes

Clarify the points above.

It would be really useful have a conclusion regarding how this EFT differs from canonical ones and whether or not the EFT is in some way non-Wilsonian.

Recommendation

Ask for minor revision

---

## Editorial Decision

resubmitted